# Tumor-intrinsic YTHDF1 drives immune evasion and resistance to immune checkpoint inhibitors via promoting MHC-I degradation

Wanzun Lin [1,2,3], Li Chen[1,2,3], Haojiong Zhang[2,3,4], Xianxin Qiu[2,3,4], Qingting Huang[2,3,4], Fangzhu Wan[1,2,3], Ziyu Le[1,2,3], Shikai Geng[1,2,3], Anlan Zhang[1,2,3], Sufang Qiu[5], Long Chen[6], Lin Kong [1,2,3] ✉ & Jiade J. Lu [2,3,4] ✉

The recently described role of RNA methylation in regulating immune cell infiltration into tumors has attracted interest, given its potential impact on immunotherapy response. YTHDF1 is a versatile and powerful m6A reader, but the understanding of its impact on immune evasion is limited. Here, we reveal that tumor-intrinsic YTHDF1 drives immune evasion and immune checkpoint inhibitor (ICI) resistance. Additionally, YTHDF1 deficiency converts cold tumors into responsive hot tumors, which improves ICI efficacy. Mechanistically, YTHDF1 deficiency inhibits the translation of lysosomal genes and limits lysosomal proteolysis of the major histocompatibility complex class I (MHC-I) and antigens, ultimately restoring tumor immune surveillance. In addition, we design a system for exosome-mediated CRISPR/Cas9 delivery to target YTHDF1 in vivo, resulting in YTHDF1 depletion and antitumor activity. Our findings elucidate the role of tumor-intrinsic YTHDF1 in driving immune evasion and its underlying mechanism.

One of the hallmarks of cancer is avoiding immune destruction[1]. To evade immune surveillance, tumor cells use intrinsic modulators to establish an inhibitory microenvironment. Therefore, the development of new therapeutic strategies that restore antitumor immunity holds promise for cancer treatment. Indeed, immune checkpoint inhibitors (ICIs), therapeutic vaccines, T-cell transfer-based therapies, monoclonal antibodies, and chimeric antigen receptor-T cell therapy are revolutionizing cancer therapy[2–5]. However, only certain patients develop objective tumor responses that lead to improved long-term survival. The genes implicated in immune evasion are becoming a topic of extensive investigation due to the potent efficacy of immunotherapies in some patients[6].

N6-methyladenosine (m6A) is recognized to be the most common, abundant, and preserved endogenous modification in eukaryotic RNAs[7]. Changes in m6A levels could have a significant impact on cancer hallmarks, such as maintaining proliferative signaling, resisting cell death, facilitating replicative immortality, triggering invasion and metastasis, and altering energy metabolism[8–12]. Recent studies highlight the important function of m6A methylation in evading immune-mediated destruction. Tumors exploit the fat mass and obesity-associated protein (FTO)-mediated regulation of glycolytic metabolism to evade immune surveillance[13]. YTHDF1 is one of the most critical m6A reader proteins that preferentially bind to methylated RNA and facilitate translation[14,15]. Although accumulating studies sufficiently

[1]Department of Radiation Oncology, Shanghai Proton and Heavy Ion Center, Fudan University Cancer Hospital, Shanghai 201321, China. [2]Shanghai Key Laboratory of Radiation Oncology (20dz2261000), Shanghai 201321, China. [3]Shanghai Engineering Research Center of Proton and Heavy Ion Radiation Therapy, Shanghai 201321, China. [4]Department of Radiation Oncology, Shanghai Proton and Heavy Ion Center, Shanghai, China. [5]Department of Radiation Oncology, Fujian Cancer Hospital & Fujian Medical University Cancer Hospital, Fuzhou 35005, China. [6]Department of Neurosurgery & Neurocritical care, Huashan Hospital, Fudan University, Shanghai, China. ✉e-mail: lin.kong@sphic.org.cn; jiade.lu@sphic.org.cn

support the oncogenic functions of YTHDF1 in promoting the proliferative, invasive, and metastatic abilities of tumors, these reports mainly evaluated the alterations in cancer cells, neglecting the contributions of the tumor microenvironment (TME)[8,16–19]. More recently, the function of YTHDF1 in regulating antitumor immunity has received increasing attention. *Ythdf1* deficiency in dendritic cells (DCs) was shown to enhance the cross-priming of CD8[+] T cells and the cross-presentation of tumor antigens in vivo[20]. Nevertheless, the crosstalk between tumor-intrinsic YTHDF1 and the immune microenvironment remains to be fully elucidated, given that tumor cell-intrinsic mechanisms are the key factors contributing to the evasion of immune surveillance.

m6A modulators have attracted considerable interest as therapeutic targets on the basis of accumulating data on their functions and involvement in cellular processes in cancer. m6A inhibitors, particularly inhibitors of FTO, including Dac51, FB23, FB23-2, CS1, CS2 and meclofenamic acid, have been developed for cancer therapy and have shown much promise in preclinical trials[13,21,22]. By specifically binding to FTO and selectively suppressing its m6A demethylase action, FB23 and FB23-2 significantly inhibit proliferation and mediate apoptotic cell death in human acute myeloid leukemia cells. While prior research has primarily been centered on FTO antagonists, other m6A-related proteins could be better targets in m6A-related malignancies. YTHDF1 is perhaps the most viable m6A reader protein, although there are few possible therapeutic antagonists of YTHDF1.

In this study, we aim to (i) validate the function of tumor-intrinsic YTHDF1 in driving immune evasion and elucidate its underlying mechanism, (ii) analyze the association between YTHDF1 expression and response to ICI therapy, and (iii) exploit efficient strategies that block YTHDF1 in vivo.

## Results

### YTHDF1 overexpression is associated with an "immune desert" phenotype and resistance to immune checkpoint inhibitor

Recent studies have highlighted the critical function of RNA methylation in driving immune evasion. In The Cancer Genome Atlas (TCGA)-skin cutaneous melanoma (SKCM) dataset, we initially calculated immune scores to infer the fractions of immune cells in tumor samples using an immune signature of 141 reported genes[23]. Among m6A regulators, *YTHDF1* RNA showed the most obvious inverse correlation with the immune score, implying its potential role in regulating immune evasion (Fig. 1A). We further performed a comprehensive analysis of 33 types of tumors from the TCGA and found that *YTHDF1* RNA showed a significantly negative correlation with the immune score in 26 of the tumor types (Supplementary Fig. 1A). Next, we generated a heatmap to visualize the relative abundance of 28 infiltrating immune cell types in high and low *YTHDF1* RNA expression groups using single-sample gene set enrichment analysis (ssGSEA). Overall, the *YTHDF1*-low samples exhibited a significant level of immune cell infiltration, indicating that they had an immune-inflammation phenotype, whereas the *YTHDF1*-high samples had a lower level of immune cell infiltration, indicating that they had an "immune desert" phenotype (Supplementary Fig. 1B). In addition, the expression of *YTHDF1* was negatively associated with the expression of *CD8A*, *CD8B*, *GZMA*, *GZMB*, and *TNFRSF18* and the activated CD8 T-cell score (Fig. 1B, C). The association between YTHDF1 expression and CD8[+] T-cell infiltration was then visualized by immunofluorescence (Fig. 1D).

We investigated the association between YTHDF1 expression and the anti-PD-1 response rate, starting with the hypothesis that YTHDF1 confers an "immune desert" phenotype and drives resistance to ICIs. We investigated RNA-sequencing (RNA-seq) data from two available cohorts of melanoma patients given anti-PD-1 therapy, the Hugo et al. cohort ($n = 18$) and the Riaz et al. cohort ($n = 56$), and the clinical response data of the patients were available[24,25]. The results

revealed that *YTHDF1* was overexpressed in progressive disease (PD) and showed a gradual decrease in expression from the PD group to the complete response (CR) group (Fig. 1E, F). *Z* scores revealed that *YTHDF1* expression was differentially enriched between the tumor groups, and most patients with PD had a positive *z* score (Fig. 1G). To evaluate the predictive efficiency of *YTHDF1* expression for the anti-PD-1 response rate, we generated a receiver operating characteristic (ROC) curve. The area under the ROC curve was almost 0.7, indicating a high predictive value (Fig. 1H). With a cutoff value of 7.98, patients were divided into low or high-expression groups. The patients in the *YTHDF1*-high group presented a 12% clinical response rate, while those in the *YTHDF1*-low group had a 57% clinical response rate (Fig. 1I). When analyzing the TCGA-SKCM dataset, we used the tumor immune dysfunction and exclusion (TIDE) method to assess the potential clinical efficacy and response to immunotherapy in different *YTHDF1* subgroups[26]. High *YTHDF1* expression was linked to resistance to immunotherapy, which was consistent with the results for the Hugo et al. cohort and the Riaz et al. cohort (Supplementary Fig. 1C, D). For further validation, we developed an in vivo mouse tumor model that was resistant to ICI therapy. We utilized the B16/F10 melanoma model and performed serial in vivo and in vitro passaging in the presence of an anti-PD-L1 and anti-CTLA-4 antibody. Following seven cycles of passaging, B16/F10 cell lines exhibiting resistance to ICI treatment (ICIs-resistant B16/F10) were derived, and were unresponsive to combination treatment in vivo (Fig. 1J). Western blot analysis showed that YTHDF1 was significantly upregulated in resistant B16/F10 tumors (Fig. 1J).

We then explored the prognostic value of *YTHDF1*. Indeed, high *YTHDF1* expression was associated with a dismal prognosis in the GSE65904 SKCM dataset (Fig. 1K). In the TCGA-SKCM dataset, even though no significant differences were observed ($p = 0.094$), the patients with low *YTHDF1* expression exhibited a favorable prognosis (Supplementary Fig. 1E). In addition, subgroup analysis revealed the significant prognostic value of *YTHDF1* in the *TGFB1*-low group but not in the *TGFB1*-high group, which indicated that the prognostic impact of *YTHDF1* is potentially associated with immunity (Supplementary Fig. 1F).

We also analyzed the associations between the response to ICI therapy and other m6A regulators. In the Hugo et al. cohort, most m6A regulators showed no significant association with resistance to ICI therapy, and an inverse correlation was observed for *YTHDC2* and *WTAP* (Supplementary Fig. 2A–C). In the Riaz et al. cohort, the expression of most m6A regulators was dysregulated in the PD and partial response (PR)/CR groups, but no difference was observed between the PD and stable disease (SD) groups (Supplementary Fig. 2D, E). These results suggest that YTHDF1 plays a more important role in ICI therapy resistance than do other m6A regulators.

### Tumor-intrinsic YTHDF1 deficiency inhibits tumorigenesis in immunocompetent mice

Tumor cells usually use intrinsic modulators to evade immune surveillance. To elucidate the biological functions of tumor-intrinsic YTHDF1 in tumorigenesis and immune evasion, we initially used the CRISPR/Cas9 gene editing system to deplete *Ythdf1* gene in the B16/F10 cell line. As shown in Supplementary Fig. 3A, B, a fragment knockout (KO) strategy was used to target exon 4 of the *Ythdf1* gene. The workflow presents the construction of the *Ythdf1* KO cell line (Supplementary Fig. 3C). Finally, single *Ythdf1*-KO clones with a deletion of 767 bp in exon 4 were generated, as demonstrated by Sanger sequencing and polymerase chain reaction (PCR) (Supplementary Fig. 3D, E). Western blotting showed that the YTHDF1 protein was completely depleted without affecting YTHDF2 or YTHDF3 expression (Supplementary Fig. 3F).

Next, we investigated the function of YTHDF1 in cell proliferation in vitro by performing (i) cell counts, (ii) cell counting kit-8

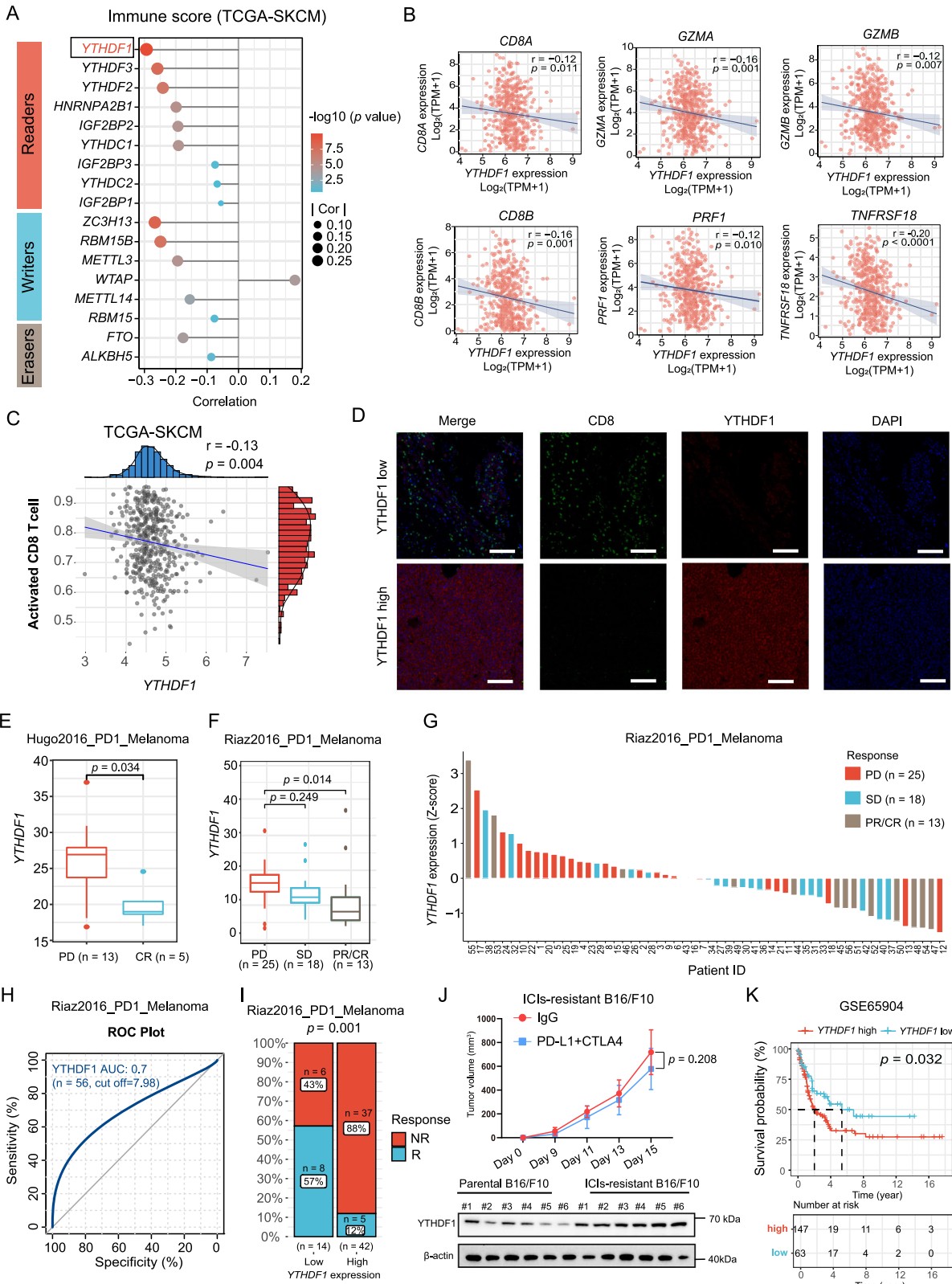

(CCK8) assays, (iii) clone formation assays, and (iv) EdU incorporation assays. However, no difference in proliferation or viability was observed between *Ythdf1*-KO B16/F10 cells and wild-type (WT) cells (Supplementary Fig. 3G–J). Then, immunocompetent mice were subcutaneously injected with *Ythdf1*-KO cells or WT cells. Interestingly, tumor-intrinsic YTHDF1 deficiency significantly inhibited tumorigenesis in immunocompetent mice, as evidenced

by a lower tumor volume (Fig. 2A), tumor weight (Fig. 2B), and luminescence intensity (Fig. 2C, D) and a favorable survival time (Fig. 2E) in the *Ythdf1*-KO group. Similar results were validated with lung metastasis models (Fig. 2F–H). In contrast to the results observed with immunocompetent mice, there were no differences in tumor volume or weight between *Ythdf1*-KO cells and WT cells in immunodeficient mice (Fig. 2I, J).

**Fig. 1 | YTHDF1 overexpression is associated with an "immune desert" phenotype and resistance to ICI therapy. A** Correlations of m6A regulator expression with immune scores in the TCGA skin cutaneous melanoma (TCGA-SKCM) dataset ($n = 471$ biologically independent samples). Two-tailed Spearman correlation is reported. **B** Scatter plot presenting the correlation of *YTHDF1* with activated CD8+ T-cell-related gene expression. The expression levels were $log_2(TPM+1)$ derived from 471 TCGA melanoma samples. Two-tailed Spearman correlation is reported. **C** Scatter plot presenting the correlation between *YTHDF1* and activated CD8 T cell score. Two-tailed Spearman correlation is reported. **D** Immunofluorescence staining for YTHDF1 protein expression (red) and CD8 T-cell infiltration (green) using a melanoma tissue microarray. Representative immunofluorescence images of 48 independent melanoma tissues. Scale bar, 100 μm. **E, F** Box plots showing the differential expression of *YTHDF1* in progressive disease (PD), stable disease (SD), partial response (PR), and complete response (CR) samples using RNA-sequencing data from **E** Hugo2016_PD1_Melanoma cohort[24] ($n = 18$ biologically independent samples) and **F** Riaz2016_PD1_Melanoma cohort[25] ($n = 56$ biologically independent samples). The upper, middle and lower horizontal lines of the box represent the upper, median and lower quartile respectively. Whiskers depict the smallest or largest values within 1.5-fold of the interquartile range, and the points outside the box represent outliers. Two-tailed unpaired Student's *t*-test (**E**) and Kruskal–Wallis with Dunn's multiple comparison test (**F**) are reported. **G** A waterfall plot depicting *YTHDF1* expression (*z* score) for the PD, SD and PR/CR samples in the Riaz2016_PD1_Melanoma cohort ($n = 56$ biologically independent samples). **H** ROC curve indicating the predictive value of *YTHDF1*. **I** Bar plot presenting the different response rates of ICI therapy. The blue bar represents responders (R), and the red bar indicates nonresponders (NR). Two-tailed $\chi^2$ test. **J** Tumor growth curves of ICIs-resistant B16/F10 cells ($n = 6$ biologically independent animals per group). Two-sided unpaired Student's *t*-test. Data are presented as mean values ± SD. Western blot revealing YTHDF1 expression in ICIs-resistant and parental B16/F10 cells. **K** Kaplan–Meier OS curve for 210 melanoma patients with high or low *YTHDF1* expression. Two-sided log-rank test. Source data are provided as a Source Data file.

These results imply that the oncogenic function of tumor-intrinsic YTHDF1 is involved in antitumor immunity.

## Tumor-intrinsic YTHDF1 deficiency triggers antitumor immunity

We hypothesized that the tumor-suppressive function of YTHDF1 deficiency is mainly attributed to the induction of antitumor immunity based on the fact that *Ythdf1* deficiency inhibited tumor growth in immunocompetent mice but not in immunodeficient mice or in vitro. To further investigate our hypothesis, we initially performed transcriptome analysis to screen the differential biological processes and pathways between *Ythdf1*-KO and WT tumor tissues in immunocompetent mice. In total, we identified 837 differentially expressed genes (DEGs) including 665 upregulated genes and 172 downregulated genes (Fig. 3A). Among the DEGs, genes encoding cytotoxic molecules (e.g., *Gzmb*, *Gzmf*, *Prf1*, *Cd8a*, and *Cd8b1*) exhibited enhanced expression. By overlapping our results with annotated immune-relevant genes, we identified 133 immune-relevant genes that were upregulated and seven that were downregulated in *Ythdf1*-KO tumors (Supplementary Fig. 4A, B). Consistently, pathway enrichment analysis revealed that immune-associated pathways were upregulated in *Ythdf1*-KO tumors (Fig. 3B). Similarly, in the TCGA-SKCM dataset, gene sets differentially enriched in the low *YTHDF1* expression group were related to cytokine–cytokine receptor interaction, antigen processing and presentation, inflammatory response, and IL-6-JAK-STAT3 signaling, all of which mainly stimulate antitumor immunity (Fig. 3C). These results revealed that immunological processes and pathways were activated in *Ythdf1*-KO tumors in immunocompetent mice. Indeed, flow cytometric analysis revealed increased levels of tumor-infiltrating CD4+ and CD8+ T cells (Fig. 3D), which were further visualized by immunohistochemical staining (Supplementary Fig. 4C, D).

Next, we analyzed cytokine expression based on the observation that the cytokine–cytokine receptor interaction pathway was activated in *Ythdf1*-KO tumors. A heatmap of RNA-seq data showed that most of the interleukin family and chemokine family were upregulated in *Ythdf1*-KO tumor tissues (Fig. 3E). In line with the RNA-seq data, the protein levels of cytokines were upregulated in *Ythdf1*-KO tumor tissues (Fig. 3F). These results indicated that tumor-intrinsic YTHDF1 deficiency suppressed tumor growth by triggering immunity.

## Single-cell RNA-sequencing analysis reveals a distinct immune landscape between YTHDF1-deficient and WT tumors

To reveal immune microenvironment heterogeneity at single-cell resolution, we performed single-cell transcriptome analysis of tumor-infiltrating CD45+ immune cells from *Ythdf1*-KO ($n = 3$) and WT ($n = 3$) tumors (Fig. 4A). A total of 29325 CD45+ immune cell profiles were obtained from six tumor samples (WT1, $n = 5197$; WT2, $n = 5358$; WT3, $n = 3917$; KO1, $n = 5131$; KO2, $n = 6432$; and KO3, $n = 3290$).

Unsupervised t-distributed stochastic neighbor embedding (t-SNE) analysis of the total cell populations identified 22 subclusters with 10 immune cell populations, including T cells, B cells, natural killer (NK) cells, neutrophils, monocytes/macrophages (Monos/Macros), proliferating mixed cells, plasmacytoid dendritic cells (pDCs), Clec9a+ conventional type 1 dendritic cells (cDC1s-Clec9a), Ccl22+ conventional type 1 dendritic cells (cDC1s-Ccl22), and conventional type 2 dendritic cells (cDC2s) (Fig. 4B). Overall, the proportions and cell numbers for different cell categories exhibited apparent differences between the KO and WT groups (Fig. 4C, D). Of note, T cells, NK cells, and proliferating mixed immune cells were enriched in *Ythdf1*-KO tumors, whereas neutrophils and B cells showed higher infiltration in WT tumors (Fig. 4D).

To gain deeper insight into the heterogeneity of the TME, we performed subgroup analyses of CD4+ T cells, CD8+ T cells, neutrophils and B cells. Among the CD4+ T cells, seven clusters including five CD4+ T-cell immune subpopulations (Fig. 4E) such as naive CD4+ T cells, effector memory CD4+ T cells (CD4+ TEMs), regulatory CD4+ T cells (Tregs), proliferating CD4+ T cells, and IFN-responsive CD4+ T cells were identified based on known markers (Supplementary Fig. 5A, B). We observed distinct proportions and cell numbers for the CD4+ subpopulation between the KO and WT groups (Fig. 4F). Most remarkably, naive CD4+ T cells composed 60% of the total CD4+ T-cell population in WT tumors, while only 6% of the CD4+ T-cell population consisted of naive CD4+ T cells in *Ythdf1*-KO tumors (Fig. 4G). Moreover, functional CD4+ T cells, including CD4+ TEMs, Tregs, proliferating CD4+ T cells, and IFN-responsive CD4+ T cells, were obviously expanded in *Ythdf1*-KO tumors (Fig. 4G). Interestingly, although Treg infiltration was increased in the *Ythdf1*-KO group, recent studies have identified a novel Treg subpopulation, fragile Tregs, with the particular function of promoting antitumor immunity[27,28]. Fragile Tregs are defined as cells that retain Foxp3 expression but exhibit a deficiency in NRP1 and decreased expression of suppressive molecules, such as CD73 and IL-10[27]. In our study, most of the increased Tregs were characterized by the depletion of *Nrp1*, *Cd73*, *Il-10* and *Il-4* and increased expression of *Ifngr* and *Gzmb*, so they might be defined as fragile Tregs (Supplementary Fig. 5C). In addition, Kyoto Encyclopedia of Genes and Genomes (KEGG) enrichment analysis revealed that an array of immune-associated pathways (such as antigen processing and presentation, Th1- and Th2-cell differentiation, and T cell receptor signaling pathway) were upregulated in TEMs, Tregs, proliferating CD4+ T cells, and IFN-responsive CD4+ T cells but were downregulated in naïve CD4+ T cells (Fig. 4H). In addition, cell proliferation-related pathways (such as pathways related to the spliceosome, DNA replication, and cell cycle) were specifically enriched in proliferating CD4+ T cells, indicating a high proliferative capacity. By pseudotime analysis, naive CD4+ T cells were located at the beginning of the pseudotime trajectory, while Tregs were located in the terminally differentiated

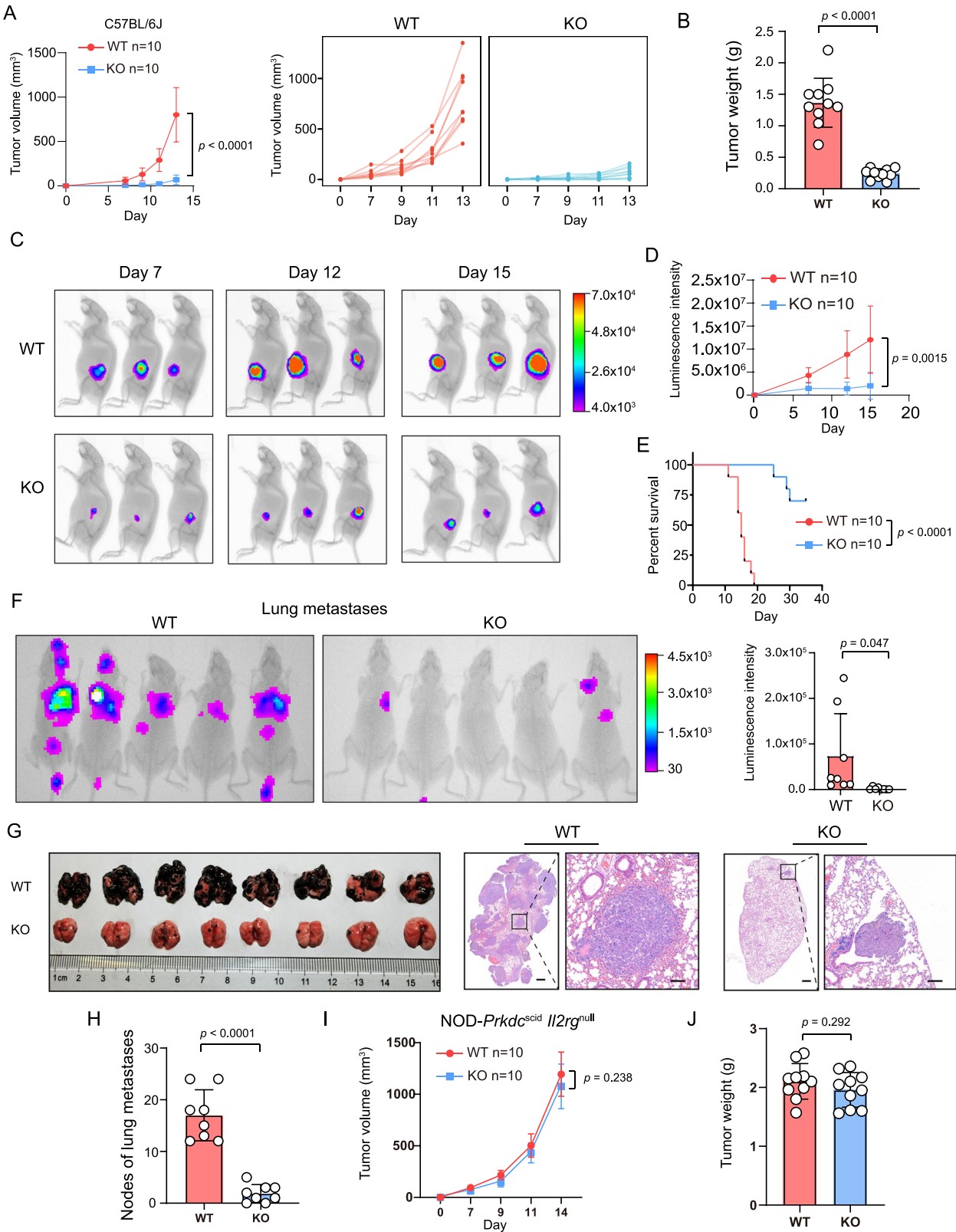

state of the branch, with TEMs and IFN-responsive CD4+ T cells being transition states spread along the axis (Supplementary Fig. 5D, E). When clustered by group, the naive CD4+ T cells were mainly distributed in the WT group, whereas CD4+ TEMs were abundantly enriched in the *Ythdf1*-KO group (Supplementary Fig. 5F). Next, we analyzed the DEGs in the *Ythdf1*-KO and WT groups in terms of different immune subpopulations. Of note, even when evaluating the

same immune subpopulations, a substantial number of DEGs was identified between the *Ythdf1*-KO and WT groups (Supplementary Fig. 5G). Specifically, for the proliferating CD4+ T-cell and CD4+ TEM subtypes, Gene Ontology (GO) functional analysis of the DEGs showed that immune pathways (such as T cell proliferation, T cell activation, and antigen processing and presentation) were upregulated in the KO group (Supplementary Fig. 5H). These results indicated that the

**Fig. 2 | Tumor-intrinsic YTHDF1 deficiency inhibits tumorigenesis in immuno-competent mice. A** Tumor growth curves of wild-type (WT) and *Ythdf1*-knockout (KO) B16/F10 melanoma cells subcutaneously inoculated into C57BL/6J mice (n = 10 biologically independent animals per group). Two-tailed unpaired Student's *t*-test. Data are presented as mean values ± SD. One of three representative experiments with similar results is shown. **B** Tumor weight of C57BL/6 mice in the WT and KO groups (n = 10 biologically independent samples per group). Two-tailed unpaired Student's *t*-test. Data are presented as mean values ± SD. **C, D** In vivo biolumines-cence imaging. Luciferase-labeled WT or KO B16/F10 cells were subcutaneously inoculated into C57BL/6J mice. The signals were measured on days 7, 12 and 15 using an In-Vivo Xtreme imaging system (**C**), and the luminescence intensities (**D**) obtained for the WT and KO groups were further compared (n = 10 biologically independent animals per group). Two-tailed unpaired Student's *t*-test. Data are presented as mean values ± SD. **E** Kaplan–Meier curves of C57BL/6J mice inoculated with WT or KO B16/F10 cells. Tumor volumes exceeding 1500 mm³ were considered events (n = 10 biologically independent animals per group). Two-tailed log-rank (Mantel–Cox) test. **F** In vivo bioluminescence image of lung metastases. Luciferase-labeled WT or KO B16/F10 cells were intravenously injected into C57BL/6J mice via the tail vein. Two weeks later, the growth of experimental lung metastases was measured using an In-Vivo Xtreme imaging system (n = 8 biologically independent animals per group). Two-tailed unpaired Student's *t*-test. Data are presented as mean values ± SD. **G, H** Images and hematoxylin & eosin staining of lung metastases (**G**) and statistical analysis of lung metastatic nodules (**H**). n = 8 biologically inde-pendent samples per group. Scale bars, 500 and 100 μm. Two-tailed unpaired Student's *t*-test. Data are presented as mean values ± SD. **I** Tumor growth curves of WT and KO B16/F10 melanoma cells subcutaneously inoculated into NOD-*Prkdc*^scid^-*Il2rg*^null^ mice (n = 10 biologically independent animals per group). Two-tailed unpaired Student's *t*-test. Data are presented as mean values ± SD. One of three representative experiments with similar results is shown. **J** Tumor weights of NOD-*Prkdc*^scid^-*IL2rg*^null^ mice in the WT and KO groups (n = 10 biologically independent samples per group). Two-tailed unpaired Student's *t*-test. Data are presented as mean values ± SD. Source data are provided as a Source Data file.

proliferating CD4⁺ T-cell and CD4⁺ TEM subtypes in the KO group exerted more powerful immune functions than those in the WT group. Indeed, flow cytometry validated that tumor-infiltrating CD4⁺ T cells from the *Ythdf1*-KO group exhibited strong production of IFN-γ and granzyme B (Fig. 4I). More importantly, by concomitant T-cell receptor (TCR) sequencing, we determined that CD4 TCR clonotype expansion was increased in *Ythdf1*-KO tumors (Fig. 4J, K), and were mostly dis-tributed in functional CD4⁺ T cells rather than naive cells (Fig. 4L), indicative of ongoing immune responses.

In the CD8⁺ T-cell subgroup analysis, we identified six clusters including 4 CD8⁺ T-cell immune subpopulations (Fig. 5A) such as effector memory CD8⁺ T cells (CD8⁺ TEMs), exhausted CD8⁺ T cells (CD8⁺ TEXs), proliferating CD8⁺ T cells, and IFN-responsive CD8⁺ T cells based on known markers (Supplementary Fig. 6A, B). Overall, *Ythdf1*-KO tumors presented a greater number of infil-trating CD8⁺ T cells (Fig. 5B). KEGG enrichment analysis revealed that a panel of immune-associated pathways (such as antigen pro-cessing and presentation, chemokine signaling pathway, and T cell receptor signaling pathway) were upregulated in CD8⁺ TEXs, IFN-responsive CD8⁺ T cells, and proliferating CD8⁺ T cells, which exhibited an activated effector state (Fig. 5C). Tumor-infiltrating lymphocytes from *Ythdf1*-KO tumors were highly enriched in acti-vated effector T-cell populations, which were assigned largely to the CD8⁺ TEM, and proliferating CD8⁺ T-cell states (Fig. 5D). We further identified the DEGs in different CD8⁺ subpopulations between the WT and *Ythdf1*-KO groups, and a substantial number of DEGs were identified (Supplementary Fig. 6C). Specifically, in the CD8⁺ TEM and CD8⁺ TEX subtypes, GO analysis of the DEGs showed that immune pathways (such as T cell activation and antigen pro-cessing and presentation) were upregulated in the KO group (Supplementary Fig. 6D). These results indicated that the pro-liferating CD8⁺ T-cell and CD8⁺ TEM subtypes in the KO group exerted more powerful immune functions than those in the WT group. Flow cytometry further validated that tumor-infiltrating CD8⁺ T cells from the KO group presented strong production of IFN-γ and granzyme B (Fig. 5E). In vitro cytotoxicity assays showed that tumor-infiltrating CD8⁺ T cells from KO tumors exhibited a greater in vitro killing capacity than those from WT tumors (Fig. 5F). In TCR-sequencing analysis, CD8 TCR diversity was decreased and clonotype expansion was elevated in *Ythdf1*-KO tumors (Fig. 5G, H), and was highly distributed in TEX, indicating antitumor-specific response (Fig. 5I). Given the critical function of CD8⁺ T cells in antitumor immunity, we next depleted CD8⁺ T cells in vivo by administrating anti-mouse CD8 monoclonal antibodies (mAbs) and subsequently monitoring tumor growth. Indeed, the depletion of CD8⁺ cells largely rescued *Ythdf1*-KO tumor growth (Fig. 5J). Notably, after the depletion of CD8⁺ cells, differential tumor growth was still observed between the WT and KO groups,

which indicated that other immune cells might also be involved in tumor suppression.

Neutrophils can exert protumor effects, including supporting angiogenesis, extracellular matrix remodeling, distant metastasis and immunosuppression. To gain further insight, we identified eight clusters (Neu_0-7) based on unsupervised t-SNE analysis (Supplementary Fig. 7A, B). Among these clusters, Neu_3, Neu_4, and Neu_7 highly expressed *Ccl3*, *Ctsb* and *Cstb* and were defined as tumor-specific neutrophils according to a previous report (Sup-plementary Fig. 7C)[29]. Here, we observed abundant infiltration of neutrophils, especially tumor-specific neutrophils, in WT tumors (Supplementary Fig. 7D). Of note, these tumor-specific neutrophils might act as tumor promoters by overexpressing *Vegfa*, *Cd274*, *Pdcd1lg2*, and *Lgals3*, leading to these cells being further defined as tumor-promoting neutrophils (Supplementary Fig. 7E). We also identified *Fabp5* as a specific marker of tumor-promoting neu-trophils (Supplementary Fig. 7F) and found that it was associated with a dismal prognosis in the TCGA-SKCM and GSE65904 SKCM datasets (Supplementary Fig. 7G). The abundant tumor-promoting neutrophil infiltration might account for the overwhelming tumor growth in the WT group.

As an integral component of the TME, tumor-infiltrating B lymphocytes exist in all stages of cancer and play important roles in shaping tumor development. Here, we observed abundant infiltra-tion of B cells in WT tumors (Supplementary Fig. 8A). Strikingly, most B cells expressed naive B-cell markers (*Cd19*, *Ighd*, and *Cd20*) without expressing activated B-cell markers (*Cd25* and *Cd30*), plasma cell markers (*Ighg1* and *Cd27*), or memory cell markers (*Cd27*, *Igha*, and *Ighg1*), leading to them being defined as naive-like B cells (Supplementary Fig. 8B, C). Moreover, these cells highly expressed immunosuppressive molecules such as *Cd274* and *Tgfb1* (Supplementary Fig. 8D).

Overall, single-cell sequencing analysis revealed that tumor-intrinsic YTHDF1 deficiency promoted the infiltration of antitumor immune cells and reduced tumor-promoting immune cells via TME remodeling.

**Tumor-intrinsic YTHDF1 deficiency enhances responses to ICI therapy in vivo**
ICI therapy has become a promising therapeutic strategy achieving encouraging therapeutic outcomes due to its durable antitumor effects. Based on our results showing that tumor-intrinsic YTHDF1 contributes to immune evasion and is associated with ICI resistance, we explored whether YTHDF1 deficiency improves the efficacy of ICI therapy in mouse models. We initially compared immune checkpoint molecules between the WT and KO groups. In tumor cells, YTHDF1 deficiency increased PD-L1 expression in vivo but not in vitro (Fig. 6A). Single-cell sequencing analysis demonstrated that PD1⁺ or CTLA4⁺

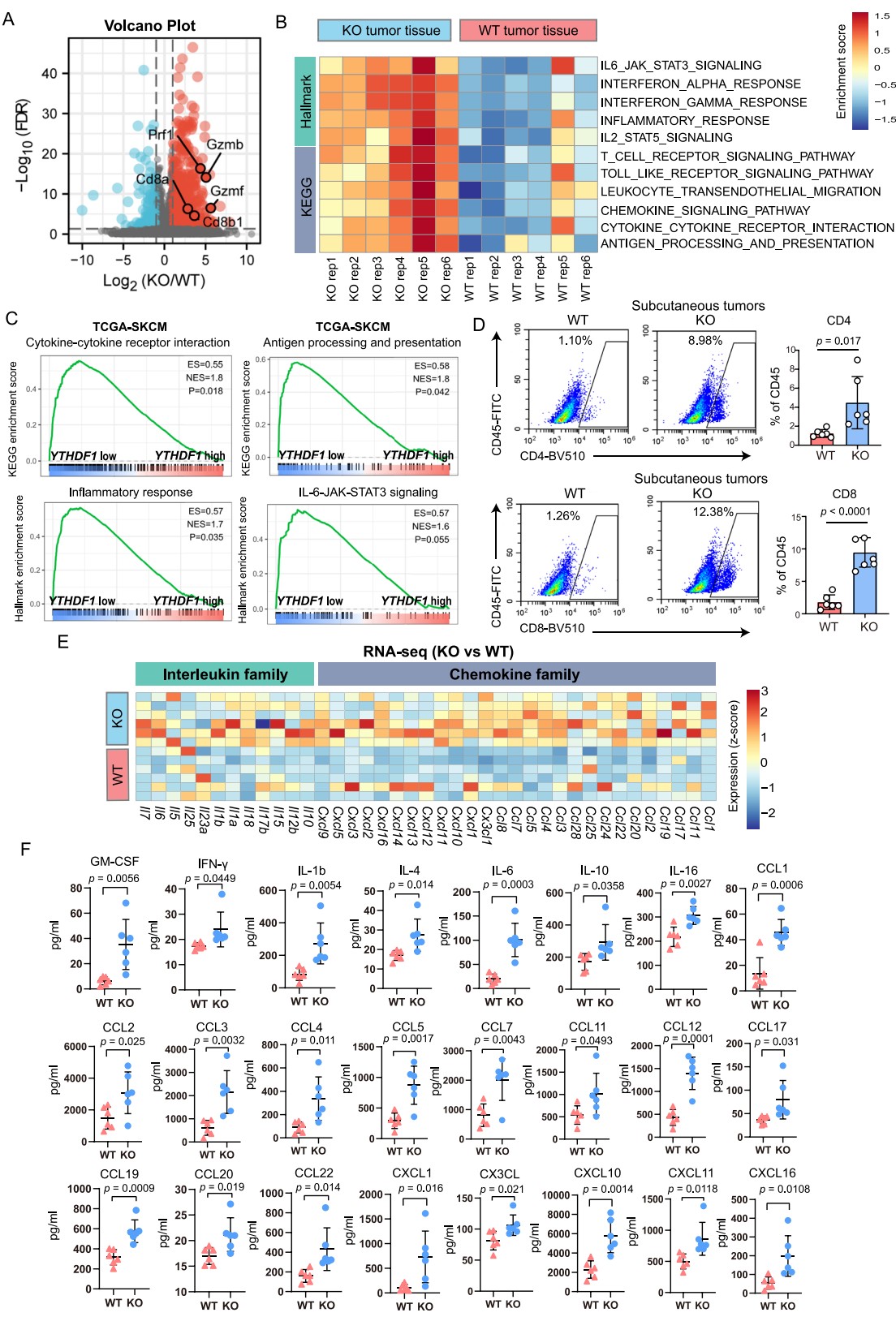

CD8[+] T cells were enriched in the KO group (Fig. 6B). These results indicated that YTHDF1 deficiency might synergize with ICI therapy to elicit robust antitumor effects. Indeed, an anti-PD-L1 or anti-CTLA-4 antibody remarkably reduced tumor volume in mice bearing *Ythdf1*-KO B16/F10 cells compared with an immunoglobulin G (IgG) antibody (Fig. 6C). Of note, 87.5% of the mice treated with the anti-CTLA-4 combination and 75% of the mice treated with the anti-PD-L1 combination exhibited complete tumor rejection and achieved tumor-free survival (Fig. 6D). In addition, YTHDF1 deficiency synergized with ICI to significantly prolong overall survival (Fig. 6E). More excitingly, mice cured by YTHDF1 deficiency and ICI exhibited resistance to WT tumor rechallenge (Fig. 6F), indicating that this combination therapy could elicit effective and long-lasting immune memory that protected the mice from tumor relapse.

**Fig. 3 | Tumor-intrinsic YTHDF1 deficiency triggers antitumor immunity.**
**A** Volcano plot of RNA-seq data for WT or KO tumors inoculated into C57BL/6J mice
($n = 6$ biologically independent samples per group). Differentially expressed genes
were identified with the threshold of |log2(fold change)| > 1 and false discovery rate
(FDR) < 0.05. **B** Heatmap of the differences in pathway activities scored by GSVA.
**C** GSEA revealing immune-related pathways correlated with *YTHDF1* expression in
the TCGA-SKCM dataset ($n = 471$ biologically independent samples). Lower *YTHDF1*
expression was closely related to cytokine–cytokine receptor interaction, antigen
processing and presentation, inflammatory response, and IL-6-JAK-STAT3 signal-
ing. **D** Flow cytometric analysis of tumor-infiltrating CD4$^+$ and CD8$^+$ T cells on day 14
after tumor cell inoculation ($n = 6$ biologically independent samples per group).
Two-tailed unpaired Student's *t*-test. Data are presen*t*ed as mean values ± SD.
**E** Heatmap of RNA-seq data for interleukin and chemokine family member
expression. **F** Validation of cytokine expression by a multicytokine assay ($n = 6$
biologically independent samples per group). Tumor tissue was homogenized, the
cell supernatant of each treatment group was collected, and cytokine levels were
detected using a panel of 31 mouse cytokines. Two-tailed unpaired Student's *t*-test.
Data are presen*t*ed as mean values ± SD. Source data are provided as a Source
Data file.

## Tumor-intrinsic YTHDF1 deficiency promotes immunogenicity by limiting lysosomal proteolysis of MHC-I

Proteins are stable molecules involved in the central performance of various cellular processes. To comprehensively elucidate the mechanism of tumor-intrinsic YTHDF1 deficiency in the induction of antitumor immunity, we initially performed proteomics to reveal the alteration of intrinsic proteins and pathways between B16/F10 WT ($n = 5$) and KO ($n = 5$) groups. In the proteomics analysis, we focus on investigating the altered pathway or a panel of genes. Totally, we identified 979 differentially expressed proteins, with 200 upregulated and 797 downregulated proteins (Fig. 7A). Noteworthy, most of the differentially expressed proteins were downregulated in *Ythdf1*-KO group, which fit well with the critical role of YTHDF1 in promoting translation. Functional enrichment analysis showed that lysosome was the most significantly altered pathway downregulated in the KO group (Fig. 7B). We further performed multi-omic analyses to achieve multiple aims: (i) to identify YTHDF1 m6A targets by m6A methylated RNA immunoprecipitation sequencing (m6A MeRIP-seq) and enhanced cross-linking and immunoprecipitation (eCLIP) sequence, (ii) and to evaluate the effect of YTHDF1 on the translation of its targets by ribosome profiling sequencing (Ribo-seq). m6A methylation of mammalian mRNA occurs mostly at the CDS and 3'UTR. In line with previous evidence, m6A peaks were enriched in the CDS and 3' UTR regions (Supplementary Fig. 9A) and exclusively detected with consensus GGAC motifs (Supplementary Fig. 9B), indicating the successful enrichment of m6A-modified mRNA. Similarly, eCLIP-seq revealed 3233 potential candidate targets of YTHDF1 (Supplementary Fig. 9C). To identify the downstream translation regulated by YTHDF1, we performed Ribo-seq analysis. Ribo-seq analysis showed that most ribosome footprints (RFs) were aligned to an annotated CDS with canonical ribosome-protected fragments (28–29 nt) and a distinctive three-nucleotide periodic footprint pattern (Supplementary Fig. 9D, E). These data enabled global analysis of translation between the WT and *Ythdf1*-KO groups. Next, we integrated these data with proteomics to further reveal the regulator function of YTHDF1. Strikingly, most lysosomal genes were identified as YTHDF1-targeted and m6A-marked transcripts followed by the attenuated translation (Fig. 7C). Given that YTHDF1 is well known to affect mRNA translation in m6A manner, our results indicate that YTHDF1 deficiency impedes lysosomal genes translation, which ultimately reduces protein expression and obstructed the generation of lysosomes. Indeed, flow cytometry and transmission electron microscope further confirmed the attenuated lysosome in the *Ythdf1*-KO group (Fig. 7D, E).

It is known that lysosomal proteolysis promotes immune evasion of tumor cells by degrading tumor antigens and MHC-I[30]. Here, we hypothesized that *Ythdf1* KO limited lysosomal proteolysis in tumor cells, ultimately enhancing the MHC-I expression and minimizing the destruction of internalized antigens. These could increase the immunogenicity of tumor cells and attract the immune armies to restore tumor immune surveillance. To test this hypothesis, we initially analyzed the surface MHC-I expression of tumor cells and found that MHC-I was significantly upregulated in KO group (Fig. 7F). Besides, pharmacological inhibition of lysosomes by bafilomycin A1 (BafA1) or

chloroquine (CQ) in the WT group could mimic MHC-I upregulation of the KO group (Fig. 7F). In contrast, YTHDF1 depletion failed to show any effect on MHC II expression (Supplementary Fig. 9F).

To further test the immunogenicity of WT and KO cells, mice were previously immunized with the whole tumor antigens (WTA) generated from WT and *Ythdf1*-KO groups and were then challenged with B16/F10 cells 15 days after priming. Briefly, WT or KO cells were sonicated to release WTA. Then, 200 µg of each whole cell lysate was mixed with 100 µg poly(I:C) vaccine adjuvant, and sub-cutaneously injected into mice three times, at 7-day intervals. After priming, C57BL/6J mice were challenged with B16/F10 cells and subjected to the downstream in vivo analysis (Fig. 7G). The results showed that priming with WTA from *Ythdf1* KO cells exerts a robust antitumor effect as evidenced by reduced tumor volume, improved survival rate and increased immune cell infiltration (Fig. 7H–L). These indicated that YTHDF1 depletion suppressed tumor antigen and MHC-I degradation, ultimately enhancing tumor recognition and restoring tumor immune surveillance.

In T cell-mediated tumor immune surveillance, recognition of MHC class I antigens on the tumor cells by the T cell receptor of CD8$^+$ cytotoxic T cells is mandatory for the effector T cells to kill tumor cells. In our analysis, single-cell TCR-sequencing determined that CD4 and CD8 TCR clonotype expansion were significantly increased in *Ythdf1*-KO tumors and were mostly distributed in functional CD4$^+$ and CD8$^+$ T cells (Figs. 4J–L and 5H–J). These also supported that T lymphocytes could efficiently recognize tumors in KO group.

## Exosome-mediated YTHDF1 depletion attenuates tumor progression via restoration of antitumor immunity in vivo

The oncogenic function of YTHDF1 is progressively being elucidated in several cancers, whereas the development of efficient inhibitors remains stagnant. Exosome-based delivery systems hold great promise for targeted delivery to specific cell populations to expand therapeutic efficacy. Therefore, we designed a system for exosome-mediated delivery of CRISPR/Cas9 to target oncogenic *Ythdf1* gene in vivo (Fig. 8A). Briefly, natural exosomes were purified from the TME. CRISPR/Cas9 plasmids targeting *Ythdf1* gene (KO exosome) or a control vector (vector exosome) were loaded into exosomes. After exosome quality control, tumor-bearing C57BL/6J mice were intratumorally injected with the engineered exosomes.

For quality control, natural exosomes, KO exosomes and vector exosomes were subjected to transmission electron microscopy (TEM), nanoparticle flow cytometry, and western blotting for characterization. Nanoparticle flow cytometry showed that particle size was mainly distributed at approximately 70 nm, and no difference in size was observed between the natural and engineered exosomes (Fig. 8B). In addition, western blotting revealed the presence of the exosomal markers CD9, TSG101, Flotillin-1, and Alix (Fig. 8C), and transmission electron microscopy confirmed the presence of exosomal structures (Fig. 8D). Moreover, confocal microscopy imaging showed that tumor cells could efficiently take up engineered exosomes (Fig. 8E). In addition, in vivo exosome tracing showed that engineered exosomes were mainly distributed in tumors at 24 h after intratumoral injection, which indicated efficient tumor targeting (Fig. 8F).

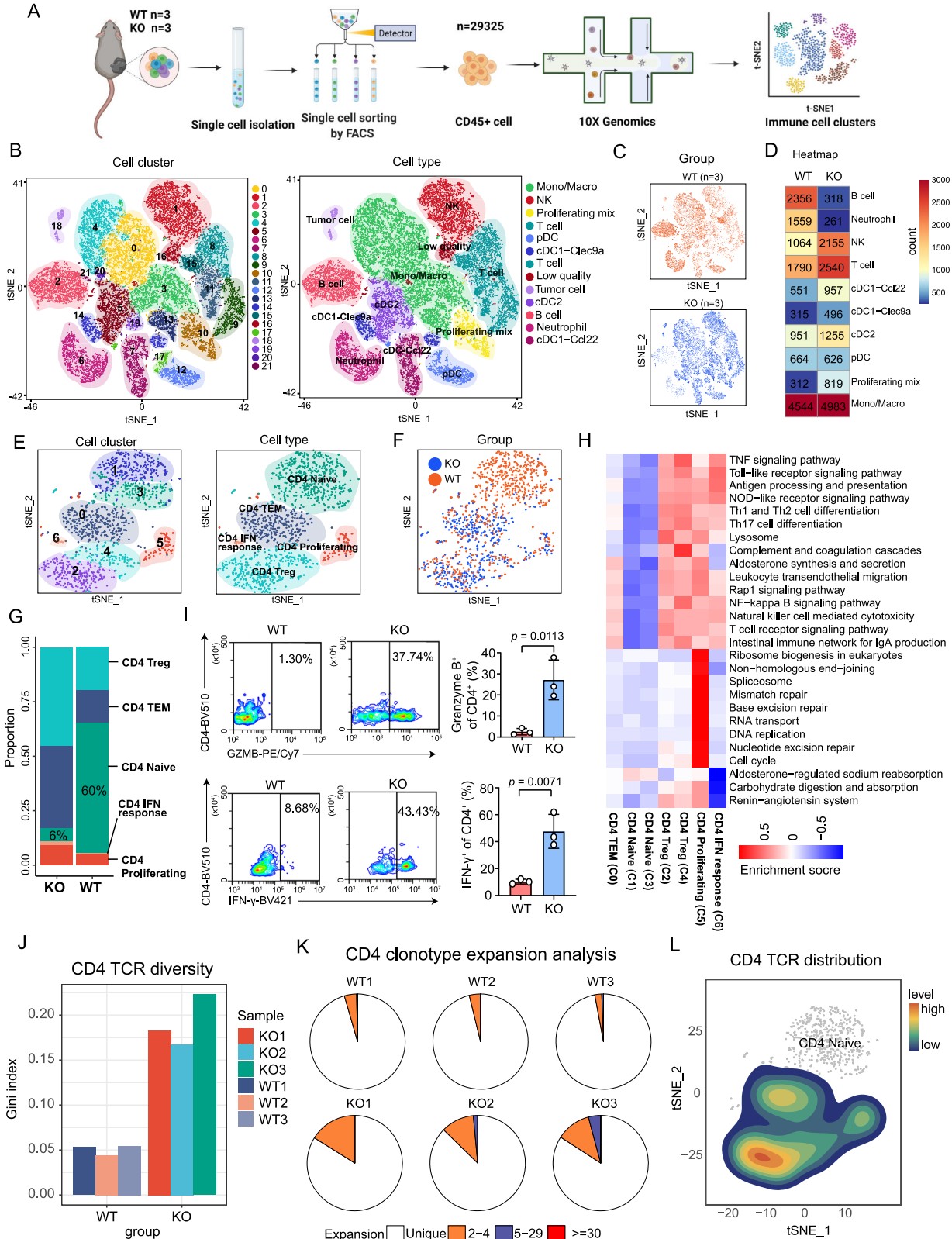

Then, we evaluated the therapeutic efficacy of engineered exosomes in vivo. B16/F10 tumor-bearing mice received an intratumoral injection of KO exosomes or vector exosomes on day 5. The same media without exosomes received the same plasmid DNA transfection procedure that was used as mock control. The growth of B16/F10 tumors was greatly suppressed in the KO exosomes treatment group compared to the vector exosomes group and mock control groups

(Fig. 8G, H). Notably, the expression of YTHDF1 was apparently depleted in the KO exosome group (Fig. 8I). More importantly, elicited immunity was confirmed by identifying increases in tumor-infiltrating CD4$^+$ and CD8$^+$ T cells in the KO exosome group (Fig. 8J).

We further evaluated the safety of engineered exosomes. As shown in Supplementary Fig. 10, there were no differences in mouse body weight (Supplementary Fig. 10A), morphology of lung, liver, and

**Fig. 4 | Single-cell RNA-sequencing (scRNA-seq) analysis revealed a distinct immune landscape between *Ythdf1*-KO and WT tumors. A** Workflow of scRNA-seq experiments. **B**, **C** The t-distributed stochastic neighbor embedding (t-SNE) plot of 29325 cells isolated from melanoma tissues from the KO (*n* = 3 biologically independent samples) and WT groups (*n* = 3 biologically independent samples), with each cell color coded for cell cluster, cell type (**B**), and sample group (**C**). **D** Heatmap showing the numbers of each immune cell subpopulation in the KO and WT groups. **E**, **F** The t-SNE plot of CD4+ T-cell subpopulations, color coded by cell cluster, cell type (**E**), and sample group (**F**). **G** Bar plot depicting the ratio of CD4+ T-cell subpopulations. **H** Heatmap of differentially activated pathways among all the CD4+ T-cell clusters. **I** Flow cytometry-based quantification of IFN-γ+ or gran-zyme B+ CD4+ T cells. Tumor-infiltrating CD4+ T cells were restimulated with PMA and ionomycin for 6 h and were activated by T-Activator CD3/CD28 Dynabeads (*n* = 3 biologically independent samples per group). Two-tailed unpaired Student's *t*-test. Data are presented as mean values ± SD. **J** Bar plot showing TCR diversity of CD4+ T cells. **K** Pie chart of CD4 clonotype expansion. **L** The t-SNE plot of CD4 TCR distribution. Source data are provided as a Source Data file.

kidney (Supplementary Fig. 10B), renal function, liver function, and myo-cardial enzymonram (Supplementary Fig. 10C) between the vector and KO exosome groups.

Together, these results demonstrated that exosome-mediated YTHDF1 depletion was a safe approach to suppress tumor growth by activating the antitumor immune response.

## Discussion

Evading immune destruction is considered an emerging hallmark of cancer, which is a key obstacle to developing successful anticancer therapy options. Due to its significant influence on immunotherapeutic treatment outcomes, the recently discovered function of RNA methylation involved in modulating immune cell infiltration into the TME has piqued curiosity. As depicted in Fig. 9, our findings illustrate that tumor-intrinsic YTHDF1 drives immune evasion via lysosomal degradation of tumor antigen and MHC-I. Starting from the observation that YTHDF1 overexpression is associated with an "immune desert" phenotype and resistance to ICB, we demonstrated that tumor-intrinsic YTHDF1 deficiency inhibited tumorigenesis in immunocompetent mice and that tumor suppression was mainly attributed to the activation of antitumor immunity. Specifically, YTHDF1 deficiency in tumors could alter the immune component of the TME by enhancing the infiltration of CD8+ T cells, and CD4+ T cells; and activating immunity-related signaling pathways. Mechanistically, YTHDF1 was shown to bind to a panel of lysosomal genes and facilitate its translation. Whereas YTHDF1 depletion limited lysosomal proteolysis in tumor cells, subsequently enhancing the MHC-I expression and minimizing the destruction of internalized antigens, which ultimately restored tumor immune surveillance and triggered a robust antitumor immunity. More importantly, to target oncogenic YTHDF1 in vivo, we developed an exosome-induced CRISPR/Cas9 delivery system, which led to YTHDF1 depletion and restored tumor immune surveillance. Our findings uncover the role of tumor-intrinsic YTHDF1 in driving immune evasion and elucidate its underlying mechanism. Targeting YTHDF1 with engineered exosomes is a potential therapeutic strategy to restore antitumor immunity.

One of the well-characterized mechanisms involved in immune evasion is the downregulation or loss of antigen presentation, which confers tumor cells with the ability to become "invisible" and avoid immune attack[31]. CD8+ T lymphocytes can recognize processed tumor antigens as small peptides presented by MHC-I molecules. This recognition and activation end with the destruction of the tumors but leaves unharmed MHC/HLA-I negative tumor cells[32-34]. Clinically, downregulation of tumor MHC-I expression is associated with unfavorable outcomes and resistance to ICIs[35-37]. In our study, YTHDF1 deficiency significantly promoted tumor MHC-I expression and subsequently led to an enhanced T-cell recognition as evidenced by the increased infiltration of CD4+ and CD8+ T cells, and expanded TCR clone type in the TME. Tumors can reduce antigen presentation through several mechanisms including reduced surface expression of MHC-I through genetic alterations, antigen depletion, and degradation. Of notice, recent studies highlight the important role of lysosomal proteolysis in promoting immune evasion of tumor cells by degrading tumor antigen and MHC-I[30,38]. Moreover, pharmacological inhibition of lysosomes could enhance the MHC-I expression and

minimize the destruction of internalized antigens[30]. Here, we demonstrated that tumor-intrinsic YTHDF1 initiates translation of lysosomal genes and promotes proteins expression by increasing the ribosomal loading of m6A-modified mRNA, which is critical for maintaining intact lysosomal function and degrading tumor antigen and MHC-I. Nevertheless, it should be noted that our findings did not analyze the signaling between tumor and TME that may regulate the expression of YTHDF1, which require further exploration in vitro or in vivo. Our findings should be interpreted with this limitation in mind.

CRISPR/Cas9 is an efficient gene editing technology with great promise. At present, immunogenicity, limited packing capability, and poor tolerability have been identified as the challenges that impede the intracellular distribution of CRISPR/Cas9 systems[39]. Exosomes (30–200 nm in diameter) are nanoscale membrane-bound vesicles that are spontaneously produced from the endocytic compartments of all living cells and transport molecular cargo, including lipids, proteins, RNA, and DNA[40]. Exosomes, unlike manufactured nanoparticle transporters, are usually nonimmunogenic and non-cytotoxic when extracted from a suitable cellular source[41]. Exosomes, as opposed to liposomes, have a variety of membrane-bound and transmembrane-anchored proteins that facilitate their evasion of phagocytic clearance while providing improved cellular absorption and subsequent distribution of their internal payload to target cells, extending their half-life in the circulation[42,43]. Therefore, exosomes have been exploited as carriers for efficient drug delivery. Previous studies have reported the great promise of engineered exosomes in delivering CRISPR/Cas9 to target oncogenes in cancers[44,45]. There are three options for delivery CRISPR system, including plasmid DNA encoding Cas9 and sgRNA, Cas9 mRNA plus sgRNA, and Cas9 ribonucleoprotein (RNP) complexed with sgRNA. Delivery of CRISPR system by plasmid DNA presents a convenient approach and has been applied by several studies[45,46]. Noteworthy, the introduction of CRISPR components as plasmids is also associated with potential off-target effects. Compared to DNA and RNA delivery, RNP delivery avoids many pitfalls allowing for fast delivery and weak off-target effects[47,48]. However, the therapeutic delivery of RNPs is currently hampered by the large size of RNPs and lacks resistant gene for drug selection. In our study, we initially created engineered exosomes that transported CRISPR/Cas9 plasmid DNA for in vivo targeting of carcinogenic *Ythdf1*. This presents a safe approach to suppressing tumor growth by restoring tumor immune surveillance. Despite this, it should be noted that further optimization of the CRISPR system delivery is needed in our further study.

## Methods

### Animal models

Specific pathogen-free female C57BL/6J mice (6–8 weeks old) and female NOD-*Prkdc*scid-*Il2rg*null mice (6–8 weeks old) were purchased from SPF (Beijing) Biotechnology Co., Ltd. All the mice were kept in an environment that was free of pathogens (at 22–25 °C, relative humidity of 45–60%, a 12 h light/dark cycle). All the mice were utilized in compliance with the Shanghai Proton and Heavy Ion Center Institutional Animal Care and Use Committee guidelines (protocol number: Q2021011). The maximal tumor size/burden of 2000 mm³ was permitted by the ethics committee. In some cases, this limit has been

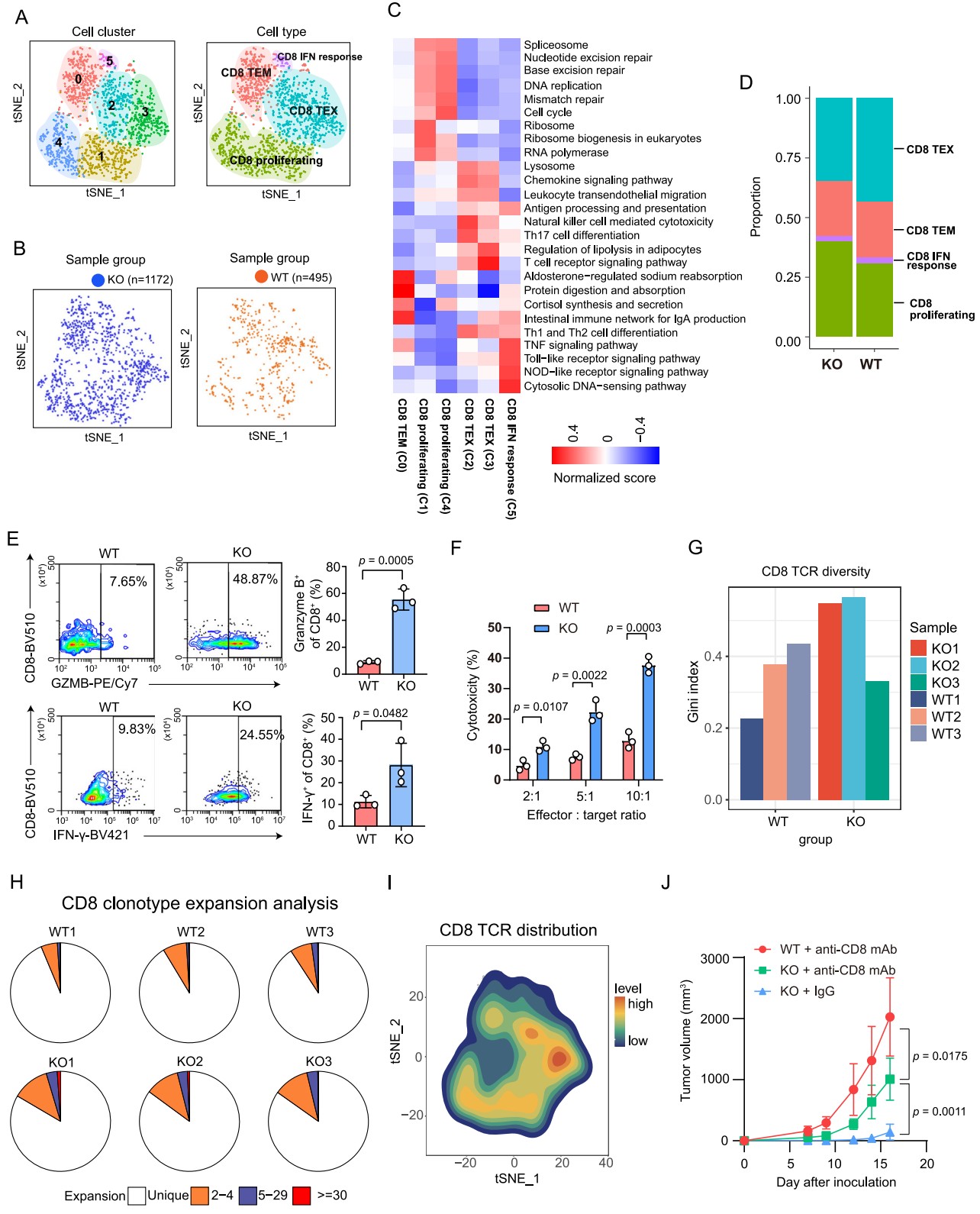

exceeded by the last day of measurement and the mice were immediately euthanized.

## Cell lines

Murine melanoma B16/F10 cells were acquired from the Cell Bank of the Shanghai Academy of Chinese Sciences and were cultured in Dulbecco's modified Eagle's medium (Gibco, catalog no. 10566016) that contained 10% fetal bovine serum (FBS) (Gibco,

catalog no. 10099141) and 1% penicillin–streptomycin (Invitrogen, catalog no. 15140155).

## *Ythdf1* knockout by the CRISPR/Cas9 system

CRISPR/Cas9-induced depletion of the *Ythdf1* gene was performed using a transposon plasmid containing expression cassettes for spCas9, a puromycin resistance gene, and chimeric guide RNA. 5′-AGCAGCCACTTCAACCCCGC TGG-3′ and 5′-GGACCATGGTGCC

**Fig. 5 | scRNA-seq analysis of CD8[+] T-cell subpopulations. A, B** The t-SNE plot of CD8[+] T-cell subpopulations, color coded by cell cluster, cell type (**A**), and sample group (**B**). **C** Heatmap of differentially activated pathways among all the CD8[+] T-cell clusters. **D** Bar plot showing the percentage of each CD8[+] immune cell population in the KO and WT groups. **E** Flow cytometry-based quantification of IFN-γ[+] or granzyme B[+] CD8[+] T cells. Tumor-infiltrating CD8[+] T cells were restimulated with PMA and ionomycin for 6 h ($n = 3$ biologically independent samples per group). Two-tailed unpaired Student's $t$-test. Data are presented as mean values ± SD. **F** In vitro cytotoxicity assays performed with activated tumor-infiltrating CD8[+] T cells and

B16/F10 target cells ($n = 3$ biologically independent samples per group). Two-tailed unpaired Student's $t$-test. **G** Bar plot showing TCR diversity of CD8[+] T cells. **H** Pie chart of CD8[+] T cell clonotype expansion. **I** The t-SNE plot of CD8 TCR distribution. **J** Tumor growth curves of WT or KO tumors in C57BL/6J mice treated previously with an IgG control or anti-CD8 antibody ($n = 6$ biologically independent animals per group). Two-way ANOVA with Tukey's multiple comparison test. Data are presented as mean values ± SD. The results are representative of two independent experiments. Source data are provided as a Source Data file.

TCGCTGA GGG-3' were created as guide RNA sequences to target exon 4 of the *Ythdf1* gene. The plasmid carrying the guide RNA sequence was electrotransfected into cells utilizing the Neon transfection system (Thermo Fisher Scientific) in accordance with the guidelines stipulated by the manufacturer. Puromycin (2 μg/mL) was utilized to select cells 24 h after transfection. Single colonies were transferred into 96-well plates 3 days later. Genomic DNA was extracted with a Quick-DNA Miniprep kit (Zymo Research) to detect the existence of deletions or insertions in *Ythdf1* target colonies, and PCR amplification was performed utilizing 2× Taq Master Mix (Dye Plus; Vazyme, catalog no. P112) and primers flanking the exon (forward: 5′-GGCAGAA GGGTGGTTTGACTG-3′; reverse: 5′-GAGCGGTGGATGTCGTCCTC-3′). Sanger sequencing (GENEWIZ, China) was employed to sequence plasmids obtained from 5-10 colony-forming units, and the sequence was analyzed by Chromas software. Positive clones were selected for downstream studies. All of the clones were kept in the circumstances identical to those of the parent cells.

### Tumor challenge and treatment

After resuspending B16/F10 cells ($5 \times 10^5$) in 0.2 mL phosphate-buffer saline (PBS), the cells were subcutaneously injected into the flank of C57BL/6J mice. Tumor volume was determined as follows: long diameter × short diameter$^2$/2, and the tumor volume was plotted against time to create a tumor growth curve. A tumor volume exceeding 1500 mm$^3$ was considered an event in the survival experiments.

For in vivo tumor imaging, briefly, tumor cells were previously labeled with luciferase. Then, mice bearing tumors were anesthetized by isoflurane inhalation, administered 15 mg/kg D-luciferin sodium salt (Selleck Chemicals, catalog no. 103404-75-7) by intraperitoneal injection, and imaged 10 min afterward using an In-Vivo Xtreme imaging system (Bruker MI SE).

For ICI therapy, mice bearing WT or *Ythdf1*-KO B16/F10 cells were intraperitoneally administered anti-mouse CTLA4 mAbs (200 μg per mouse; BioXCell, catalog no. BE0164, clone: 9D9), anti-PD-L1 mAbs (200 μg per mouse; BioXCell, catalog no. BE0101, clone: 10F.9G2) or appropriate isotype control mAbs on days 7, 9 and 11. With regard to in vivo depletion of CD8[+] T cells, anti-mouse CD8 mAbs (200 μg per mouse; BioXCell, catalog no. BE0117, clone: YTS169.4) were intraperitoneally administered on days −6, −3, and −1 before tumor challenge, and the same dose was given on days 7, 9 and 11 after tumor challenge.

### Exosome isolation

Exosomes derived from B16/F10 tumor tissue were isolated via ultracentrifugation. Briefly, 0.1 mg/mL DNase I (Roche, catalog no. 10104159001) and 1 mg/mL Collagenase D (Roche, catalog no. 11088858001) were used to digest tumor cells at 37 °C for 30 min. EDTA was utilized to stop sample digestion, and the supernatants were centrifuged for 5 min at 500 × g to remove any intact cells. The supernatants were then separated by centrifugation at 2000 × g for 10 min after being transferred to fresh polycarbonate tubes. To remove shed microvesicles (200–1000 nm), the supernatants were filtered, funneled into fresh polycarbonate tubes and centrifuged for 30 min at 10,000 × g. Subsequently, the supernatants were collected,

filtered through 0.22-μm membrane filters (Merck Millipore) and centrifuged at 100,000 × g for 2 h. Finally, the exosomes were resuspended in 1× PBS for subsequent use.

### Characterization of exosomes

Exosome size was evaluated utilizing nanoparticle flow cytometry (NanoFCM; SNA-D1, UK), and NF Professional 1.0 software was used to explore their structural properties. Transmission electron microscopy (Tecnai 12, Philips) was employed to examine exosomal structure. The exosome-specific surface markers TSG101 (1:2000; Abcam, catalog no. ab125011), Flotillin-1 (1:2000; Abcam, catalog no. ab133497), Alix (1:2000; Abcam, catalog no. ab275377), and CD9 (1:1000; Cell Signaling Technology, catalog no. 98327) were evaluated utilizing western blot analysis.

### Exosome loading using plasmid DNA and in vivo exosome treatments

The Exo-Fect Exosome Transfection Kit (System Biosciences, catalog no. EXFT20A-1) was employed to load cargo into exosomes in accordance with the recommendations of the manufacturer. Briefly, 5 μg plasmid DNA (*Ythdf1* sgRNA-spCas9 plasmid or control sgRNA-spCas9 plasmid), 10 μL Exo-Fect solution, 200 μg exosomes, and sterile PBS were mixed in a 150 μL total transfection solution and subjected to 10 min of incubation at 37 °C in a shaker. Thirty microliters of ExoQuick-TC reagent were introduced into the transfected exosomal sample suspension with gentle shaking and incubated at 4 °C for 30 min to terminate the reaction. The sample was further centrifuged in a microfuge for 3 min at 10,000 × g. Next, the transfected exosomes were treated with DNase I (0.15 units/μL, Sigma-Aldrich) to exclude the residual plasmid DNA, and collected by ultracentrifugation at 100,000 × g for 70 min. The media without exosomes received the same plasmid DNA transfection and DNase I treatment procedure was used as mock control. For in vivo exosome treatments, mice received intratumoral injections of 200 μg exosome proteins (vector exosome or KO exosome) in 30 μL PBS. The exosome protein quantitation was determined by bicinchoninic acid (BCA) assay.

### Exosome labeling and in vivo live imaging

Briefly, exosomes were labeled with PKH67 (10 μM; Sigma, catalog no. MINI67) following the manufacturer's procedures. After staining, the exosomes were ultracentrifuged to remove the unbound dye and were washed with PBS to remove the residual dye further. The cellular uptake and intracellular distribution of exosomes were determined with a Zeiss LSM510 confocal microscope (ZEISS). The images were processed by ZEN software (blue edition). The cell contours were imaged by staining with phalloidin (1:1000; Abcam, ab176759). The media that was not exposed to the tumor and received the same exosome isolation and fluorescent labeling procedure was used as mock control. To trace the biodistribution of exosomes in vivo, mice received intratumoral injections of 200 μg exosome proteins in 30 μL PBS, and the fluorescence signal was acquired at 24 h post-injection using an In-Vivo Xtreme imaging system (Bruker).

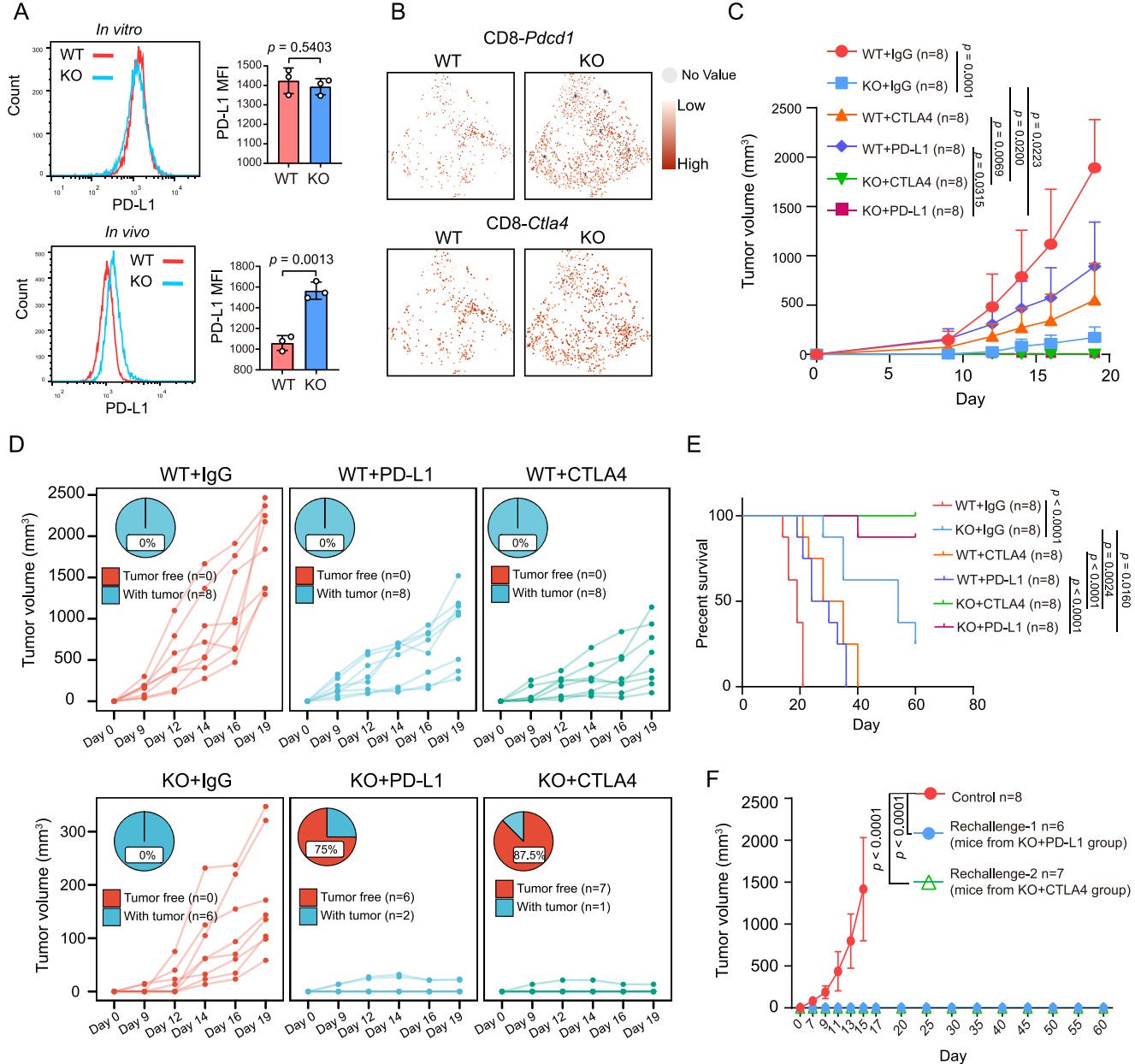

**Fig. 6 | Tumor-intrinsic YTHDF1 deficiency enhances ICI therapy responses in vivo. A** The protein expression of surface PD-L1 was analyzed by flow cytometry and is shown as the mean fluorescence intensity (MFI). $n = 3$ biologically independent samples per group. Two-tailed unpaired Student's $t$-test. Data are presented as mean values ± SD. **B** The t-SNE plot of *Pdcd1* and *Ctla4* in CD8+ T cells. **C** Tumor growth curves of WT and KO tumors. Mice were intraperitoneally treated with 200 μg anti-PD-L1 or 200 μg anti-CTLA4 on days 7, 9 and 11 after tumor inoculation. A rat immunoglobulin G (IgG) isotype antibody was applied as a control. Two-way ANOVA with Tukey's multiple comparison test. Data are presented as mean values ± SD. The statistical comparisons among these groups were listed as follows: WT + IgG vs KO + IgG, $p = 0.0001$; WT + IgG vs WT + PD-L1, $p = 0.0084$; WT + IgG vs WT + CTLA4, $p = 0.0004$; WT + IgG vs KO + PD-L1,

$p = 0.0001$; WT + IgG vs KO + CTLA4, $p = 0.0001$; KO + IgG vs KO + PD-L1, $p = 0.0223$; KO + IgG vs KO + CTLA4, $p = 0.020$; WT + CTLA4 vs KO + CTLA4, $p = 0.0315$; WT + PD-L1 vs KO + PD-L1, $p = 0.0069$. **D** Individual tumor growth curves and percentages of mice with complete tumor rejection. **E** Kaplan–Meier curves for WT and KO tumor-bearing C57BL/6J mice treated with an IgG isotype, anti-PD-L1 (200 μg) or anti-CTLA4 antibody (200 μg). Tumor volumes exceeding 1500 mm³ were considered events ($n = 8$ biologically independent animals per group). Two-tailed log-rank (Mantel–Cox) test. **F** Tumor growth curves of mice rechallenged with WT B16/F10. The results are representative of two independent experiments. Two-way ANOVA with Tukey's multiple comparison test. Data are presented as mean values ± SD. Source data are provided as a Source Data file.

### Multicytokine assay
Briefly, tumor tissue was homogenized, and the cell supernatant of each treatment group was collected for a multicytokine assay. Cytokine levels were evaluated with a Luminex 200 system (Luminex) using a panel of 31 mouse cytokines (LX-MultiDTM-31) in accordance with the recommendations of the manufacturer. The selected cytokines included TNF-alpha, IFN-γ, GM-CSF, IL-10, IL-16, IL-1b, IL-2, IL-4, IL-6, CX3CL, CXCL1, CXCL10, CXCL11, CXCL12, CXCL13, CXCL16, CXCL5,

CCL11, CCL12, CCL17, CCL19, CCL2, CCL20, CCL22, CCL24, CCL27, CCL3, CCL4, CCL5, CCL7, and CCL1.

### Flow cytometric (FCM) analysis of tumor-infiltrating CD4+ and CD8+ T cells
Some mice were euthanized at a later stage of the experiment, and the harvested tumor samples were digested for 30 min at 37 °C with 0.1 mg/mL DNase I and 1 mg/mL Collagenase D (Roche).

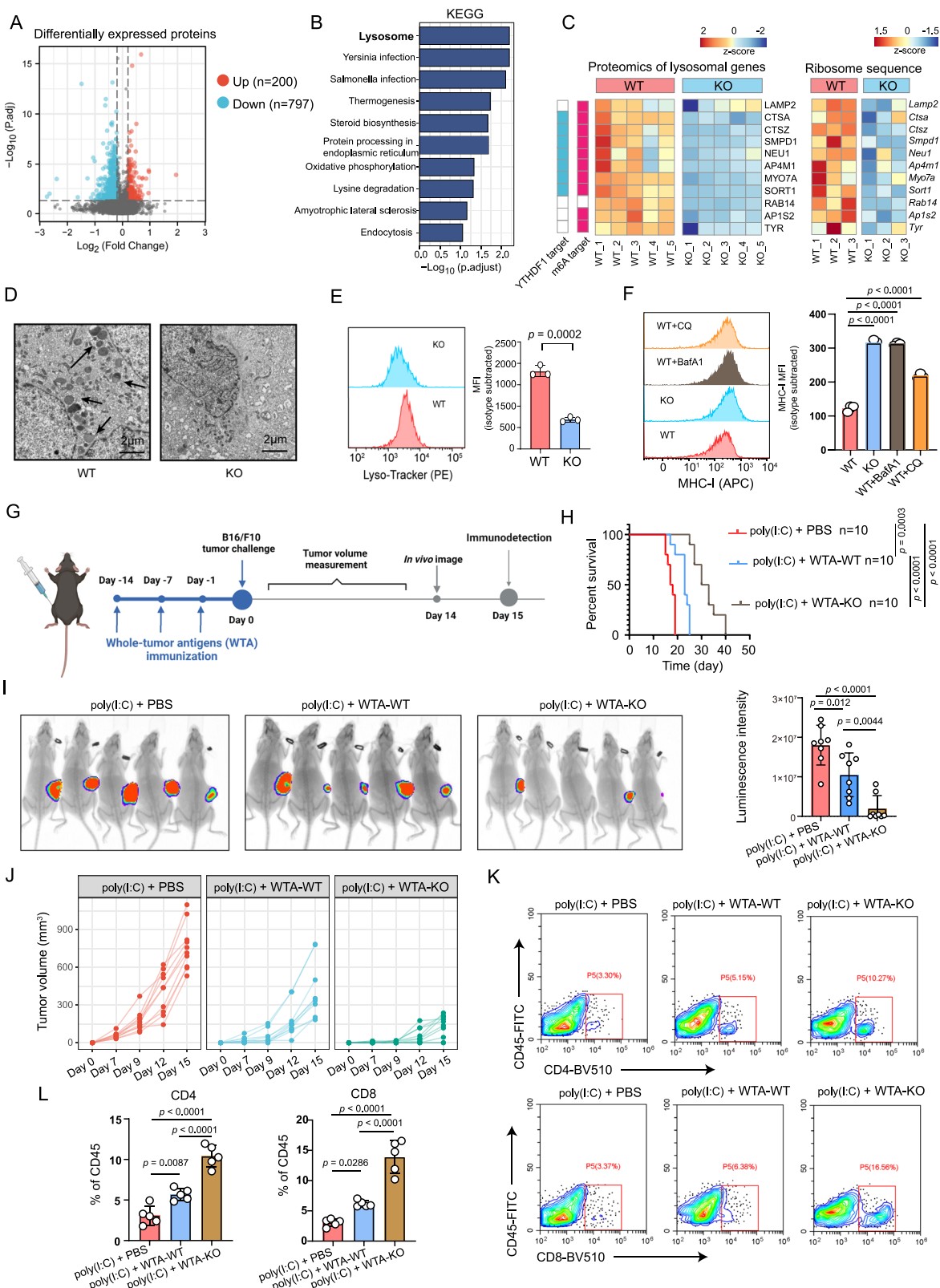

**Nature Communications** | (2023)14:265

After extracting the single-cell suspensions by filtration through a 70-μm cell strainer (MACS SmartStrainers, Miltenyi Biotec, catalog no. 130-098-462), the suspensions were resuspended and pre-incubated with Fc-block (1:200, BD Pharmingen, catalog no. 553142) before staining to block nonspecific Fc-receptor–mediated binding. Cell surface marker staining was conducted with fluorescein isothiocyanate-conjugated anti-mouse CD45 (1:200, BD Pharmingen, catalog no. 553079), BV510-conjugated anti-mouse CD4 (1:200, BD Pharmingen, catalog no. 563106), or BV510-conjugated anti-mouse CD8 (1:200, BD Pharmingen, catalog no. 563068), and fixable viability dye (1:1000, BD Pharmingen, catalog no. 565388) staining was performed on ice in the dark for 40 min. Tumor-infiltrating CD8$^+$ and CD4$^+$ T cells were further evaluated utilizing a flow cytometer (CytoFLEX S; Beckman Coulter) and the

**Fig. 7 | Tumor-intrinsic YTHDF1 promotes lysosomal proteolysis of MHC-I.**
**A** Volcano plot of proteomics data for WT ($n = 5$ biologically independent samples) and *Ythdf1*-KO ($n = 5$ biologically independent samples) B16/F10 cells. Differentially expressed proteins were identified with the threshold of |fold change| > 1.2 and $p$ value < 0.05. **B** KEGG enrichment of proteins downregulated in *Ythdf1*-KO group. **C** Heatmaps of the lysosomal genes' protein level and translation level. **D** Transmission electron microscopy of WT and *Ythdf1*-KO B10/F10 cells. The arrows indicate lysosome. One of three representative experiments with similar results is shown. **E** The lysosome level was analyzed by flow cytometry staining of lyso-tracker (PE) and is shown as the mean fluorescence intensity (MFI). $n = 3$ biologically independent samples per group. Two-tailed unpaired Student's $t$-test. Data are presented as mean values ± SD. **F** Quantitative estimates of cell surface MHC-I levels in WT and *Ythdf1*-KO B16F10 cells and effect of BafA1 (40 nM) or CQ (20 µM) treatment on cell surface MHC-I levels of B16/F10. $n = 3$ biologically independent samples per group. One-way ANOVA with Dunnett's multiple comparisons test.

Data are presented as mean values ± SD. **G** Experimental schedule of whole-tumor antigens (WTA) immunization for B16/F10 tumor model. WT or KO cells were sonicated to release whole tumor antigens. Mice were immunized with 200 µg whole tumor antigen generated from WT (WTA-WT) or KO (WTA-KO) group three times at 7-day intervals. 100 µg poly(I:C) was used as a vaccine adjuvant to enhance antigen presentation. **H** Kaplan–Meier survival curves ($n = 10$ biologically independent animals per group). Two-tailed log-rank (Mantel–Cox) test. **I** In vivo bioluminescence image of B16/F10 tumors ($n = 8$ biologically independent animals per group). One-way ANOVA with Tukey's multiple comparison test. Data are presented as mean values ± SD. **J** Tumor growth curves ($n = 10$ biologically independent animals per group). **K**, **L** Flow cytometric analysis of tumor-infiltrating CD4[+] and CD8[+] T cells ($n = 5$ biologically independent samples per group). One-way ANOVA with Tukey's multiple comparison test. Data are presented as mean values ± SD. One of two representative in vivo experiments with similar results is shown. Source data are provided as a Source Data file.

data were analyzed by CytExpert software. Gating strategy was provided in Supplementary Fig. 11.

### Restimulation of tumor-infiltrating T cells in vitro
T cells were restimulated with a Leukocyte Activation Cocktail Kit (BD Pharmingen, catalog no. 550583) containing the protein transport inhibitor brefeldin A, a calcium ionophore (ionomycin), and phorbol 12-myristate 13-acetate (PMA). After 6 h of incubation at 37 °C, the cells were stained for 30 min with anti-CD4 or anti-CD8 on ice and subsequently subjected to intracellular staining using a Fixation/Permeabilization kit (BD Pharmingen, catalog no. 554714). Specifically, cell fixation and permeabilization were performed by adding fixation/permeabilization solution on ice in the dark for 20 min, and the cells were subsequently washed twice using BD Perm/Wash buffer. After adding antibodies against IFN-γ (1:200, BD Pharmingen, catalog no. 563376) and Granzyme B (1:200, BioLegend, catalog no. 372214), the cells were subjected to 1 h of incubation on ice. Flow cytometry was used to identify cells producing cytokines.

### T-cell killing assay
CD8[+] T cells were acquired from tumor tissues from the WT or *Ythdf1*-KO group using magnetic bead sorting (Miltenyi Biotec, catalog no. 130-116-478) and were cultured in a medium containing mouse T-Activator CD3/CD28 Dynabeads (bead-to-cell ratio = 1:1; Thermo Fisher Scientific, catalog no. 11452D), recombinant mouse IL-2 (20 ng/mL; R&D Systems, catalog no. 402-ML-100), 10% FBS, 10 mM HEPES (Gibco, catalog no. 15630080), and 100 µM NEAA (Gibco, catalog no. 11140076) for 48 h. Then, the activated T cells were cocultured with WT or KO cells at a ratio of 2.5:1, 5:1, or 10:1. After 12 h of coculture, Trypan Blue staining (Beyotime Biotechnology, catalog no. C0011) was utilized to quantify apoptotic tumor cells.

### EdU incorporation assay
The BeyoClick EdU Cell Proliferation Kit (Beyotime Biotechnology, catalog no. C0078S) was utilized to execute an EdU incorporation experiment in accordance with the guidelines stipulated by the manufacturer. Briefly, cells were grown in 10 µM EdU for 4 h at 37 °C and a $CO_2$ concentration of 5%. Then, cell fixation and permeabilization were performed. The cells were subjected to 30 min of incubation in Click Additive Solution at ambient temperature after being rinsed three times with PBS, and fluorescence imaging was performed with a confocal laser scanning microscope.

### Immunohistochemistry (IHC) analysis
IHC-related experimental techniques have been reported previously. Here, IHC was performed with anti-mouse CD8 (1:1000; Abcam, catalog no. ab217344) staining to examine tumor-infiltrating T cells and with anti-mouse CD4 (1:500; Abcam, catalog no. ab183685) staining

followed by incubation with a horseradish peroxidase (HRP)-conjugated secondary antibody (1:10,000, Abcam, catalog no. ab205718) and DAB (1:50, Abcam, ab64238). Images were acquired utilizing an Olympus BX43 microscope (Olympus). The images were processed by Andor SOLIS software.

### Colony formation assays
In total, 1000 cells were inoculated in 6-well plates for colony formation experiments. Colonies were fixed with 4% paraformaldehyde after 14 days, followed by staining using a crystal violet solution (Beyotime Biotechnology, catalog no. C0121). Colony counts and images were acquired using a colony counting machine (GelCount; Oxford Optronix Ltd.).

### In vitro cell growth assay
CCK8 tests were utilized to measure in vitro cell proliferation. The cells in 96-well plates were subjected to incubation for an additional 1 h at 37 °C after being treated with a CCK8 solution (Beyotime Biotechnology, catalog no. C0037). Afterward, absorbance values at a wavelength of 450 nm were recorded, and cell viability was estimated.

### Western blot analysis
Briefly, B16/F10 cells or tumor tissues were lysed with RIPA buffer (Beyotime Biotechnology, catalog no. P0013B) containing protease inhibitor cocktail (Thermo Fisher Scientific, catalog no. 78425). Nuclear proteins were obtained using a Nuclear and Cytoplasmic Protein Extraction Kit (Beyotime Biotechnology, catalog no. P0027). The protein concentration was evaluated by the BCA Assay Kit (Beyotime Biotechnology, catalog no. P0010S), and 30 µg of total protein were electrophoresed via SDS-PAGE and blotted onto a polyvinylidene fluoride (PDVF, Beyotime Biotechnology, catalog no. FFP33) membrane. After blocking, the PVDF membranes were incubated with various primary antibodies including anti-YTHDF1 (1:2000; Abcam, catalog no. ab220162), anti-YTHDF2 (1:1000; Abcam, catalog no. ab220163), and anti-YTHDF3 (1:1000; Abcam, catalog no. ab220161) at 4 °C overnight. On the next day, the PVDF membrane was incubated with the second antibody (1:5000) for 2 h and washed three times with TBST. The bands were then imaged using a Bio-Rad system following the instructions. The uncropped blots for the main and supplementary figures are provided in the Source Data file.

### Immunofluorescence
For tissue immunofluorescence staining, a melanoma tissue microarray procured from Shanghai Outdo Biotechnology was utilized to evaluate the connection between the expression level of YTHDF1 and the infiltration status of CD8[+] T cells. Multiplex immunofluorescence experiments were carried out in the following ways utilizing consecutive staining cycles: once slides had been deparaffinized, rehydrated, and subjected to antigen retrieval,

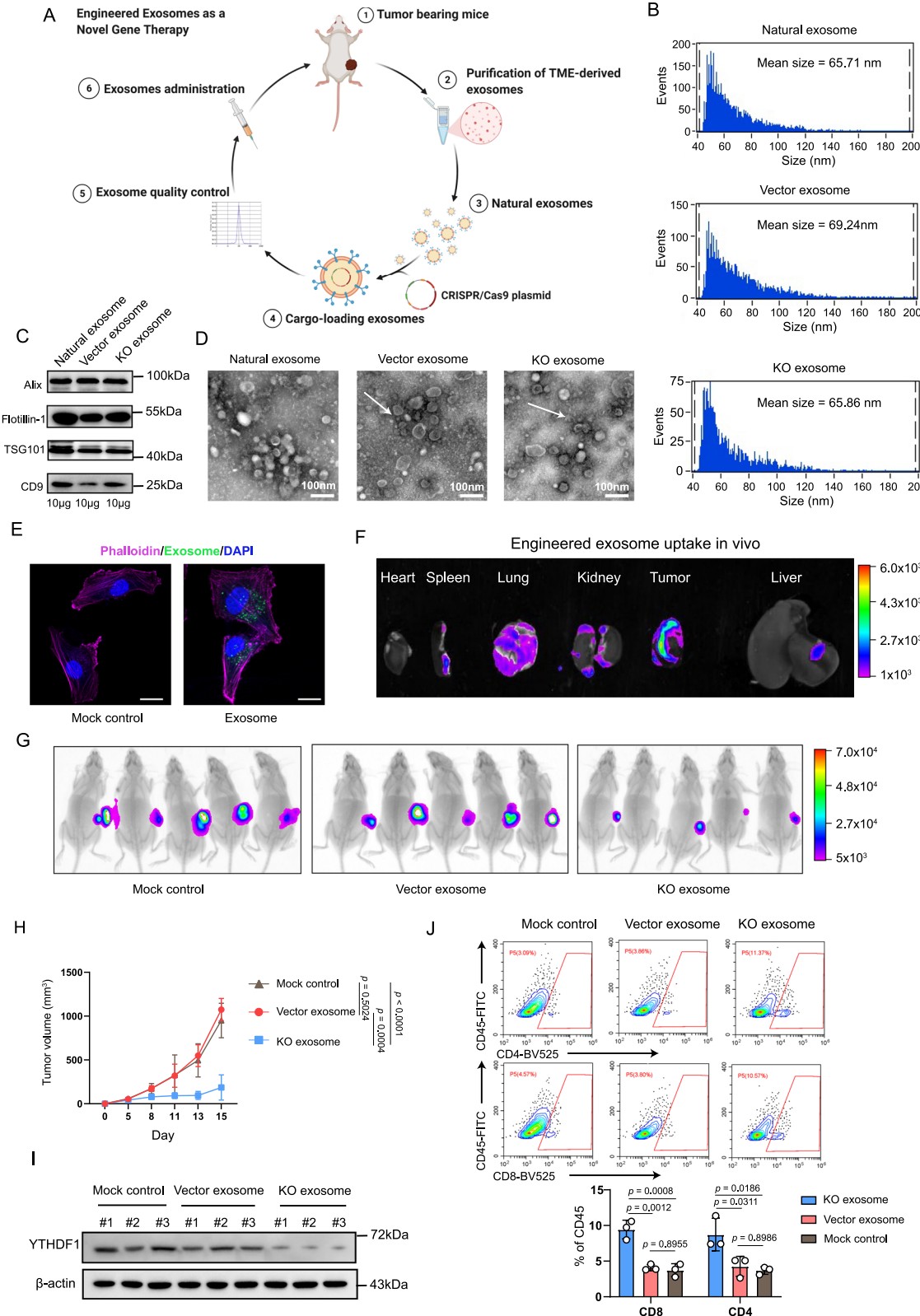

they were subjected to serial staining with antibodies specific for YTHDF1 (1:50; Abcam, catalog no. ab290734) and CD8 (1:500; cell signaling, catalog no. 98941). Afterward, each sample was treated with anti-rabbit polymeric HRP and labeled utilizing Opal Fluorophore Reagents. In a serial fashion, each antigen was labeled with distinct fluorophores.

## Analysis of cell surface MHC-I expression

B16/F10 cell surface MHC-I expression was determined by flow cytometry. Briefly, B16/F10 cells were stained with APC-labeled anti-H-2Kb antibodies (1:100; Thermo Fisher, catalog no. 17595882) for 30 min on ice. The expression of MHC-I was shown as the mean fluorescence intensity. For lysosome inhibition, mouse B16F10 cells were treated

**Fig. 8 | Exosome-mediated YTHDF1 depletion attenuates tumor progression via restoration of antitumor immunity in vivo. A** Workflow of engineered exosome production and administration. Briefly, we purified natural exosomes from the tumor microenvironment (TME). Natural exosomes were loaded with CRISPR/Cas9 plasmids targeting *Ythdf1* gene (KO exosome) or a control vector (vector exosome). **B** Particle size distribution of exosomes measured via nanoparticle flow cytometry. **C** Immunoblotting of exosomal markers. One of three representative experiments with similar results is shown. **D** Transmission electron microscopy revealing the morphology and size of exosomes. Scale bar, 100 nm. One of two representative experiments with similar results is shown. **E** Exosomes uptake assay with detection by confocal microscopy analyses of B16/F10 cells. Exosomes were fluorescently labeled using PKH67 (green) and incubated with B16/F10 cells for 24 h. The cell contours were imaged by staining with phalloidin (purple). The media that was not exposed to the tumor and received the same exosomes isolation and fluorescent labeling procedure was used as mock control. Scale bar, 20 μm. One of three

representative experiments with similar results is shown. **F** In vivo exosomes tracing at 24 h after intratumoral injection of 200 μg fluorescently labeled KO exosomes. One of three representative experiments with similar results is shown. **G** Luminescence intensity of tumors in C57BL/6J mice. B16/F10 tumor-bearing mice were intratumorally administered 200 μg KO exosomes or vector exosomes on day 7 (*n* = 5 biologically independent animals per group). **H** Tumor growth curve of C57BL/6J mice (*n* = 5 biologically independent animals per group). Two-way ANOVA with Tukey's multiple comparison test. Data are presented as mean values ± SD. **I** Immunoblotting of YTHDF1 and β-actin. **J** Flow cytometry-based quantification of tumor-infiltrating CD4[+] and CD8[+] T cells in the KO exosome, vector exosome and mock control groups (*n* = 3 biologically independent samples per group). One-way ANOVA with Tukey's multiple comparison test. Data are presented as mean values ± SD. The results are representative of two independent experiments. Source data are provided as a Source Data file.

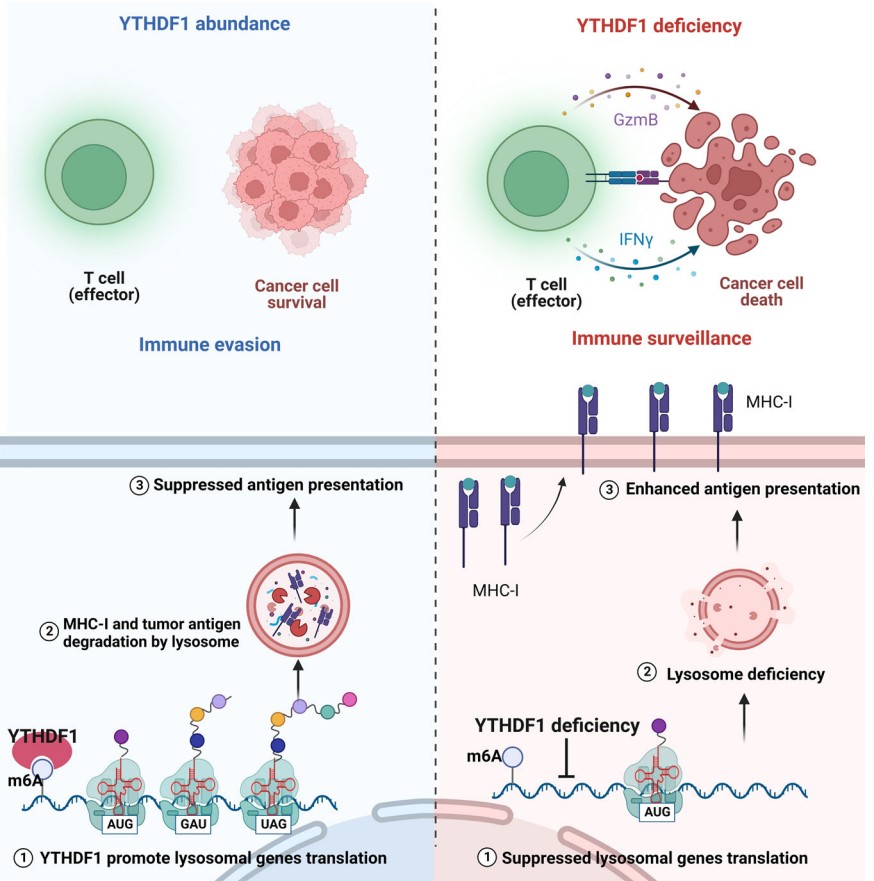

**Fig. 9 | Diagram illustrating the mechanism by which tumor-intrinsic YTHDF1 drives immune evasion.** Briefly, YTHDF1 binds to a panel of lysosomal genes and facilitates its translation. On the contrary, YTHDF1 depletion limited lysosomal

proteolysis in tumor cells, subsequently enhancing the MHC-I expression and minimizing the destruction of internalized antigens, which ultimately improve tumor immune surveillance and triggered a robust antitumor immunity.

with BafA1 (40 nM; MedChemExpress, catalog no. HY-100558) or CQ (20 μM; MedChemExpress, catalog no. HY-17589A) and followed by flow cytometry analysis of MHC-I.

### Whole tumor antigen immunization
WTA were generated by sonicating WT or *Ythdf1*-KO cells. Then, 200 μg of each whole cell lysate was dissolved in 0.1 mL PBS, mixed with 100 μg poly(I:C) (InvivoGen, catalog no. vac-pic) adjuvant, and subcutaneously injected into mice three times, at 7-day intervals. After priming, C57BL/6J mice were challenged with B16/F10 cells and subjected to the downstream in vivo analysis.

### RNA isolation, library construction, sequencing, and analysis
In accordance with the procedure stipulated by the manufacturer, total RNA was isolated with a TRIzol reagent kit (Invitrogen, catalog no. 15596018). RNA quality was ascertained utilizing an Agilent 2100 Bioanalyzer (Agilent Technologies) and confirmed with RNase-free agarose gel electrophoresis (Agilent Technologies). mRNA isolated from eukaryotic cells was enriched utilizing Oligo(dT) beads after total RNA extraction. The enriched mRNA was further segmented into small pieces utilizing fragmentation buffer, followed by reverse transcription into cDNA with the NEBNext Ultra RNA Library Prep Kit for Illumina (New England Biolabs, catalog no. 7530). After repairing the ends

of the purified double-stranded cDNA segments, a base was introduced before ligation to Illumina sequencing adaptors for further analysis. Utilizing AMPure XP Beads (1.0×), the ligation reaction products were subjected to purification. Ligated fragment size was estimated with agarose gel electrophoresis, and PCR was conducted to amplify them. Gene Denovo Biotechnology Co. conducted sequencing of the generated cDNA library on an Illumina Novaseq6000.

Raw reads comprising adaptors or low-quality bases were collected from the sequencer devices and processed further using Fastp (version 0.20.0)[49]. The following parameters were used: (1) adaptor-containing reads were eliminated, (2) reads with over 10% of their nucleotides (N) unknown were eliminated, and (3) low-quality reads that contained over 50% low-quality ($Q$-value ≤ 20) bases were eliminated. During the mapping process, we utilized the short-read alignment program Bowtie2 (version 2.2.8) to map reads to the ribosomal RNA (rRNA) repository[50]. Next, the rRNA-mapped reads were deleted. The residual clean reads were subjected to the assembly process and computation of gene abundance. The assembly of mapped reads from each sample was performed utilizing StringTie v1.3.1 as part of an approach that was based on reference data[51,52]. Utilizing the RSEM program, a fragment per kilobase of transcript per million mapped reads (FPKM) score was computed for each transcribed area to assess the abundance of and variations in the expression of the transcribed region[53].

The DESeq2 program was employed to conduct a differential expression study of RNAs between two distinct groups[54]. The genes/transcripts with an FDR < 0.05 and absolute fold change ≥ 2 were deemed to be differentially expressed. GO and KEGG studies were carried out to assess functional enrichment for the purpose of defining the biological activities of DEGs. The estimated t-test was subjected to FDR correction, with a criterion of FDR ≤ 0.05 employed as the threshold.

## MeRIP-seq and bioinformatic analysis

Following total RNA isolation, the enriched RNA was segmented into small pieces (approximately 100 nt) with the use of a fragmentation buffer. The processed RNA was divided into two parts, one of which was used as the input. The other RNA fraction was enriched with an m6A-specific antibody (Abcam, ab208577). With the use of random primers, the enriched RNA was transcribed into cDNA. Afterward, the cDNA segments were end-repaired, followed by ligation to Illumina sequencing adaptors. Ultimately, we obtained a qualified library for sequencing, which was sequenced on an Illumina NovaSeq™ 6000.

Fastp software (version: 0.20.0) was utilized to analyze the sequencing results to obtain high-quality clean reads in accordance with four severe filtering requirements: (1) elimination of adaptor-containing reads, (2) elimination of reads that contained over 10% unidentified nucleotides (N), (3) elimination of reads that were entirely composed of A bases, and (4) elimination of low-quality reads that encompassed over 50% low-quality ($Q$-value ≤ 20) bases. For peak calling, exomePeak2 (version: 1.0.0) software was used to determine the read-enriched regions from MeRIP-seq data[55]. On the basis of the uniquely mapped reads, the $p$ value for a given region was computed utilizing a dynamic Poisson distribution. When the $p$ value was < 0.05, the region was designated as a peak. Peak-associated genes were validated utilizing the peak's genomic position and gene annotation data. In addition, the distribution of peaks on different functional regions, including the 3′UTR, 5′UTR, and CDS, was further assessed. For motif analysis, MEME suite (http://meme-suite.org/) and DREME (http://memesuite.org/tools/dreme) were used to detect any significant sequence motifs in each transcript sequence associated with peaks.

## Isolation, sequencing, and analysis of ribosome-protected fractions

The ribosomal profiling technique was carried out as described previously, with several changes as detailed below[56]. In brief, to block

translational elongation, cells were cultured with cycloheximide (Sigma) at a final dose of 100 μg/mL and subjected to 2 min of incubation. Then, after cell lysis, the samples were centrifuged at 20,000 × g and 4 °C for 10 min, and the supernatants were collected. For the preparation of ribosome footprints, 6 μL DNase I (New England Biolabs) and 10 μL RNase I (New England Biolabs) were added to 400 μL lysate, which was then incubated for 45 min at ambient temperature. Ten microliters of SUPERase·In RNase inhibitor (Ambion) was added to terminate nuclease digestion. One hundred microliters of digested ribosome footprints was poured into size exclusion columns (Illustra MicroSpin S-400 HR Columns) before being centrifuged for 2 min at 600 × g. After that, the elution was supplemented with 10 μL 10% (wt/vol) SDS, and ribosome footprints larger than 17 nt were extracted with an RNA Clean and Concentrator-25 kit (Zymo Research). In addition, rRNA was extracted utilizing an approach described previously[57]. Finally, magnetic beads (Vazyme) were employed to thoroughly purify the ribosome footprints.

With the use of the NEBNext® Multiple Small RNA Library Prep Set for Illumina®, Ribo-seq libraries were created after the acquisition of ribosome footprints as described above. In summary, adaptors were attached to both extremities of ribosome footprints, followed by PCR amplification and reverse transcription. An Illumina HiSeq™ X10 platform was utilized to sequence the 140- to 160-bp PCR products, which were enriched to develop a cDNA library.

Fastp was used to filter out low-quality reads. Raw reads constituting over 50% low-quality bases or over 10% N bases were eliminated from the analysis. It was necessary to trim adaptor sequences. Reads with a length ranging from 20 to 40 bp were retained for further investigation. To map reads to the rRNA, GenBank, and Rfam databases, the short-read alignment program Bowtie2 was employed. The reads mapped to rRNAs, small nucleolar RNAs (snoRNA), transfer RNAs (tRNA), small nuclear RNAs (snRNA), and miRNA were eliminated. Whole-genome synchronization of processed RNA-seq reads was performed utilizing STAR with the 2-pass setting enabled[58]. We then discarded the reads that corresponded to traditional noncoding RNAs (such as long noncoding RNAs and microRNA precursors). Subsequently, according to the location of the 5′ end of the synchronization, the residual ribosome footprints were allocated to distinct genomic characteristics (5′UTR, CDS, 3′UTR, and intron). The number of reads in the open reading frame of a coding gene was estimated utilizing the program RSEM, and the level of gene expression was standardized by utilizing the FPKM technique.

## Proteomics sample preparation

To inhibit the protease activity, we utilized a lysis buffer composed of a 1x protease inhibitor cocktail to suspend appropriate samples. Moreover, the mixture was allowed to settle at a temperature of 4 °C for 30 min, with constant vortexing every 10 min. Furthermore, the BCA method was employed to determine the protein supernatant concentration after carrying out the centrifugation process at 20,000 g for a duration of 20 min at 4 °C.

For digestion, 2 μL 0.5 M Tris (2-carboxyethyl) phosphine was utilized for 100 μg protein solution at a temperature of 37 °C for 60 min, and alkylation was conducted in darkness at room temperature using 4 μL 1 M iodoacetamide for a period of 40 min. To achieve protein precipitation, a fivefold volume of cold acetone was added and kept overnight at a temperature of −20 °C. After performing centrifugation at 12,000 g for a period of 20 min at 4 °C, the pellet was rinsed two times with 1 mL pre-chilled 90% aqueous acetone solution, and re-suspension of the pellet was performed using 100 μL 10 mM triethylammonium bicarbonate buffer. Furthermore, the addition of a 1:50 trypsin-to-protein mass ratio of trypsin (Promega) was carried out, and incubation was conducted overnight at 37 °C. Moreover, C18 ZipTip was utilized to carry out the desalting process of the peptide mixture, quantification was performed using Pierce™ Quantitative

Colorimetric Peptide Assay, and lyophilization was conducted using SpeedVac.

TMT-11Plex (Thermo Fisher) reagents were utilized to label the trypsin-digested peptides following the protocols provided by the manufacturer. Every TMT reagent unit was thawed, and reconstitution was performed using 50 μL acetonitrile at room temperature for 2 h. Subsequently, the addition of hydroxylamine was done for a period of 15 min at room temperature, and pooling, desalting, and vacuum-drying of each sample were conducted.

## High pH reverse phase separation and nano-HPLC-MS/MS analysis

Buffer A was employed for the dissolution of the peptide mixture. The composition of buffer A was as follows: 20 mM ammonium formate prepared in water, pH of 10.0, regulated by ammonium hydroxide. In addition, the fractionation procedure was carried out using high pH separation utilizing the Ultimate 3000 system (Thermo Fisher) coupled with a reverse phase column (XBridge C18 column, with the following dimensions: 4.6 mm × 250 mm, 5 μm, Waters Corporation). Moreover, High pH separation was carried out by utilizing a linear gradient, starting from a 5% concentration of buffer B, which was gradually increased to 45% over 40 min. The composition of Buffer B was as follows: 20 mM ammonium formate prepared in 80% concentration of ACN, pH of 10.0, regulated using ammonium hydroxide. The column re-equilibration was carried out at the initial condition for 15 min, and the flow rate of the column was sustained at 1 mL/min at 30 °C. In total, the collection of 12 fractions was successfully completed, and the drying of all the fractions was carried out in a vacuum concentrator for the subsequent step.

The re-dissolution of peptides was carried out in solvent A, which was composed of: a 0.1% formic acid solution prepared in water and evaluated using Q-Exactive HF-X coupled to an EASY-nanoLC 1200 system (Thermo Fisher). The peptide sample of volume 3 μL was loaded onto an analytical column (25 cm, 75 μm inner diameter, 1.9 μm resin), and separation was carried out by setting a 90 min-gradient, which started at 6% buffer B concentration (80% ACN with 0.1% FA) for 1 min. Gradually, the concentration was increased to 60% in 79 min, and in 2 min, the concentration was increased to 90% and held there for a period of 9 min. The flow rate of the column was sustained at 250 nL/min at 55 °C, and 2 kV was fixed as the electrospray voltage.

The mass spectrometry was performed following data-dependent acquisition mode and automatically switched between MS and MS/MS mode. The survey of full scan MS spectra (m/z 350-1800) was acquired in the Orbitrap with 60,000 resolution. Moreover, the precursor ions were selected into the collision cell for fragmentation by higher-energy collision dissociation; the normalized collection energy was 32. The MS/MS resolution was set at 30,000, with the automatic gain control target of 5e4, the maximum injection time of 50 ms, and dynamic exclusion of 10 s.

## Proteomics data analysis

The PEAKS Studio version 10.6 (Bioinformatics Solutions Inc.) was employed to process tandem mass spectra. Trypsin was assumed to be a digestive enzyme when PEAKS DB was set up to search the GDP21120524 database. PEAKS DB was searched with a parent ion tolerance of 10 ppm and a fragment ion mass tolerance of 0.02 Da. TMT-11Plex (K, N-term) and Carbamidomethylation (C) were specified as the fixed modification, whereas, Acetylation (Protein N-term) and Oxidation (M) were specified as the variable modifications. Filtration of peptides was carried out using 1% FDR and proteins were filtered by means of one unique peptide. Reporter ions were used to evaluate the ratio of quantification among various samples, and normalization was derived from the total intensity of every label in each quantifiable peptide.

## eCLIP-seq

As previously mentioned[59], eCLIP was conducted with a few minor adjustments. The UV cross-linking (254 nm, 400 mJ/cm²) was employed for the stabilization of the RNA binding protein (RBP)-RNA interactions, followed by using 1 mL iCLIP lysis buffer to carry out lysis. Furthermore, RNase A (Sangon Biotech, NO. B600476) was utilized for regulated digestion, and RBP−RNA complexes' immunoprecipitation was performed with the aid of the YTHDF1 antibody (20 μg/test; Abcam, catalog no. ab220162) using Dynabeads™ Protein A (Thermo Fisher, catalog no. 10002D), and stringent washes. Moreover, after completing dephosphorylation by employing T4 PNK (Vazyme, catalog no. N102) and FastAP (Thermo Fisher, catalog no. EF0654), ligation of a barcoded RNA adapter was carried out at the 3' end (T4 RNA Ligase, NEB). On-bead ligations were carried out to easily wash away unincorporated adapters, and a high concentration of PEG8000 was used to increase ligation efficiency to over 90%. Size selection was conducted using standard protein gels followed by transfer to nitrocellulose membranes. Sample digestion was carried out by employing the proteinase K (TransGen Biotech, catalog no. GE201) digestion buffer to extract RNA. Furthermore, RNA Reverse transcription was conducted using HiScript® III Reverse Transcriptase (Vazyme, catalog no. R302), and VAHTS DNA Clean Beads (Vazyme, catalog no. N411) treatment was performed for the removal of excessive oligonucleotides. The ligation of the second DNA adapter was carried out at the 3' end of the cDNA fragment (T4 RNA Ligase, NEB) in a high concentration of DMSO and PEG8000 in order to reduce ligation inhibition caused by secondary structures. After conducting cleanup, PCR amplification was performed, and size selection was performed with the aid of VAHTS DNA Clean Beads (Vazyme, catalog no. N411). The cDNA libraries were then sequenced by employing Illumina.

## Single-cell dissociation

A scRNA-seq was carried out at the laboratory of NovelBio Co., Ltd. by experienced research personnel. We extracted tissues surgically and stored them in MACS Tissue Storage Solution (Miltenyi Biotec, catalog no. 130-100-008) until they could be processed. The tissue specimens were tested according to the procedures outlined below. Subsequently, the samples were prepared by washing in PBS, mincing into tiny pieces (about 1 mm³) on ice, and digestion with reagents from a Mouse Tumor Dissociation Kit (Miltenyi Biotec, catalog no. 130-096-730) in accordance with the package recommendations. Following digestion, the specimens were filtered through a 70-micron cell strainer and centrifuged at 300 × g for 5 min. Following the elimination of the supernatant, the red blood cells were lysed by suspending the pelleted cells in red blood cell lysis solution (Miltenyi Biotec, catalog no. 130-094-183). Anti-mouse CD45 (BD Pharmingen, catalog no. 553079) was employed to label the cell surface. The cells were labeled for 45 min at 4 °C while protected from light. DAPI (Sigma #D8417, 1 μg/mL) was added to the cell solution just before sorting. In this study, cells were separated utilizing a BD FACSMelody cell sorter (BD Biosciences) and then recovered in a solution containing 20% FBS/PBS. Subsequently, 80,000–100,000 DAPI- CD45+ cells were extracted and analyzed from each specimen. Gating strategy of flow-sorting was provided in Supplementary Fig. 12. It was necessary to wash and resuspend the single-cell suspension in PBS comprising 0.04% BSA. The suspension was further refiltered through a 35-micron cell strainer before staining with AO/PI for viability evaluation utilizing a Countstar Fluorescence Cell Analyzer.

## Single-cell sequencing

A number of tools, including the 10X Genomics Chromium Controller Device, a Chromium Single Cell 5' library & gel bead kit, and the V(D)J enrichment kit (10X Genomics, Pleasanton, CA), were employed to create scRNA-seq and V(D)J libraries. The following

procedures were performed: cells were concentrated to approximately 1000 cells per microliter and placed into each channel to generate single-cell gel beads-in-emulsion (GEMs). Following the reverse transcription stage, the GEMs were fragmented, and the barcoded cDNA was subjected to purification and amplification. The amplified barcoded cDNA was employed to develop TCR-enriched libraries and 5' gene expression libraries. When creating a 5' library, the amplified barcoded cDNA was fragmented, A-tailed, ligated to adaptors, and then amplified utilizing index PCR. When constructing a V(D)J library, mouse T-cell V(D)J sequences were enriched from amplified cDNA, which was then fragmented, A-tailed, ligated to adaptors, and amplified with index PCR. The Qubit High Sensitivity DNA Assay (Thermo Fisher Scientific) was employed to quantify the final libraries, and each library's size distribution was assessed with a High Sensitivity DNA chip on a Bioanalyzer 2200 (Agilent). Sequencing of each library was carried out with a 150-bp paired-end run on an Illumina sequencer (Illumina).

## Statistical analysis of single-cell RNA data

NovelBio Co., Ltd. completed the scRNA-seq data analysis utilizing the NovelBrain Cloud Analysis Platform, which was developed by NovelBio (www.novelbrain.com). To obtain clean data, we employed Fastp with the default setting for screening adaptor sequences and removing low-quality reads. The feature-barcode matrix was again generated by mapping reads to the mouse genome (Ensembl 100) utilizing Cell Ranger v6.1.1. We performed a downsample analysis on specimens that had been sequenced on the basis of the aligned barcoded reads for each cell sample, and we successfully obtained an aggregated matrix. It was determined that cells with more than 200 expressed genes and a mitochondrial UMI rate of less than 20% satisfied the cell quality screening, and mitochondrial genes were deleted from the expression tables.

To acquire scaled data, Seurat software (version 4.0.3, available at https://satijalab.org/seurat/) was employed for cell standardization and regression based on the expression table illustrating each sample's UMI counts and mitochondrial rate percentages. Principal component analysis was performed with scaled data, including the top 2000 genes with a high degree of variability, and the top 10 principal components were utilized for t-SNE development. After obtaining the unsupervised cell cluster result premised on the top 10 primary principal components, we used the FindAllMarkers tool in conjunction with the Wilcoxon rank-sum test technique to identify marker genes. The criteria below were used: (1) $\log_2 FC > 0.25$, (2) $p$ value < 0.05, and (3) min.pct > 0.1. In addition, clusters with similar cell types were chosen for marker analysis, graph-based clustering, and repeat t-SNE analysis to determine the cell type in detail.

To analyze single-cell TCR-sequencing data, we used Cell Ranger (version 6.1.1, 10× Genomics) and compared the data to a mouse VDJ reference given by 10× Genomics. For subsequent analyses, including evaluations of the distribution of CDR3 length, the utilization trends of VDJ gene fragments and combinations, and the variety of TCR repertoires, cells containing at least one TCR α or β chain were kept. If two or more cells in a sample contained comparable α-β pairs, these T lymphocytes were designated clonal T lymphocytes, and they all shared a distinct clonotype ID, which was recorded for each sample. TCR-based analysis was performed exclusively on cells that were recognized as T cells to combine the findings of the TCR analysis with the data on gene expression. We identified T-cell clonotypes shared among clusters or samples.

## DEG analysis of single-cell sequencing data

The FindMarkers function, together with the Wilcoxon rank-sum technique, was employed to identify DEGs in compared samples using the following criteria: (1) $\log_2 FC > 0.25$, (2) $p$ value < 0.05, and (3) min.pct > 0.1.

## Pathway analysis of single-cell sequencing data

The KEGG database was utilized to determine the significant pathways enriched in genetic markers and DEGs. Fisher's exact test was adopted to identify statistically significant pathways, and the cutoff value for significance was set by the $p$ value and the FDR.

## TCGA and GEO data analyses

RNA-seq expression (level 3) profiles and corresponding clinical information for 34 different types of tumors were downloaded from the TCGA database (https://portal.gdc.com). Gene expression was normalized to the TPM value for further analysis. The Hugo et al. cohort and the Riaz et al. cohort, which are under the accession numbers GSE78220 and GSE91061, respectively, were downloaded from the GEO database[24,25]. Survival analysis was performed using RNA-seq data and clinical information from GSE65904 SKCM dataset[60,61]. The Survminer package (version 0.4.7) and survival package (version 3.2-3) were used to generate survival curves.

The immune score was calculated using the ESTIMATE package in R, a method that uses gene expression signatures to infer the fractions of immune cells in tumor samples[23]. ssGSEA was performed to quantify the 28 types of immune cells infiltrating each melanoma sample and was accomplished using the "GSVA" and "GSEABase" packages in R software. The ssGSEA enrichment score was then used as the measure of immune cell infiltration in each sample[62]. The gene signatures for the different immune cell types were obtained from previously published data[63].

GSEA was used to identify the significantly altered signaling pathways in each group, as identified by their enrichment in the MSigDB collection (c2.kegg.v7.1 symbols.gmt; c5.bp.v7.1 symbols.gmt). Gene set permutations were set at 1000 repeats for each analysis.

TIDE analysis was conducted to evaluate the ICB response. Jiang et al. designed an analytic technique known as TIDE (http://tide.dfci.harvard.edu/), which enables the prediction of the ICB response using two major tumor immune evasion mechanisms: T-cell dysfunction induced in tumors with high cytotoxic T-lymphocyte (CTL) infiltration and T-cell infiltration inhibited in tumors with low CTL levels[26].

## ROC curve analysis

The ROC curve was applied to evaluate the predictive efficiency of *YTHDF1* expression for the anti-PD-1 response rate. The *YTHDF1* expressions of response and non-response groups were input and the curve was generated and visualized using pROC package and ggplot2 package in R software. The area under the curve was calculated as a single measure to discriminate efficacy. When the ROC curve produced an area under the curve above 0.7, a cutoff value was determined with high specificity and positive predictive value.

## Statistical analysis

GraphPad Prism 8.4.1 was used to analyze statistical data. When comparing two groups, two-tailed unpaired Student's $t$-test or Wilcoxon test were utilized. When there were more than two groups to be compared, one-way ANOVA or two-way ANOVA was used. The two-tailed log-rank test was employed to compute the $p$ values for Kaplan–Meier survival curves. Data are presented as the mean ± SD, and the exact $p$ values were labeled in the figures.

## Reporting summary

Further information on research design is available in the Nature Portfolio Reporting Summary linked to this article.

# Data availability

The mass spectrometry proteomics data have been deposited in the ProteomeXchange Consortium via the PRIDE partner repository with the dataset identifier PXD036938. The raw sequence data of bulk RNA-seq data generated in this study have been deposited in the Sequence

Read Archive (SRA) database with accession numbers PRJNA822063. The raw sequence data of ribosome RNA-seq data generated in this study have been deposited in the SRA database with accession numbers PRJNA822766. The raw sequence data of MeRIP-seq data generated in this study have been deposited in the SRA database with accession numbers PRJNA911485. The raw sequence data of single-cell RNA-seq data generated in this study have been deposited in the SRA database with accession numbers PRJNA911622. RNA-seq expression (level 3) profiles and corresponding clinical information for 34 different types of tumors were downloaded from the TCGA database (https://portal.gdc.com). The Hugo et al. cohort used in this study is available in the GEO database under the accession code GSE78220. The Riaz et al. cohort used in this study is available in the GEO database under the accession code GSE91061. The melanoma publicly available data used in this study are available in the GEO database under accession code GSE65904. The remaining data are available within the article, Supplementary information or Source Data file. Source Data are provided with this paper.

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

## Acknowledgements

This work was supported by the Science and Technology Development Fund of Shanghai Pudong New Area (project number PKJ2020-Y53, received by J.G.), Joint Breakthrough Project for New Frontier Technologies of the Shanghai Hospital Development Center (project number SHDC12019120, received by J.J.L.), the Natural Science Foundation of China (project number 82072986, received by S.F.Q.; 82103059, received by H.J.Z.), and Shanghai Natural Science Foundation (project number 22ZR1458700, received by L.K.). Figures 4A, 7G, 8A and 9 and Supplementary Fig. 3A–D were created with BioRender.com. We are grateful to Gene Denovo Biotechnology and NovelBio Biotechnology for assisting in sequencing and bioinformatics analysis.

## Author contributions

J.J.L. and L.K. conceptualized the study. W.Z.L., Li.C., H.J.Z., X.X.Q., Q.T.H., F.Z.W., Z.Y.L., S.K.G., A.L.Z., and Lo.C. performed methodology. W.Z.L., and S.F.Q. analyzed data. W.Z.L. wrote the original draft. J.J.L. and L.K. supervised the manuscript.

## Competing interests

The authors declare no competing interests.
