## [Peer Review File · Nature Communications]

Tumor-intrinsic YTHDF1 drives immune evasion and resistance to immune checkpoint inhibitors via promoting MHC-I degradationREVIEWER COMMENTS

Reviewer #1 (Remarks to the Author): with expertise in RNA Methylation, m6A, YTHDF1

This is a quite extensive work aimed to address tumor intrinsic effect of YTHDF1 in shaping tumor micro environment (TME) and response to immunotherapy. Both database analysis and immunocompetent mouse model show tumor-intrinsic immunosuppressive role of YTHDF1. The authors went on to collect a large set of data to support their observation. Here the manuscript appears to fall apart.

1. The authors discussed RNA-seq results from wt and KO cells. However, if YTHDF1 is promoting translation RNA level changes are secondary to protein level changes. The authors need to start with identifying m6A-modified transcripts, revealing targets of YTHDF1 and then perform ribo-seq to integrate these data together. From this analysis the authors could identify direct target transcript of YTHDF1. I strongly urge the authors to employ CLIP-seq with antibody targeting native YTHDF1. The number of transcripts revealed by RIP-seq is simply too large. Protein level changes versus RNA level changes should be compared.

2. With target transcripts identified authors can perhaps continue to explain gene expression changes affected by YTHDF1 through its translational role. And immunosuppressive effects.

3. The single-cell seq of TME is nice but this section is long without focus. Most can be moved to SI. A key question the authors should address here is why and how tumor-intrinsic changes caused by YTHDF1 induces TME changes. It is very hard to think translation of one gene could have such a profound effect.

4. Ideally, the authors can employ cas13b-FTO or ALKBH5 system to remove m6A on the target transcript and follow either mRNA level or translation changes to establish causal relationship.

5. Discussion is too long. I would cut at least half and focus on how tumor intrinsic effect of YTHDF1 shapes TME. Note that the previous YTHDF1 DC work also showed a negative correlation of T cell infiltration versus stroma YTHDF1 level in patient samples. Any signaling between tumor and TME that may coordinate expression of YTHDF1?

Reviewer #2 (Remarks to the Author): with expertise in Pygo, Wnt signaling

In the submitted paper, the authors investigate the role of the RNA-methylation reader YTHDF1, in modulating the immune response to tumor cells. In particular, they show that high YTHDF1 expression in skin cutaneous melanoma (SKCM) cells is associated with an absent immune response against these cells, and resistance to immune checkpoint inhibition (ICI). Via scRNA-seq data, the authors identify population of actively engaged differentiated immune cells (e.g., more functional CD4+, Tregs and IFN-responsive CD4+ T-cells) likely capable of neutralizing the tumor cells, only when YTHDF1 has been depleted. This is possibly the result of enhanced expression of cytotoxic molecules upon YTHDF1 knockout (KO), and the activation of immunological processes and pathways. Among the target mRNA of YTHDF1, the authors identify that encoding for the Wnt transcriptional regulator PYGO1. Consistently, PYGO1 protein is reduced in YTHDF1-KO cells, and this is paralleled by lower nuclear beta-catenin.

Finally, the authors develop an exosome-based strategy to deliver CRISPR-Cas9 components to tumor cells in vivo and achieve stable downregulation of YTHDF1, with the consequence of significantly enhancing ICI approaches.

The article presents an impressive series of analyses which convincingly show that tumor intrinsic YTHDF1 is an important regulator of RNA translation efficiency, which in turn drives the tumor's ability to escape the immune checkpoint regulators. The article is well written, of interest for a broad audience, and certainly worth being reported in Nature Communications. I also find commendable

that the authors, in addition to looking for a mechanism, strive to develop a strategy to target tumor specific YTHDF1 in order to render the tumors sensitive to ICI. Overall, the article strength lies in the robust characterization of the interplay between the immunological components that act in the presence of high or low tumor intrinsic YTHDF1. As it will be apparent below, my primary concerns regard the connection between the role of YTHDF1 and the activation of Wnt signalling. This part, in my opinion, is relatively weak and should be supported by more convincing data or, alternatively, toned down in its importance.

Main points:

1) It is not clear why the authors focus on Pygo1 in the first place. Considering the decrease in mRNA translation upon YTHDF1-KO, 105 target mRNA do not appear neither in the RIP-seq nor in the MeRIP-seq lists. While I agree with the authors that focusing on the 37 that are common in the three lists is the safe choice, assays such as RIP-seq and MeRIP-seq could easily have false negatives. Therefore, I feel it is plausible that several others among the differentially translated mRNA might be real YTHDF1 targets. Are there potentially relevant YTHDF1 targets among these? The need of looking for other contributing factors becomes relevant in particular considering the quite weak – in my opinion – connection with PYGO1 (see comments below).

2) The data supporting that PYGO1, in this context, affects beta-catenin nuclear versus cytosolic distribution is not fully convincing. For instance, the loading control of the nuclear fraction, Histone H3, is broadly saturated, and does not allow assessment of the subtle decrease in nuclear beta-catenin protein. Moreover, could the detection of a lower amount of nuclear beta-catenin be the consequence of reduced stabilization of the protein, rather than of its nuclear import? Perhaps, when looking at total beta-catenin, the small decreased is masked by the largely more abundant membranous (as opposed to cytosolic>nuclear) beta-catenin? Moreover, this observation is rendered more cryptic by the fact that, in the same blot, other nuclear proteins (LEF1 and TCF7) are reduced. Is it possible that, rather than nuclear import, YTHDF1 KO causes a more general effect on translation thereby reducing the abundance of many more targets that were identified (see my comment 1)?

3) It comes as surprise that PYGO1 has such an important role. Several developmental studies clearly attributed a much more important involvement of PYGO2 in Wnt/beta-catenin signalling, where, notably, knockout of Pygo1 has essentially no detectable effect (e.g., PMID: 17425782; PMID: 23637336). I wonder whether the footprints identified on Pygo1 mRNA could also be attributed to Pygo2? Have Pygo2 mRNA and protein relative abundance been measured in this context? Is it possible that PYGO1 and PYGO2 are redundant, and that YTHDF1 stabilizes the mRNA encoding for both the paralog proteins?

4) The authors use the Wnt inhibitor LGK974 to confirm the involvement of Wnt/beta-catenin signalling. However, LGK974 inhibits the secretion of all the Wnt ligands (PMID: 24277854) and has therefore broad effects extending to both canonical and non-canonical pathways. While the effect observed is in line with a potential role of the PYGO1>beta-catenin axis, more targeted approaches should be employed to effectively show that PYGO1 is the main player (see comment below).

5) Related to the above, I believe that one key experiment could clearly determine if PYGO1 is the key downstream effector of YTHDF1, that is the assessment of PYGO1-KO in tumor cells. This might be elegantly achieved via exosomes as done for YTHDF1. If the authors' model is correct, PYGO1 depletion should mimic – at least in part – the loss of YTHDF1.

6) How could the authors assess the YTHDF1-KO exosomes also do not target several other non-tumor tissues in vivo? The strategy is elegant, but it lacks evidence of safety and absent toxicity.

7) One of the key points (e.g., from Figure 3) is that YTHDF1-KO in tumors leads to activation of immunological processes and pathways in immunocompetent mice, which in turn recruits more tumor-infiltrating CD4+ and CD8+ T cells. Could this occur because, upon YTHDF1-KO, tumor cells become “simply” less fit and/or senescent-like, and attract the immune armies as a consequence of this? The

authors themselves recognize that this effect results from the enhanced expression of cytotoxic molecules upon YTHDF1 KO and the subsequent activation of immunological pathways. This explanation might not require the involvement of the PYGO1 > beta-catenin mechanism suggested. How do the authors reconcile these two alternative mechanisms?

8) In Figure 6 the authors show that YTHDF1 deficiency increases PD-L1 expression in vivo but not in vitro. First, it is not clear why there is this difference between the two contexts. Second, shouldn't PD-L1 increase induce a protective role in tumor cells in the absence of ICI treatment?

Other minor points:

9) The authors should consider being consistent with the gene/mRNA/protein nomenclature, as sometime (e.g., in the abstract), they refer to YTHDF1 or Ythdf1.

10) I believe the authors should explain better how they have developed the ROC curve (lines 99-101) to make the reader understand how reliable it is.

11) Line 110-111: correct in "was resistant to ICI therapy."

12) Line 111: "We utilized the B16/F10 mouse model.." could the authors explain this better? B16/F10 is a mouse-derived cell line.

13) The authors write "subgroup analysis revealed the significant prognostic value of YTHDF1 in the TGFB1-low group but not the TGFB1-high group, which indicated that the prognostic impact of YTHDF1 is associated with immunity". It is not clear to me what the relation is between TGFB signalling in melanoma and the conclusion of involvement of the immune system.

14) Lines 258-260, the authors write "we determined that CD4 TCR diversity and clonotype expansion were increased in Ythdf1-KO tumors (Figure 4J and 4K), and were mostly distributed in functional CD4+ T cells rather than naive cells (Figure 4L), suggesting tumor specificity". It is not clear why the authors could conclude "tumor specificity" – I suggest rephrasing or provide a better explanation to this statement.

15) In Figure 4B, left panel, it is not clear how the cell clusters should be interpreted: please provide a better explanation on the difference between the number-indicated cell clusters and cell type derivation. For example, it seems that a uniform cell population of Mono/Macro is deduced from clusters 4, 0 and 3. How is this possible?

Reviewer #3 (Remarks to the Author): with expertise in cancer immunology, scRNAseq

In the manuscript "Tumor-intrinsic YTHDF1 drives immune evasion and resistance to immune 2 checkpoint inhibitors by regulating the Pygo1/β-catenin axis" the authors describe how YTHDF1 effects in tumor cells regulate the immune response in the tumor microenvironment.

1. In figure 3B, the authors should check the labels, as the data shows in the KO tumor tissue, there is actually negative enrichment of immune related pathways, which is in opposition to fig 3A and their conclusions.

2. Lines 207-211: "Overall, the proportions and cell numbers for different cell categories exhibited apparent differences between the KO and WT groups (Figure 4C). Of note, T cells, NK cells, and proliferating mixed immune cells were enriched in Ythdf1-deficient tumors, whereas neutrophils and B cells showed higher infiltration in WT tumors (Figure 4D)." Can the authors include some statistical analysis to back-up their claim? I believe that is probably true for T cells and Nk cells. Regarding conclusions about these analysis, it is worth mentioning the reduction in B cells needs to be understood in line of higher infiltration of other cells in the KO samples. So the likely reason for the proportional reduction of B cells is simply because there are more of other cells. The authors should

also elaborate on the increase of tumor cells (is this technical? Are they similar?) Maybe exclusion of the tumor cells from this analysis would also be necessary, to keep proportional comparisons adequate.

3. Lines 239-242: "By pseudotime analysis, proliferating CD4+ T cells were located at the beginning of the pseudotime trajectory, while naive CD4+ T cells were located in the terminally differentiated state of the branch, with TEMs, Tregs, and IFN-responsive CD4+ T cells being transition states spread along the axis (Extended data figure 5D-E)." The analysis of the pseudotime is very open for misinterpretations (with labeling naïve cells as terminally differentiated). Additionally, as expected, cycling cells drive a large part of the observed effects and complicates rather than helps in the data interpretation. I would suggest exclude the proliferating cells (ideally) or perform some cell cycle correction then repeat the trajectory analysis, and interpret those results under current understanding of T cell differentiation from naïve to other subsets.

4. Figure 5E should be excluded. Direct comparison of cell numbers in single cell between conditions without any normalization to the actual number of collected cells is not scientifically sound.

5. The results from figure 5 are largely driven by a technical/biological aspect of the data, as seen in extended data. The high presence of specific alpha and beta chains are driving the cluster assignments. Even though they are "true biological differences" the capture of TCR alpha and beta chains without specific enrichment protocols is heavily biased. So I would suggest exclusion or correction for all the detected alpha and beta chains in the CD8 T cells. Then repeat of the analysis displayed in this figure.

6. Figure 5G displays a very nice cytotoxic effect of CD8 T cells. Could the authors repeat the same experiment but using CD4 T cells? They identified populations with high granzyme levels that they claim to not be immunosuppressive, so some data to support that would be appreciated.

7. Neutrophils are not always easy to capture in droplet based single cell methods. Can the authors provide some evidence why these are indeed neutrophils and not some monocyte populations?

Reviewer #4 (Remarks to the Author): with expertise in immunology, extracellular vesicles for therapeutic delivery

Review:

This study shows a very interesting topic of the role of RNA methylation and in particular YTHDF1 in regulating immune cell infiltration into tumors and a novel approach to disrupt YTHDF1 dependent immune evasion using exosomes. To my knowledge, this work is original with significant results but the authors should provide more analysis of the exosome mediated gene editing. I have a few major points I would like to be addressed and multiple minor points below:

Major points:

1. Extended Data Figure 3 and the legend are lacking in important details. In the figure Panel J on the left there is no indication about what the different subpanels are, I can imagine the left blue stain is Dapi, the middle is EdU, and the right is merged but this is missing from the figure or the legend. Also I am assuming results comparing wild-type (WT) and knock-out (KO) cells are using WT cells that have not been Puro selected, but this should be described in the legend. Lastly, in Panel C there is a very useful overview of the workflow but I believe it misses the important step of Puromycin selection before the single clone cell screening. Lastly there is a typo on the panel B plasmid map, it reads "spCase9" but should be corrected to "spCas9"

2. In Figures 6C and 6E it is extremely surprising that only some of the comparisons gave statistical significance but not others, did the authors omit statistical comparison among some groups such as WT+IgG vs KO+IgG, KO+CTLA4, or KO+PDL1 which all show no significance? For Figure 6C how can it be explained that WT+CTLA4 and WT+PDL1 have lower tumor volumes with comparable standard deviations to WT+IgG yet only the WT+CTLA4 and WT+PDL1 have statistically significant difference to KO conditions but not the WT+ IgG? Likewise, for Figure 6E how can it be explained that WT+CTLA4 and WT+PDL1 have better survival than WT+IgG yet only the WT+CTLA4 and WT+PDL1 have statistically significant difference to KO conditions but not the WT+ IgG? If only some comparisons were done this should be specified in the figure legend or better if all statistical comparisons are included in the figure.

3. Figure 9D is not adequate to conclude that “typical exosomal structures” are observed as stated by the authors. One problem is that few or only 1 structure is observed per panel, ideally a wide field of view with more vesicles should be used and if desired a single vesicle can be highlighted with an insert zoom. Another problem is that the highlighted structures are closer to 150-200nm in size despite the nanoparticle flow cytometry detecting what appears to be a modal size of 50nm and author’s reported mean size of 65-70nm. Please repeat the TEM measurements with higher concentration of exosomes to visualize multiple particles per field of view. Here is an article with a detailed protocol and see Figure 3.22.2 as good example of what should be aimed for, to have multiple particles in the field of view: DOI: 10.1002/0471143030.cb0322s30

4. There is much evidence that exosome tracking with lipid-anchored fluorescent dyes can lead to artefacts such as dye aggregates or lipoprotein <https://doi.org/10.1080/20013078.2019.1582237> . In the methods section no dye removal step is mentioned after staining so if it hasn’t been done the exosomes should be recentrifuged after staining to pellet and then resuspend in fresh buffer to remove some of the unbound dye. Furthermore, the authors should include at least for the in vitro uptake a minimal control experiment where a “mock EV” condition, consisting of fresh media that was not treated with cells, has been exposed to the same procedure for exosomes isolation and fluorescent labelling, and used to observed the uptake of any dye aggregates this procedure may form. Authors may also try to use previously described strategies of labelling using fluorescent protein fusion constructs that are recognised to be more robust for exosome tracking experiments doi: 10.1038/ncomms8029 .

5. While a very convenient method Exo-Fect transfection of exosomes requires important experimental controls to ensure that the observed cargo transfer is happening by exosomes and not potential residual Exo-Fect and cargo left-over being transferred. To exclude the possibility that the DNA plasmid is being transferred outside exosomes, the authors should include a DNase I treatment step of the Exo-Fect transfected exosomes and they should also include a similar “mock EV” condition described above where the buffer without exosomes is transfected with Exo-Fect. Ideally these two control conditions can be tested by injection intratumorally in mice, or at an absolute minimum they should be tested in vitro on B16F10 cells followed by YTHDF1 and PYGO1 WB. These two conditions would definitively prove that the exosomes are the carriers of the DNA cargo. Furthermore, authors should include more discussion on the use of exosomes for gene editing, comparing their approach of DNA plasmid transfer via exosomes to other works transferring CRISPR Cas9 ribonucleoprotein with EVs such as <https://doi.org/10.1002/jev2.12225> . Are there any potential advantages or disadvantages in a DNA plasmid vs ribonucleotide transfer approach?

Minor points:

6. FTO is found throughout the manuscript but I think it would be useful on the first instance in the introduction to rewrite as follows: “Tumors exploit the fat mass and obesity-associated protein (FTO)-mediated regulation of glycolytic metabolism to evade immune surveillance¹³”.

7. Figure 1 legend should be corrected as follows: (E, F) Box plots showing the differential expression of YTHDF1 in progressive disease (PD), stable disease (SD), partial response (PR), and complete response (CR) samples using RNA sequencing data from the (D) Hugo2016_PD1_Melanoma cohort (n = 18) and Riaz2016_PD1_Melanoma cohort (n = 56).

8. On page 4 there is a mixup about the extended figure 2. Authors refer to this being the CRISPR KO design but in fact it should be extended figure 3. Please correct.

9. It appears various schematics were made using biorender.com, if so it should be included in the methods somewhere.

10. Figure 5B and the manuscript text describe that there is a greater number of infiltrating CD8+ T cells. However, it is difficult to discern since the WT and KO dots are together in the same Figure 5B. Can the authors change this to instead have side by side figures with the WT and KO groups as was done for Figure 4C. Alternatively it could also work if a number was given next to the WT and KO labels.

11. In lines 272-275 of the Manuscript authors state that “Tumor-infiltrating lymphocytes (TILs) from Ythdf1-KO tumors were highly enriched in activated effector T-cell populations, which were assigned largely to the CD8+ TEM, CD8+ TEX, and proliferating CD8+ T-cell states (Figure 5E).” However, there is no statistically significant difference in TEX and proliferating CD8+ cells. The authors should

only claim something like "...Ythdf1-KO tumors were highly enriched in activated effector T-cell populations of the CD8+ TEM CD8+ T-cell states, while some not statistically significant increase was observed for TEX and proliferating CD8+ T-cell states (Figure 5E)."

12. In Figure 7E, along the legend please include the number of differential TE transcripts for the highlighted m6A+Ythdf1 target as you have done for the UP and down. Also please be consistent and use either "Up" with "Down" or "UP" with "DOWN" for the legend.

13. In line 408 there seems to be a typo, the decreased Tcf1 and Lef1 expression is only seen in the WB shown in Figure 8C, not in the confocal microscopy of Figure 8B and 8C. Please revise "(Figure 8B and C)." to "(Figure 8C)."

14. Please revise Figure 8G so that both immunodeficient and immunocompetent tumour volumes have the same y-axis range (max value and marks).

15. Figure 9B has low resolution/pixelated axis labels and numbers. Please correct to have labels and numbers of similar resolution than the other labels throughout the manuscript.

16. In multiple sections 200ug of exosomes are described to be used, however there are no details as to what is measured to be 200ug (exosome protein or lipid?) or what technique is used to quantify the exosomes (eg microBCA, Bradford, SPV, etc).

Point-by-point response

Reviewer #1 (Remarks to the Author)

This is a quite extensive work aimed to address tumor intrinsic effect of YTHDF1 in shaping tumor micro environment (TME) and response to immunotherapy. Both database analysis and immunocompetent mouse model show tumor-intrinsic immunosuppressive role of YTHDF1. The authors went on to collect a large set of data to support their observation. Here the manuscript appears to fall apart.

Response: We sincerely appreciate the time and effort that you dedicated to providing feedback on our manuscript and are grateful for the insightful comments and valuable improvements to our paper. We have revised the manuscript extensively according to the comments and supplemented eCLIP-seq, proteomics, and substantial *in vitro* and *in vivo* experiments to further elucidate how tumor-intrinsic YTHDF1 drives immune evasion. We also appreciate the encouraging comments about the manuscript and hope that the correction will meet with approval.

1. The authors discussed RNA-seq results from wt and KO cells. However, if YTHDF1 is promoting translation RNA level changes are secondary to protein level changes. The authors need to start with identifying m6A-modified transcripts, revealing targets of YTHDF1 and then perform ribo-seq to integrate these data together. From this analysis the authors could identify direct target transcript of YTHDF1. I strongly urge the authors to employ CLIP-seq with antibody targeting native YTHDF1. The number of transcripts revealed by RIP-seq is simply too large. Protein level changes versus RNA level changes should be compared.

Response: Thanks for these valuable comments. We agree that assays such as RIP-seq could easily have false positives. **Therefore, we performed enhanced CLIP-sequence to identify YTHDF1 targets and reduce false positives.** eCLIP is an enhanced version of the crosslinking and immunoprecipitation (CLIP) assay, and is used to identify the binding sites of RNA binding proteins (RBPs). Because the CLIP and iCLIP methods often result in high duplication rates and low library complexity, eCLIP seeks to improve the efficiency and quality of library production (van Nostrand, et. al, *Robust transcriptome-wide discovery of RNA-binding protein binding sites with enhanced CLIP (eCLIP)*, Nature Methods (2016),13: 508–514). By simplifying the generation of paired IgG and size-matched input controls, eCLIP improves specificity in the discovery of authentic binding sites and maintains single-nucleotide binding resolution. As presented by the binding peaks of YTHDF1, eCLIP showed low signal

noise compared to RIP-seq.

Compared to RNA, proteins are more stable molecules involved in the central performance of various cellular processes. To comprehensively elucidate the mechanism regarding how tumor-intrinsic YTHDF1 drives immune evasion, we initially supplemented proteomics to reveal the alteration of intrinsic proteins and pathways between B16/F10 WT (n=5) and KO (n=5) groups. **In the proteomics analysis, we focus on investigating the altered pathway or a panel of genes instead of one single gene.** Totally, we identified 979 differentially expressed proteins, with 200 up-regulated and 797 down-regulated proteins (Figure 7A). Noteworthy, most of the differentially expressed proteins were down-regulated in *Ythdf1*-KO group, which fit well with the critical role of YTHDF1 in promoting translation. **Functional enrichment analysis showed that lysosome was the most significantly altered pathway down-regulated in KO group (Figure 7B).** Next, we integrated eCLIP-seq, MeRIP-seq and Ribo-seq to further reveal the regulator function of YTHDF1. Strikingly, most lysosomal genes were identified as YTHDF1-targeted and m⁶A-marked transcripts followed by attenuated translation

(Figure 7C). Given that YTHDF1 is well known to affect mRNA translation in m6A manner, our results indicate that YTHDF1 deficiency impedes lysosomal genes translation, which ultimately reduces protein expression and obstructed the generation of lysosomes. Indeed, flow cytometry and transmission electron microscope further confirmed the attenuated lysosome in the *Ythdf1*-KO group (Figure 7D and 7E).

It is known that lysosomal proteolysis promotes immune evasion of tumor cells by degrading tumor antigen and MHC-I (*Nature* 2020, PMID: 32376951). Here, we hypothesized that *Ythdf1* knockout limited lysosomal proteolysis in tumor cells, ultimately enhancing the MHC-I expression and minimizing the destruction of internalized antigens. **These could increase the immunogenicity of tumor cells and make them become “simply” less fit, and attract the immune armies to restore tumor immune surveillance.** To test this hypothesis, we initially analyzed the surface MHC-I expression of tumor cells and found that MHC-I was significantly up-regulated in KO group (Figure 7F). Besides, pharmacological inhibition of lysosomes by bafilomycin A1 (BafA1) or chloroquine (CQ) in the WT group could mimic MHC-I up-regulation of the KO group (Figure 7F). In contrast, YTHDF1 depletion failed to show any effect on MHC II expression (Extended data figure 9F).

Figure 7. Tumor-intrinsic YTHDF1 promotes lysosomal proteolysis of MHC-I. (A) Volcano plot of proteomics data for WT (n = 5) and *Ythdf1*-KO (n = 5) B16/F10 cells. Differentially expressed proteins were identified with the threshold of [fold change] > 1.2 and P value < 0.05. (B) KEGG enrichment of proteins down-regulated in *Ythdf1*-KO group. (C) Heatmaps of the lysosome genes' protein level and translation level. (D) Transmission electron microscopy of WT and *Ythdf1*-KO B10/F10 cells. The arrows indicate lysosome. (E) The lysosome level was analyzed by flow cytometry staining of lyso-tracker (PE) and is shown as the mean fluorescence intensity (MFI). (F) Quantitative estimates of cell surface MHC-I levels in WT and *Ythdf1*-KO B16F10 cells and effect of BafA1 (40 nM) or CQ (20 μM) treatment on cell surface MHC-I levels

of B16/F10.

Next, to test the immunogenicity of WT and *Ythdf1*-KO cells, mice were previously immunized with the whole tumor antigens (WTA) generated from WT and *Ythdf1*-KO groups and were then challenged with B16/F10 cells 15 days after priming. Briefly, WT or KO cells were sonicated to release whole tumor antigens. Then, 200 µg of each whole cell lysate was dissolved in 0.1 mL PBS, mixed with 100 µg poly(I:C) vaccine adjuvant, and subcutaneously injected into mice three times, at 7-day intervals. After priming, C57BL/6J mice were challenged with B16/F10 cells and subjected to the downstream *in vivo* analysis (Figure 7G).

G

The results showed that priming with whole tumor antigens from *Ythdf1*-KO group exerts a robust anti-tumor effect as evidenced by reduced tumor volume, improved survival rate, and increased immune cell infiltration (Figure 7H-7L). These indicated that YTHDF1 depletion suppressed tumor antigen degradation, ultimately enhancing tumor recognition and restoring tumor immune surveillance.

Figure 7. Tumor-intrinsic YTHDF1 promotes lysosomal proteolysis of MHC-I. (G) Experimental schedule of whole-tumor antigens (WTA) immunization for B16/F10 tumor model. WT or KO cells were sonicated to release whole tumor antigens. Mice were immunized with 200 μg whole tumor antigen generated from WT (WTA-WT) or KO (WTA-KO) group three times at 7-day intervals. 100 μg poly(I:C) was used as a vaccine adjuvant to enhance antigen presentation. (H) Kaplan–Meier survival curves (n = 10 per group). (I) *In vivo* bioluminescence image of B16/F10 tumors. (J) Tumor growth curves (n = 8 per group). (K, L) Flow cytometric analysis of tumor-infiltrating CD4⁺ and CD8⁺ T cells (n = 5 per group). One of two representative *in vivo* experiments is shown. P values were determined using a two-tailed Student's t test or two-tailed log-rank (Mantel-Cox) test. P values of tumor volumes were evaluated by two-way ANOVA. (NS $P > 0.05$, * $P < 0.05$, ** $P < 0.01$, *** $P < 0.001$, **** $P < 0.0001$).

In T cell-mediated tumor immune surveillance, recognition of MHC class I antigens on the tumor cells by the T cell receptor of CD8⁺ cytotoxic T cells is mandatory for the effector T cells to kill tumor cells. Therefore, characterization of TCR repertoire in tissue presents a promising and highly informative source for predicting and monitoring both host-tumor and antitumor immune conditions. In our analysis, single-cell TCR sequencing determined that CD4 and CD8 TCR clonotype expansion were significantly increased in *Ythdf1*-KO tumors and were mostly distributed in functional CD4⁺ and CD8⁺ T cells (Figure 4J-L and Figure 5H-J). These

also supported that T lymphocytes could efficiently recognize tumors in KO group.

Figure 4 | (J) Bar plot showing Gini index of CD4 TCR diversity. **(K)** Pie chart of CD4 clonotype expansion. **(L)** The t-SNE plot of CD4 TCR distribution.

Figure 5 | (H) Bar plot showing Gini index of TCR diversity. **(I)** Pie chart of CD8⁺ T cell clonotype expansion. **(J)** The t-SNE plot of CD8 TCR distribution.

We have supplemented these results (line 339-404) and corresponding methods in the revision, and discussed them as follows (line 471-494):

One of the well-characterized mechanisms involved in immune evasion is the downregulation or loss of antigen presentation, which confers tumor cells with the ability to become “invisible” and avoid immune attack³¹. CD8⁺ T lymphocytes can recognize processed tumor antigens as small peptides presented by MHC-I molecules. This recognition and activation end with the destruction of the tumors but leaves unharmed MHC/HLA-I negative tumor cells³²⁻³⁴. Clinically, downregulation of tumor MHC-I expression is associated with unfavorable outcomes and resistance to immune checkpoint inhibitors³⁵⁻³⁷. In our study, YTHDF1 deficiency significantly promoted tumor MHC-I expression and subsequently led to an enhanced T-cell recognition as evidenced by the increased infiltration of CD4⁺ and CD8⁺ T cells, and expanded TCR clone type in the tumor microenvironment. Tumors can reduce antigen presentation through several mechanisms including reduced surface expression of MHC-I through genetic alterations, antigen depletion, and degradation. Of notice, recent studies highlight the important role of lysosomal proteolysis in promoting immune evasion of tumor cells by degrading tumor antigen and MHC-I^{30, 38}. Moreover, pharmacological

inhibition of lysosomes could enhance the MHC-I expression and minimize the destruction of internalized antigens³⁰. Here, we demonstrated that tumor-intrinsic YTHDF1 initiates translation of lysosomal genes and promotes proteins expression by increasing the ribosomal loading of m6A-modified mRNA, which is critical for maintaining intact lysosomal function and degrading tumor antigen and MHC-I.

2. With target transcripts identified authors can perhaps continue to explain gene expression changes affected by YTHDF1 through its translational role. And immunosuppressive effects.

Response: Thanks for the suggestion. We supplemented eCLIP and proteomics to identify YTHDF1 targets and reduce false positives. Please refer to our response to comment 1.

Collectively, as depicted in the mechanistic diagram, our findings illustrate that tumor-intrinsic YTHDF1 drives immune evasion via lysosomal degradation of tumor antigen and MHC-I. Starting from the observation that YTHDF1 overexpression is associated with an “immune desert” phenotype and resistance to ICB, we demonstrated that tumor-intrinsic YTHDF1 deficiency inhibited tumorigenesis in immunocompetent mice and that tumor suppression was mainly attributed to the activation of antitumor immunity. Specifically, YTHDF1 deficiency in tumors could alter the immune component of the TME by enhancing the infiltration of CD8⁺ T cells, and CD4⁺ T cells; and activating immunity-related signaling pathways. Mechanistically, YTHDF1 was shown to bind to a panel of lysosomal genes and facilitate its translation. Whereas, YTHDF1 depletion limited lysosomal proteolysis in tumor cells, subsequently enhancing the MHC-I expression and minimizing the destruction of internalized antigens, which ultimately restored tumor immune surveillance and triggered a robust anti-tumor immunity.

3. The single-cell seq of TME is nice but this section is long without focus. Most can be moved to SI. A key question the authors should address here is why and how tumor-intrinsic changes caused by YTHDF1 induces TME changes. It is very hard to think translation of one gene could have such a profound effect.

Response: Thanks for the suggestion. In the single-cell seq of TME, we focused on analyzing CD8 and CD4 T cells, and moved other immune cell populations analyses into supplementary images.

As we responded in comment 1, in this revision, we focus on investigating the altered pathways or a panel of genes regulated by YTHDF1 instead of one single gene. Our results indicate that YTHDF1 promotes a panel lysosomal genes translation, subsequently enhancing the lysosomal degradation of tumor antigen and MHC-I, which ultimately drives immune evasion and resistance to immune checkpoint inhibitors.

4. Ideally, the authors can employ cas13b-FTO or ALKBH5 system to remove m6A on the target transcript and follow either mRNA level or translation changes to establish causal relationship.

Response: Thank you for your professional advice. Indeed, the application of cas13b-FTO or ALKBH5 system is novel and it could be ideal to employ this system to establish a causal relationship. We have tried hard to establish a cas13b-FTO or ALKBH5 system. Unfortunately, due to technical limitations, we failed in this part. Nevertheless, it should be noted that our findings were in line with the previous report that lysosomal genes are the main targets controlled by YTHDF1 in dendritic cells (*Nature*. 2019; PMID: 30728504).

5. Discussion is too long. I would cut at least half and focus on how tumor intrinsic effect of YTHDF1 shapes TME. Note that the previous YTHDF1 DC work also showed a negative correlation of T cell infiltration versus stroma YTHDF1 level in patient samples. Any signaling between tumor and TME that may coordinate expression of YTHDF1?

Response: Thanks for the suggestion. We have streamlined the discussion and focused on how the tumor intrinsic effect of YTHDF1 shapes TME as follows (line 471-494):

One of the well-characterized mechanisms involved in immune evasion is the downregulation or loss of antigen presentation, which confers tumor cells with the ability to become “invisible” and avoid immune attack³¹. CD8⁺ T lymphocytes can recognize processed tumor antigens as small peptides presented by MHC-I molecules. This recognition and activation end with the destruction of the tumors but leaves unharmed MHC/HLA-I negative tumor cells³²⁻³⁴. Clinically, downregulation of tumor MHC-I expression is associated with unfavorable outcomes and resistance to immune checkpoint inhibitors³⁵⁻³⁷. In our study, YTHDF1 deficiency significantly promoted tumor MHC-I expression and subsequently led to an enhanced T-cell recognition as evidenced by the increased infiltration of CD4⁺ and CD8⁺ T cells, and expanded TCR clone type in the tumor microenvironment. Tumors can reduce antigen presentation through several mechanisms including reduced surface expression of MHC-I through genetic alterations, antigen depletion, and degradation. Of notice, recent studies highlight the important role of lysosomal proteolysis in promoting immune evasion of tumor cells by degrading tumor antigen and MHC-I^{30, 38}. Moreover, pharmacological inhibition of lysosomes could enhance the MHC-I expression and minimize the destruction of internalized antigens³⁰. Here, we demonstrated that tumor-intrinsic

YTHDF1 initiates translation of lysosomal genes and promotes proteins expression by increasing the ribosomal loading of m6A-modified mRNA, which is critical for maintaining intact lysosomal function and degrading tumor antigen and MHC-I. Nevertheless, it should be noted that our findings did not analyze the signaling between tumor and TME that may regulate the expression of YTHDF1, which require further exploration *in vitro* or *in vivo*. Our findings should be interpreted with this limitation in mind.

Point-by-point response

Reviewer #2 (Remarks to the Author)

The article presents an impressive series of analyses which convincingly show that tumor intrinsic YTHDF1 is an important regulator of RNA translation efficiency, which in turn drives the tumor's ability to escape the immune checkpoint regulators. The article is well written, of interest for a broad audience, and certainly worth being reported in Nature Communications. I also find commendable that the authors, in addition to looking for a mechanism, strive to develop a strategy to target tumor specific YTHDF1 in order to render the tumors sensitive to ICI. Overall, the article strength lies in the robust characterization of the interplay between the immunological components that act in the presence of high or low tumor intrinsic YTHDF1. As it will be apparent below, my primary concerns regard the connection between the role of YTHDF1 and the activation of Wnt signalling. This part, in my opinion, is relatively weak and should be supported by more convincing data or, alternatively, toned down in its importance.

Response: On behalf of my co-authors, we thank you very much for giving us an opportunity to revise our manuscript. We appreciate you very much for the positive and constructive comments on our manuscript. Those comments are valuable and helpful for revising and improving our paper, as well as the important guiding significance to our future research. We have revised the manuscript extensively according to the comments and supplemented substantial *in vitro* and *in vivo* experiments to further elucidate how tumor-intrinsic YTHDF1 drives immune evasion. We also appreciate the encouraging comments about the manuscript and hope that the correction will meet with approval.

Main points:

1) It is not clear why the authors focus on Pygo1 in the first place. Considering the decrease in mRNA translation upon YTHDF1-KO, 105 target mRNA do not appear neither in the RIP-seq nor in the MeRIP-seq lists. While I agree with the authors that focusing on the 37 that are common in the three lists is the safe choice, assays such as RIP-seq and MeRIP-seq could easily have false negatives. Therefore, I feel it is plausible that several others among the differentially translated mRNA might be real YTHDF1 targets. Are there potentially relevant YTHDF1 targets among these? The need of looking for other contributing factors becomes relevant in particular considering the quite weak – in my opinion – connection with PYGO1 (see comments below).

Response: Thank you so much for the constructive suggestions and opinions. We agree that assays such as RIP-seq and MeRIP-seq could easily have false negatives and looking for other contributing factors is needed. **Therefore, we performed enhanced CLIP-sequence (also suggested by reviewer1) to identify YTHDF1 targets and reduce false negatives.** eCLIP is an enhanced version of the crosslinking and immunoprecipitation (CLIP) assay, and is used to identify the binding sites of RNA binding proteins (RBPs). Because the CLIP and iCLIP methods often result in high duplication rates and low library complexity, eCLIP seeks to improve the efficiency and quality of library production (van Nostrand, et. al, *Robust transcriptome-wide discovery of RNA-binding protein binding sites with enhanced CLIP (eCLIP)*, Nature Methods (2016),13: 508–514). By simplifying the generation of paired IgG and size-matched input controls, eCLIP improves specificity in the discovery of authentic binding sites and maintains single-nucleotide binding resolution. As presented by the binding peaks of YTHDF1, eCLIP showed low signal noise compared to RIP-seq.

We agree with the reviewer that looking for other contributing factors is needed.

As proteins are the central performer of various cellular processes, we initially supplemented proteomics to elucidate further the mechanism regarding how tumor-intrinsic YTHDF1 drives immune evasion and to reveal the alteration of intrinsic regulators and pathways between B16/F10 WT (n=5) and KO (n=5) groups. **In the proteomics analysis, we focus on investigating the altered pathway or a panel of genes instead of one single gene.** Totally, we identified 979 differentially expressed proteins, with 200 up-regulated and 797 down-regulated proteins (Figure 7A, see the figure next page). Noteworthy, most of the differentially expressed proteins were down-regulated in *Ythdf1*-KO group, which fit well with the critical role of YTHDF1 in promoting translation. **Functional enrichment analysis showed that lysosome was the most significantly altered pathway down-regulated in KO group (Figure 7B).** Next, we integrated eCLIP-seq, MeRIP-seq and Ribo-seq to further reveal the regulator function of YTHDF1. Strikingly, most lysosomal genes were identified as YTHDF1-targeted and m6A-marked transcripts followed by attenuated translation (Figure 7C). Given that YTHDF1 is well known to affect mRNA translation in m6A manner, our results indicate that YTHDF1 deficiency impedes lysosomal genes translation, which ultimately reduces protein expression and obstructed the generation of lysosomes. Indeed, flow cytometry and transmission electron microscope further confirmed the attenuated lysosome in the *Ythdf1*-KO group (Figure 7D and 7E).

It is known that lysosomal proteolysis promotes immune evasion of tumor cells by degrading tumor antigen and MHC-I (*Nature* 2020, PMID: 32376951). Here, we hypothesized that *Ythdf1* knockout limited lysosomal proteolysis in tumor cells, ultimately enhancing the MHC-I expression and minimizing the destruction of internalized antigens. **These could increase the immunogenicity of tumor cells and make them become “simply” less fit, and attract the immune armies to restore tumor immune surveillance.** To test this hypothesis, we initially analyzed the surface MHC-I expression of tumor cells and found that MHC-I was significantly up-regulated in KO group (Figure 7F). Besides, pharmacological inhibition of lysosomes by bafilomycin A1 (BafA1) or chloroquine (CQ) in the WT group could mimic MHC-I up-regulation of the KO group (Figure 7F). In contrast, YTHDF1 depletion failed to show any effect on MHC II expression (Extended data figure 9F).

Figure 7. Tumor-intrinsic YTHDF1 promotes lysosomal proteolysis of MHC-I. (A) Volcano plot of proteomics data for WT (n = 5) and *Ythdf1*-KO (n = 5) B16/F10 cells. Differentially expressed proteins were identified with the threshold of $|\text{fold change}| > 1.2$ and P value < 0.05 . (B) KEGG enrichment of proteins down-regulated in *Ythdf1*-KO group. (C) Heatmaps of the lysosomal genes' protein level and translation level. (D) Transmission electron microscopy of WT and *Ythdf1*-KO B10/F10 cells. The arrows indicate lysosome. (E) The lysosome level was analyzed by flow cytometry staining of lyso-tracker (PE) and is shown as the mean fluorescence intensity (MFI). (F) Quantitative estimates of cell surface MHC-I levels in WT and *Ythdf1*-KO B16F10 cells and effect of BafA1 (40 nM) or CQ (20 μM) treatment on cell surface MHC-I levels of B16/F10.

Next, to test the immunogenicity of WT and *Ythdf1*-KO cells, mice were previously immunized with the whole tumor antigens (WTA) generated from WT and *Ythdf1*-KO groups and were then challenged with B16/F10 cells 15 days after priming. Briefly, WT or KO cells were sonicated to release whole tumor antigens. Then, 200 μg of each whole cell lysate was dissolved in 0.1 mL PBS, mixed with 100 μg poly(I:C) vaccine adjuvant, and subcutaneously injected into mice three times, at 7-day intervals. After priming, C57BL/6J mice were challenged with B16/F10 cells and subjected to the downstream *in vivo* analysis (Figure 7G).

The results showed that priming with whole tumor antigens from *Ythdf1*-KO group exerts a robust anti-tumor effect as evidenced by reduced tumor volume, improved survival rate, and increased immune cell infiltration (Figure 7H-7L). These indicated that YTHDF1 depletion suppressed tumor antigen degradation, ultimately enhancing tumor recognition and restoring tumor immune surveillance.

Figure 7. Tumor-intrinsic YTHDF1 promotes lysosomal proteolysis of MHC-I. (G) Experimental schedule of whole-tumor antigens (WTA) immunization for B16/F10 tumor model. WT or KO cells were sonicated to release whole tumor antigens. Mice were immunized with 200 μ g whole tumor antigen generated from WT (WTA-WT) or KO (WTA-KO) group three times at 7-day intervals. 100 μ g poly(I:C) was used as a vaccine adjuvant to enhance antigen presentation. (H) Kaplan–Meier survival curves (n = 8 per group). (I) *In vivo* bioluminescence image of

B16/F10 tumors. **(J)** Tumor growth curves (n = 10 per group). **(K, L)** Flow cytometric analysis of tumor-infiltrating CD4⁺ and CD8⁺ T cells (n = 5 per group). One of two representative *in vivo* experiments is shown. P values were determined using a two-tailed Student's t test or two-tailed log-rank (Mantel-Cox) test. P values of tumor volumes were evaluated by two-way ANOVA. (NS P > 0.05, *P < 0.05, **P < 0.01, ***P < 0.001, ****P < 0.0001).

In T cell-mediated tumor immune surveillance, recognition of MHC class I antigens on the tumor cells by the T cell receptor of CD8⁺ cytotoxic T cells is mandatory for the effector T cells to kill tumor cells. Therefore, characterization of TCR repertoire in tissue presents a promising and highly informative source for predicting and monitoring both host-tumor and antitumor immune conditions. In our analysis, single-cell TCR sequencing determined that CD4 and CD8 TCR clonotype expansion were significantly increased in *Ythdf1*-KO tumors and were mostly distributed in functional CD4⁺ and CD8⁺ T cells (Figure 4J-L and Figure 5H-J). These also supported that T lymphocytes could efficiently recognize tumors in KO group.

Figure 4 | (J) Bar plot showing Gini index of CD4 TCR diversity. **(K)** Pie chart of CD4 clonotype expansion. **(L)** The t-SNE plot of CD4 TCR distribution.

Figure 5 | (H) Bar plot showing Gini index of TCR diversity. **(I)** Pie chart of CD8⁺ T cell clonotype expansion. **(J)** The t-SNE plot of CD8 TCR distribution.

We have supplemented these results (line 339-404) and corresponding methods in the revision, and discussed them as follows (line 471-494):

One of the well-characterized mechanisms involved in immune evasion is the downregulation or loss of antigen presentation, which confers tumor cells with the ability to become “invisible” and avoid immune attack³¹. CD8⁺ T lymphocytes can recognize processed tumor antigens as small peptides presented by MHC-I molecules. This recognition and activation end with the destruction of the tumors but leaves unharmed MHC/HLA-I negative tumor cells³²⁻³⁴. Clinically, downregulation of tumor MHC-I expression is associated with unfavorable outcomes and resistance to immune checkpoint inhibitors³⁵⁻³⁷. In our study, YTHDF1 deficiency significantly promoted tumor MHC-I expression and subsequently led to an enhanced T-cell recognition as evidenced by the increased infiltration of CD4⁺ and CD8⁺ T cells, and expanded TCR clone type in the tumor microenvironment. Tumors can reduce antigen presentation through several mechanisms including reduced surface expression of MHC-I through genetic alterations, antigen depletion, and degradation. Of notice, recent studies highlight the important role of lysosomal proteolysis in promoting immune evasion of tumor cells by degrading tumor antigen and MHC-I^{30, 38}. Moreover, pharmacological inhibition of lysosomes could enhance the MHC-I expression and minimize the destruction of internalized antigens³⁰. Here, we demonstrated that tumor-intrinsic YTHDF1 initiates translation of lysosomal genes and promotes proteins expression by increasing the ribosomal loading of m6A-modified mRNA, which is critical for maintaining intact lysosomal function and degrading tumor antigen and MHC-I.

Collectively, as depicted in the mechanistic diagram, our findings illustrate that tumor-intrinsic YTHDF1 drives immune evasion via lysosomal degradation of tumor antigen and MHC-I. Starting from the observation that YTHDF1 overexpression is associated with an “immune desert” phenotype and resistance to ICB, we demonstrated that tumor-intrinsic YTHDF1 deficiency inhibited tumorigenesis in immunocompetent mice and that tumor suppression was mainly attributed to the activation of antitumor immunity. Specifically, YTHDF1 deficiency in tumors could alter the immune component of the TME by enhancing the infiltration of CD8⁺ T cells, and CD4⁺ T cells; and activating immunity-related signaling pathways. Mechanistically, YTHDF1 was shown to bind to a panel of lysosomal genes and facilitate its translation. Whereas YTHDF1 depletion limited lysosomal proteolysis in tumor cells, subsequently enhancing the MHC-I expression and minimizing the destruction of internalized antigens, which ultimately restored tumor immune surveillance and triggered a robust anti-tumor immunity.

2) The data supporting that PYGO1, in this context, affects beta-catenin nuclear versus cytosolic distribution is not fully convincing. For instance, the loading control of the nuclear fraction, Histone H3, is broadly saturated, and does not allow assessment of the subtle decrease in nuclear beta-catenin protein. Moreover, could the detection of a lower amount of nuclear beta-catenin be the consequence of reduced stabilization of the protein, rather than of its nuclear import? Perhaps, when looking at total beta-catenin, the small decreased is masked by the largely more abundant membranous (as opposed to cytosolic>nuclear) beta-catenin? Moreover, this observation is rendered more cryptic by the fact that, in the same blot, other nuclear proteins (LEF1 and TCF7) are reduced. Is it possible that, rather than nuclear import, YTHDF1 KO causes a more general effect on translation thereby reducing the abundance of many more targets that were identified (see my comment 1)?

Response: Thank you for the comments. We agreed that more YTHDF1 targets should be identified. **As we responded in comment 1, in this revision, we focus on investigating the altered pathway or a panel of genes induced by YTHDF1 instead of one single gene.** This explanation does not require the involvement of the PYGO1 > beta-catenin mechanism. Therefore, we streamlined the manuscript by adding lysosome analysis and removing PYGO1 results (see our response to comment1). Besides, our data did not exclude potential contributions from other targets of YTHDF1; further investigation of complex regulatory pathways mediated by the m6A axis is necessary to expand current knowledge and uncover additional features of antitumor immunity discovered here.

3) It comes as surprise that PYGO1 has such an important role. Several developmental studies clearly attributed a much more important involvement of PYGO2 in Wnt/beta-catenin signalling, where, notably, knockout of Pygo1 has essentially no detectable effect (e.g., PMID: 17425782; PMID: 23637336). I wonder whether the footprints identified on Pygo1 mRNA could also be attributed to Pygo2? Have Pygo2 mRNA and protein relative abundance been measured in this context? Is it possible that PYGO1 and PYGO2 are redundant, and that YTHDF1 stabilizes the mRNA encoding for both the paralog proteins?

Response: Many thanks for providing these references and suggestions. We have reviewed footprints of PYOG2 and performed western blot of PYGO2. And no difference was observed between WT and KO groups (see the figure below).

After a careful literature review, we agreed that PYGO1 may not play such an important role. A look into other YTHDF1 targets is needed. As the evidences we presented in Response 1, in our study, lysosomal degradation of tumor antigen and MHC-I provides more compelling evidence than PYGO1 when explaining how tumor-intrinsic YTHDF1 drives immune evasion.

4) The authors use the Wnt inhibitor LGK974 to confirm the involvement of Wnt/beta-catenin signalling. However, LGK974 inhibits the secretion of all the Wnt ligands (PMID: 24277854) and has therefore broad effects extending to both canonical and non-canonical pathways. While the effect observed is in line with a potential role of the PYGO1>beta-catenin axis, more targeted approaches should be employed to effectively show that PYGO1 is the main player (see comment below).

5) Related to the above, I believe that one key experiment could clearly determine if PYGO1 is the key downstream effector of YTHDF1, that is the assessment of PYGO1-KO in tumor cells. This might be elegantly achieved via exosomes as done for YTHDF1. If the authors' model is correct, PYGO1 depletion should mimic – at least in part – the loss of YTHDF1.

Response: Although the mechanisms may be multiple, lysosomal degradation of tumor antigen and MHC-I provides more compelling evidence than PYGO1 when explaining how tumor-intrinsic YTHDF1 drives immune evasion. As we have provided extensive work, the involvement of PYGO1 may complicate the logic of the manuscript. Further investigation of complex regulatory pathways mediated by the m6A axis is necessary to expand current knowledge and uncover additional features of antitumor immunity discovered here.

6) How could the authors assess the YTHDF1-KO exosomes also do not target several other non-tumor tissues *in vivo*? The strategy is elegant, but it lacks evidence of safety and absent toxicity.

Response: We performed *in vivo* exosome tracing to show the distribution of engineered exosomes at 24 h after intratumoral injection. Briefly, the exosomes were labeled with fluorochrome PKH67. To trace the biodistribution of exosomes *in vivo*, mice received intratumoral injections of 200 µg exosomes in 30 µl PBS, and the fluorescence signal was acquired at 24 h post-injection using an In-Vivo Xtreme imaging system (Bruker). The results showed that the engineered exosomes were mainly absorbed by tumors *in vivo* (Figure 8F, see the figure below). Besides, previous studies revealed that no significant sub-lethal or lethal effects and abnormal development were observed in *Ythdf1*-deficient mice (Nature, 2019, PMID: 30728504;

Nature, 2018, PMID: 30401835). This presents a potential safety strategy to target YTHDF1 *in vivo*.

Figure 8F | *In vivo* exosome tracing at 24 h after intratumoral injection of 200 μg fluorescently labeled *Ythdf1*-KO exosomes

To further evaluate the safety of engineered exosomes, we recorded the body weights of the mice. Moreover, we performed blood biochemistry and HE staining of liver, lung and kidney to evaluate the potential hepatotoxicity, nephrotoxicity and myocardial toxicity of *Ythdf1*-KO exosomes. As shown in Extended data figure 10, there were no differences in mouse body weight (Extended data figure 10A), morphology of lung, liver, and kidney (Extended data figure 10B), renal function, liver function, and myo-cardial enzymonram (Extended data figure 10C) between the vector and KO exosome groups.

Extended data figure 10. Toxicity test of engineered exosomes treatment. (A) Body weight. (B) HE staining of liver, lung, and kidney tissue. Scale bar, 200 μm (C) Evaluation of blood biochemistry, including albumin (ALB), alkaline phosphatase (ALP), aspartate aminotransferase (AST), total bilirubin (TBIL), direct bilirubin (DBIL), total bile acid (TBA), creatinine (Crea), Urea, cholesterol (CHOL), high density lipoprotein cholesterol (HDL-C), low density lipoprotein cholesterol (LDL-C), triglyceride (TG), creatine kinase (CK), Creatine kinase MB isoenzyme (CK-MB), lactic dehydrogenase (LDH), glucose (GLU), and glycated serum protein (GSP). P values were determined using a two-tailed Student's t test (NS $P > 0.05$).

7) One of the key points (e.g., from Figure 3) is that YTHDF1-KO in tumors leads to activation of immunological processes and pathways in immunocompetent mice, which in turn recruits more tumor-infiltrating CD4+ and CD8+ T cells. Could this occur because, upon YTHDF1-KO, tumor cells become “simply” less fit and/or senescent-like, and attract the immune armies as a consequence of this? The authors themselves recognize that this effect results from the enhanced expression of cytotoxic molecules upon YTHDF1 KO and the subsequent activation of immunological pathways. This explanation might not require the involvement of the PYGO1 > beta-catenin mechanism suggested. How do the authors reconcile these two alternative mechanisms?

Response: Thank you for enlightening us with the idea that upon YTHDF1-KO, tumor cells become “simply” less fit and/or senescent-like, and attract the immune armies as a consequence of this. Indeed, our supplemented results indicated that YTHDF1 depletion limited lysosomal proteolysis in tumor cells, subsequently enhancing the MHC-I expression and minimizing the destruction of internalized antigens, which ultimately improve tumor immune surveillance and triggered a robust anti-tumor immunity. This explanation does not require the involvement of the PYGO1 > beta-catenin mechanism suggested.

We also explored whether tumor cells become senescent-like by performing β -galactosidase staining. However, no difference is observed.

8) In Figure 6 the authors show that YTHDF1 deficiency increases PD-L1 expression in vivo but not in vitro. First, it is not clear why there is this difference between the two contexts. Second, shouldn't PD-L1 increase induce a protective role in tumor cells in the absence of ICI treatment?

Response: In the tumor microenvironment, cancer cells are threatened with

immunosurveillance by both innate and adaptive immunity. Abundant inflammatory cytokines exist in this region and orchestrate the balance of anti-tumor immunity. However, cancer cells also hijack the inflammatory pathways (adaptive signaling pathways) to enhance PD-L1 expression, a self-protective mechanism against the immune system [1,2]. Previous studies have proved that inflammatory cytokines (e.g., IFN-g, TNF-a, TLR3, IL-4/6/17/27) in the tumor microenvironment could promote PD-L1 expression of tumor cells [3-7].

In our study, YTHDF1 deficiency in tumors could enhance the infiltration of NK cells, CD8⁺ T cells, and CD4⁺ T cells, and promote inflammatory cytokines expression (e.g., IFN-g, IL-4, IL-6) in the tumor microenvironment. These contribute to the up-regulation of PD-L1 in the tumor cells. It is, therefore, reasonable to synergize with ICI therapy to elicit robust antitumor effects. Meanwhile, YTHDF1 was not able to regulate PD-L1 expression under *in vitro* conditions due to the absence of inflammatory cytokines.

References

1. Cha, J.-H., Chan, L.-C., Li, C.-W., Hsu, J.L. & Hung, M.-C. Mechanisms Controlling PD-L1 Expression in Cancer. *Mol Cell* 76, 359-370 (2019).
2. Dong, H. et al. Tumor-associated B7-H1 promotes T-cell apoptosis: a potential mechanism of immune evasion. *Nature Medicine* 8, 793-800 (2002).
3. Garcia-Diaz, A. et al. Interferon Receptor Signaling Pathways Regulating PD-L1 and PD-L2 Expression. *Cell Rep* 19, 1189-1201 (2017).
4. Quandt, D., Jasinski-Bergner, S., Müller, U., Schulze, B. & Seliger, B. Synergistic effects of IL-4 and TNF α on the induction of B7-H1 in renal cell carcinoma cells inhibiting allogeneic T cell proliferation. *J Transl Med* 12, 151 (2014).
5. Wang, X. et al. Inflammatory cytokines IL-17 and TNF- α up-regulate PD-L1 expression in human prostate and colon cancer cells. *Immunol Lett* 184 (2017).
6. Carbotti, G. et al. IL-27 induces the expression of IDO and PD-L1 in human cancer cells. *Oncotarget* 6, 43267-43280 (2015).
7. Zhang, N. et al. The EGFR pathway is involved in the regulation of PD-L1 expression via the IL-6/JAK/STAT3 signaling pathway in EGFR-mutated non-small cell lung cancer. *Int J Oncol* 49, 1360-1368 (2016).

Other minor points:

9) The authors should consider being consistent with the gene/mRNA/protein nomenclature, as sometime (e.g., in the abstract), they refer to YTHDF1 or Ythdf1.

Response: Thanks for the suggestion. We have referred to the Guidelines for

Formatting Gene and Protein Names.

For Mice, gene symbols are italicized, with only the first letter in upper-case (e.g., *Afp*). Protein symbols are not italicized, and all letters are in upper-case (e.g., AFP).

For Humans, gene symbols contain three to six italicized characters that are all in upper-case (e.g., *AFP*). Gene symbols may be a combination of letters and Arabic numerals (e.g., 1, 2, 3), but should always begin with a letter; they generally do not contain Roman numerals (e.g., I, II, III), Greek letters (e.g., α , β , γ), or punctuation. Protein symbols are identical to their corresponding gene symbols except that they are not italicized (e.g., AFP).

We have unified the name with YTHDF1 except for describing mouse gene with *Ythdf1*.

10) I believe the authors should explain better how they have developed the ROC curve (lines 99-101) to make the reader understand how reliable it is.

Response: Thank you for the suggestion. We have supplemented the method of ROC curve analysis in the revision as follows (line 1058-1065):

ROC curve analysis

The ROC curve was applied to evaluate the predictive efficiency of YTHDF1 expression for the anti-PD-1 response rate. The YTHDF1 expressions of response and non-response groups were input and the curve was generated and visualized using pROC package and ggplot2 package in R software. The area under the curve was calculated as a single measure to discriminate efficacy. When the ROC-curve produced an area under the curve above 0.7, a cut-off value was determined with high specificity and positive predictive value.

11) Line 110-111: correct in “was resistant to ICI therapy.”

Response: We have corrected this description in the manuscript.

12) Line 111: “We utilized the B16/F10 mouse model.” could the authors explain this better? B16/F10 is a mouse-derived cell line.

Response: Sorry for the vague description. In our study, mice were inoculated subcutaneously with B16/F10 cells to establish a melanoma model. We have corrected this description as follows:

“We utilized the B16/F10 melanoma model”

13) The authors write “subgroup analysis revealed the significant prognostic value of YTHDF1 in the TGFB1-low group but not the TGFB1-high group, which indicated that the prognostic impact of YTHDF1 is associated with immunity”. It is not clear to me what the relation is between TGFB signalling in melanoma and the conclusion of involvement of the immune system.

Response: The subgroup analysis of TGFB1-low and -high was enlightened by a previously published paper (*Nature Medicine* 2018; PMID: 30127393). In this study, the authors analyzed the prognostic value of cytotoxic T lymphocyte score in the TCGA melanoma study and demonstrated that a higher CTL score indicates a better patient survival, but only when TGFB1 has a low expression level (see the figure below).

This observation corroborates the known role of the cytokine TGF β (encoded by TGFB1) in promoting tumor immunosuppressive microenvironment and immunotherapy resistance. CTL score (associated with anti-immunity) fails to impact prognosis under an immunosuppressive microenvironment (TGFB1 high).

Similarly, our subgroup analysis revealed the significant prognostic value of YTHDF1 in the TGFB1-low group but not in the TGFB1-high group. This may indicate YTHDF1-low contributes to favor prognosis via anti-immunity under a non-immunosuppressive microenvironment (TGFB1 low). This prognostic effect was blocked under an immunosuppressive microenvironment (TGFB1 high). Nevertheless, it should be noted that our subsequent biological validation of YTHDF1 further supports this explanation. We also rephrased our description as follows (line 122):

“which indicated that the prognostic impact of YTHDF1 is potentially associated with immunity”

14) Lines 258-260, the authors write “we determined that CD4 TCR diversity and clonotype expansion were increased in *Ythdf1*-KO tumors (Figure 4J and 4K), and were mostly distributed in functional CD4⁺ T cells rather than naive cells (Figure 4L), suggesting tumor specificity”. It is not clear why the authors could conclude “tumor specificity” – I suggest rephrasing or provide a better explanation to this statement.

Response: Thank you for highlighting this issue. The T cell receptor (TCR) endows T cells with antigen specificity and is central to nearly all aspects of T cell function. In this way, the TCR serves as a molecular barcode that tracks processes such as migration, differentiation, and proliferation of T cells. Recent technological advances have enabled sequencing of the TCR from single cells alongside deep molecular phenotypes on an unprecedented scale. The single cell TCR-seq (scTCR-seq) has been useful to inform immune response following PD-1-based immunotherapy in cancer patients. We have rephrased the description as follows (line 257):

“We determined that CD4 TCR clonotype expansion were increased in *Ythdf1*-KO tumors (Figure 4J and 4K), and were mostly distributed in functional CD4⁺ T cells rather than naive cells (Figure 4L), **indicative of ongoing immune responses.**”

15) In Figure 4B, left panel, it is not clear how the cell clusters should be interpreted: please provide a better explanation on the difference between the number-indicated cell clusters and cell type derivation. For example, it seems that a uniform cell population of Mono/Macro is deduced from clusters 4, 0 and 3. How is this possible?

Response: Thanks for the question. In our analysis, the number-indicated cell clusters were determined by unsupervised hierarchical clustering analysis and the specific cell population was further determined by the marker gene expression. In detail, PCA was constructed based on the scaled data with the top 2000 high variable genes and the top 10 principals were used for tSNE construction. Utilizing the graph-based cluster method, we acquired the unsupervised cell cluster result based on the PCA top 10 principal and we calculated the marker genes by FindAllMarkers function with wilcox rank sum test algorithm. As shown by the heatmap (see the figure next page), the cell clusters presented distinct expression patterns.

Although defined by unsupervised clustering, clusters 4, 0 and 3 showed similar expression patterns with certain modules (as indicated by red rectangle in the heatmap) and expressed a panel of classic Mono/Macro genes (see the figure below).

Point-by-point response

Reviewer #3 (Remarks to the Author)

In the manuscript “Tumor-intrinsic YTHDF1 drives immune evasion and resistance to immune 2 checkpoint inhibitors by regulating the Pygo1/ β -catenin axis” the authors describe how YTHDF1 effects in tumor cells regulate the immune response in the tumor microenvironment.

Response: We would like to express our great appreciation to you for the in-depth reading of the manuscript and for giving constructive suggestions which are valuable for improving our paper. Those comments especially on scRNA-seq section are expertised and insightful. We learned a lot from the comments and tried our best to revise the manuscript according to the comments.

1. In figure 3B, the authors should check the labels, as the data shows in the KO tumor tissue, there is actually negative enrichment of immune related pathways, which is in opposition to fig 3A and their conclusions.

Response: Sorry for the typing error. We have corrected this mislabeling of figure 3B in the revision (see the figure below).

2. Lines 207-211: “Overall, the proportions and cell numbers for different cell categories exhibited apparent differences between the KO and WT groups (Figure 4C). Of note, T cells, NK cells, and proliferating mixed immune cells were enriched in Ythdf1-deficient tumors, whereas neutrophils and B cells showed higher infiltration in WT tumors (Figure 4D).” Can the authors include some statistical analysis to back-up their claim? I believe that is probably true for T cells and NK cells. Regarding conclusions about these analysis, it is worth mentioning the reduction in B cells needs to be understood in line of higher infiltration of other cells in the KO samples. So the likely reason for the proportional reduction of B cells is simply because there are more of other cells. The authors should also elaborate on the increase of tumor cells (is this technical? Are they similar?) Maybe exclusion of the tumor cells from this analysis would also be necessary, to keep proportional comparisons adequate.

Response: In this version, we showed the numbers of each immune cell subpopulation in Figure 4D (see the figure below). We can see that in addition to the proportional reduction, the cell number of B cells and neutrophils show a robust decrease in the KO group.

For the single-cell sequence, CD45⁺ immune cells were initially isolated by FACS sorting. The gating strategy of FACS sorting was supplemented in extended data Figure 11 (see the figure below). **A small portion of tumor cells is probably a technical limitation caused by FACS sorting of CD45⁺ cells from tumor tissue. We have excluded the tumor cells from this analysis.**

3. Lines 239-242: “By pseudotime analysis, proliferating CD4⁺ T cells were located at the beginning of the pseudotime trajectory, while naive CD4⁺ T cells were located in the terminally differentiated state of the branch, with TEMs, Tregs, and IFN-responsive CD4⁺ T cells being transition states spread along the axis (Extended data figure 5D-E).” The analysis of the pseudotime is very open for misinterpretations (with labeling naïve cells as terminally differentiated). Additionally, as expected, cycling cells drive a large part of the observed effects and complicates rather than helps in the data interpretation. I would suggest exclude the proliferating cells (ideally) or perform some cell cycle correction then repeat the trajectory analysis, and interpret those results under current understanding of T cell differentiation from naïve to other subsets.

Response: We highly appreciate this professional comment. This suggestion is expertised and helpful. We didn't realize the proliferating cells in pseudotime analysis could complicate data interpretation and induce misinterpretations. Indeed, as you suggested, we excluded the proliferating cells and repeated pseudotime analysis. **The results showed that naive CD4⁺ T cells were located at the beginning of the pseudotime trajectory, while Tregs were located in the terminally differentiated state of the branch**, with TEMs and IFN-responsive CD4⁺ T cells being transition states spread along the axis (see the figure below). We have revised the Extended data figure 5D-F and the corresponding results in the manuscript.

4. Figure 5E should be excluded. Direct comparison of cell numbers in single cell between conditions without any normalization to the actual number of collected cells is not scientifically sound.

Response: Thanks for the suggestion. We have excluded Figure 5E in the revision.

5. The results from figure 5 are largely driven by a technical/biological aspect of the data, as seen in extended data. The high presence of specific alpha and beta chains are driving the cluster assignments. Even though they are “true biological differences” the capture of TCR alpha and beta chains without specific enrichment protocols is heavily biased. So I would suggest exclusion or correction for all the detected alpha and beta chains in the CD8 T cells. Then repeat of the analysis displayed in this figure.

Response: We appreciate this excellent suggestion. To eliminate this bias, we excluded all detected TCR alpha and beta chains genes and reperformed CD8 subpopulation analysis including Fig.5A, B, C, D, I, and extended data Fig 6A, B, C D.

Overall, the results are in line with our previous analysis. We have supplemented these analyses in the revision making the analysis sufficiently rigorous. Of note, these changes did not alter our previous conclusion.

6. Figure 5G displays a very nice cytotoxic effect of CD8 T cells. Could the authors repeat the same experiment but using CD4 T cells? They identified populations with high granzyme levels that they claim to not be immunosuppressive, so some data to support that would be appreciated.

Response: Thank you for highlighting this issue. We repeated the same experiment using CD4 T cells. Unfortunately, we didn't observe positive results. It may be due to the fact that the direct cytotoxic effect of CD4 T cells requires peptide-MHC II recognition of tumor cells. However, the majority of cancers lack intrinsic MHC II expression, which limits the cytotoxic effect of CD4 T cells
(Curr Opin Immunol. 2022 Feb; 74: 18–24).

Editorial Note: Figure redacted due to its Creative Commons license.

In our study, both WT and KO B16/F10 cells barely expressed MHC-II. On the contrary to MHC-II, MHC-I was detected in B16/F10 cells and can be further upregulated by *Ythdf1* knockout, which confers CD8 T cells the ability of tumor-cell killing.

Nevertheless, a major role of CD4⁺ T cells is the provision of help for anti-tumour CTLs through both direct and indirect mechanisms. Activated CD4⁺ T cells secrete various cytokines, which directly activate CD8⁺ CTLs by driving their effector function, differentiation, and proliferation.

7. Neutrophils are not always easy to capture in droplet based single cell methods. Can the authors provide some evidence why these are indeed neutrophils and not some monocyte populations?

Response: Thank you for the question. The definition of mouse neutrophils referred to a previously published paper (Cell, 2020; PMID: 32302573). The representative genes of neutrophils and monocyte were listed below.

Mouse myeloid cell cluster names	Functional properties	Representative genes		aCD40	aCSF1R	
		TFs	Other genes	D2	D10	R
mM01_Mast-Cpa3	Mast	Gata2, Nfil2, Klf7	Cpa3, Mela2, Il1fm1, Hdc, Slc41a2, Csf1, Syt13, Hgf, Il6, Il4, Ccl3, Cdh1, Cxcr2, Cd9, Cd60, Il18ap	NA	NA	NA
mM02_Neutrophil-Csf3r	Neutrophil	Ets2, Mxd1, Klf2, Egr1, Csmpl1	Csf3r, Cxcr2, Cd24a, C5ar1, Trem1, Il1r2, S100a8, S100a9, Cxcl2, Il1rn, Clec4d, Clec4e, Upp1	↑	NA	↑
mM03_pDC-Siglech	pDC	Tcf4, Runx2, Bcl11a, Tsc22d1, Ilf7	Siglech, Ly6d, Cox6a2, Smim5, Klk1, Ppgrp1, Lety1, Ubp1, Ccr9, Cd7, Sell, Cd164, Pltp, P2ry14, Ptpns	NA	NA	NA
mM04_cDC2-cd209a	cDC2	Bcl3, Bhlhe40, Cbfa2t3	Cd209a, Cd300a, Cd83, Il7b7, Vrk1, Napsa, Ms4a4c	↓ ↓	→	↑ ↑
mM05_cDC2-Ilgax	cDC2		Ilgax, Cd300a, Adrbk2, Il4i1, Mccmp1, Tmem176a, Slc38a2, Ramp1	↓	NA	NA
mM06_cDC1-Clec9a	cDC1	Batf3, Cbfa2t3, Pa2g4	Xor1, Il7b7, Arsb, Ckb, Fgd2, Naga, Pak1, Rab7b, Wotj4, Ppm1m, Cd24a, Flt3, Cd83, Slamf7, Slamf8	↓	NA	↑
mM07_cDC1-Ccl22	cDC1	Relb, Etv3, Batf3, Aebp2, Nkfb2	Ccl22, Ccl5, Il15, Ccr7, Il15ra, Plxnc1, Pmp, Cd40, Birc2, Fcscr1, Anxa3, Cacnb3, Nudt17, Socs2, Tspan9, Serpinb6b	↑ ↑	NA	↑ ↑
mM08_Mono-Ly6c2	Monocyte	Stat2, Mxd1, Bach1, Bcl3	Ly6c2, Mgst1, Ccl9, Smox, Il13, Il205, Ilm2, Isg20, Plaur, Tlr2, Cd14, Cd300k, Tgm2, Cxcl10, Fnl1, F13a1, Ccl2	↓ ↓ ↓ ↓	↓	↑
mM09_Mono-Nr4a1		Nr4a1, Rara, Nr1h3, Nr3, Klf4, Bcl3	Cd300a, Il17ra, Tnfrsf21, Tnfrsf13, Cd302, Cd14, Il1tm6, Gstm1, Idh1, Gsr, Adgr5, Fnl1, F13a1, Anxa1, Ramp1, C3	↓ ↓ ↓	↑	NA
mM10_Mono-Ilgal		Nr4a1, Pou2f2, Klf4, Lyf1	Ilgal, Ace, Cd300a, Ceacam1, Spn, Tnfrsf1b, Il17ra, Lrp1, Adgr4, Trem14, Ear2, Skt10	↑	↑	NA
mM11_Macro-Mafb	Macrophage	Mafb, Cebpb	C1qa/b, Axl, Spint1, C3ar1, Fhl1, Ccr5, Ccr1, Fogr1, Fogr4, Ly6l, Cfb, Mmp14, Clec5a, Ier3	↓ ↓ ↓ ↓	↓	↑
mM12_Macro-Maf		Maf	C1qa/b/c, Axl, Ccl12, Trem2, Tgfb1, Cd81, Cd72, Cd63, Abhd12, Adgre1, Ms4a7, Nltpc, Olfml3, Tmem119, Hpgds	NA	↓ ↓ ↓	↓ ↓ ↓
mM13_Macro-Ccl12		Ifi7, Mafb	C1qa/b/c, Ccl12, Sdc3, Lgmn, H2-Aa	↑	↓ ↓	→
mM14_Macro-Mgl2		Klf4	Mgl2, Lrp1, Fnl1, Pltp, Axl, Ear2, Tnfp3, Birc5	→	↓ ↓	↓ ↓
mM15_Macro-Vegfa		Cebpb, Bhlhe40, Aif4, Tgfb1, Aif2	Spp1, Vegfa, Mmp12, Adam8, Cd274, Cd63, Thbs1, C3ar1, Il1rn, Clec4d, Emp1, Arg1, Erc1l, Hlpa, Hnox1, Sgk1	NA	NA	↑ ↑

In our study, cluster 6 expressed a high level of neutrophil markers (*Csf3r*, *Cxcr2*, *S100a8*, and *S100a9*) but a low level of monocyte markers (*Ccl2*, *Ly6c2*, *Fnl1*, and *F13a1*), and is therefore defined as neutrophils. We have supplemented this evidence in the extended data Figure 7A.

A

Point-by-point response

Reviewer #4 (Remarks to the Author)

This study shows a very interesting topic of the role of RNA methylation and in particular YTHDF1 in regulating immune cell infiltration into tumors and a novel approach to disrupt YTHDF1 dependent immune evasion using exosomes. To my knowledge, this work is original with significant results but the authors should provide more analysis of the exosome mediated gene editing. I have a few major points I would like to be addressed and multiple minor points below:

Response: We thank you so much for your insightful peer review. Those comments, especially regarding exosomes, are valuable and helpful for revising and improving our paper. We have studied the comments carefully and supplemented the required experiments in the revision. We also highly appreciate the encouraging comments about the manuscript and hope that the correction will meet with approval.

1. Extended Data Figure 3 and the legend are lacking in important details. In the figure Panel J on the left there is no indication about what the different subpanels are, I can imagine the left blue stain is Dapi, the middle is EdU, and the right is merged but this is missing from the figure or the legend. Also I am assuming results comparing wild-type (WT) and knock-out (KO) cells are using WT cells that have not been Puro selected, but this should be described in the legend. Lastly, in Panel C there is a very useful overview of the workflow but I believe it misses the important step of Puromycin selection before the single clone cell screening. Lastly there is a typo on the panel B plasmid map, it reads “spCase9” but should be corrected to “spCas9”

Response: Thank you for the attention to detail. As you indicate, the left blue stain is Dapi, the middle is EdU, and the right is merged. We have labeled this information in Extended Data Figure 3 J (see figure below) and supplemented it in the legend (line1305).

Extended data figure 3. (J) Visualization of DNA replication by EdU incorporation. Blue = DAPI, and red = EdU.

The missing step of Puromycin selection has been added in Extended Data Figure 3 C (see figure below). Besides, WT cells were exposed to the same procedure as KO cells except for transfection with the scramble plasmid. We have supplemented it in the legend (line 1301).

Extended data figure 3. (C) Workflow for the generation of *Ythdf1* knockout cell clones. WT cells were exposed to the same procedure as KO cells except for transfection with the scramble plasmid.

Thank you for reminding. We have corrected this typo in Extended Data Figure 3 B (see figure below).

2. In Figures 6C and 6E it is extremely surprising that only some of the comparisons gave statistical significance but not others, did the authors omit statistical comparison among some groups such as WT+IgG vs KO+IgG, KO+CTLA4, or KO+PDL1 which all show no significance? For Figure 6C how can it be explained that WT+CTLA4 and WT+PDL1 have lower tumor volumes with comparable standard deviations to WT+IgG yet only the WT+CTLA4 and WT+PDL1 have statistically significant

difference to KO conditions but not the WT+ IgG? Likewise, for Figure 6E how can it be explained that WT+CTLA4 and WT+PDL1 have better survival than WT+IgG yet only the WT+CTLA4 and WT+PDL1 have statistically significant difference to KO conditions but not the WT+ IgG? If only some comparisons were done this should be specified in the figure legend or better if all statistical comparisons are included in the figure.

Response: Thank you for the comments. In figure 2, we have demonstrated a significant tumor growth inhibition in KO group versus WT group. **In this section, we aimed to investigate whether immune checkpoint inhibitors (PD1 and CTLA-4) could amplify the anti-tumor effect induced by Ythdf1 knockout.** To make the results concise, we did not provide all statistical comparisons but focused on comparing tumor growth in KO+IgG group versus KO+CTLA4, or KO+PDL1 group. If we provide all statistical comparisons among the six groups, there will be 14 different comparisons. To make the results more detailed and streamlined, we supplemented some important statistical comparisons, as you suggested, in the figure 6C legends of the revision (see the legend below).

Figure 6. Tumor-intrinsic YTHDF1 deficiency enhances ICI therapy responses *in vivo*. (C) Tumor growth curves of WT and KO tumors. Mice were intraperitoneally treated with 200 μ g anti-PD-L1 or 200 μ g anti-CTLA4 on days 7, 9 and 11 after tumor inoculation. A rat immunoglobulin G (IgG) isotype antibody was applied as a control. The statistical comparisons among these group were listed as follows: WT+IgG vs KO+IgG, ***P < 0.001; WT+IgG vs WT+PD-L1, **P < 0.01; WT+IgG vs WT+CTLA4, ***P < 0.001; WT+IgG vs KO+PD-L1, ***P < 0.001; WT+IgG vs KO+CTLA4, ***P < 0.001; KO+IgG vs KO+PD-L1, *P < 0.05; KO+IgG vs KO+CTLA4, *P < 0.05; WT+CTLA4 vs KO+CTLA4, *P < 0.05; WT+PD-L1 vs KO+PD-L1, *P < 0.05.

3. Figure 9D is not adequate to conclude that “typical exosomal structures” are observed as stated by the authors. One problem is that few or only 1 structure is observed per panel, ideally a wide field of view with more vesicles should be used and if desired a single vesicle can be highlighted with an insert zoom. Another problem is that the highlighted structures are closer to 150-200nm in size despite the nanoparticle flow cytometry detecting what appears to be a modal size of 50nm and author’s reported mean size of 65-70nm. Please repeat the TEM measurements with higher concentration of exosomes to visualize multiple particles per field of view. Here is an article with a detailed protocol and see Figure 3.22.2 as good example of what should be aimed for, to have multiple particles in the field of view: DOI: 10.1002/0471143030.cb0322s30

Response: As you suggested, we concentrated exosomes in the TEM measurements and provided a view of multiple particles per field (see the figure below). The

exosome size distribution in TEM is in accord with the nanoparticle flow cytometry.

D

4. There is much evidence that exosome tracking with lipid-anchored fluorescent dyes can lead to artefacts such as dye aggregates or lipoprotein <https://doi.org/10.1080/20013078.2019.1582237>. In the methods section no dye removal step is mentioned after staining so if it hasn't been done the exosomes should be recentrifuged after staining to pellet and then resuspend in fresh buffer to remove some of the unbound dye. Furthermore, the authors should include at least for the *in vitro* uptake a minimal control experiment where a “mock EV” condition, consisting of fresh media that was not treated with cells, has been exposed to the same procedure for exosomes isolation and fluorescent labelling, and used to observed the uptake of any dye aggregates this procedure may form. Authors may also try to use previously described strategies of labelling using fluorescent protein fusion constructs that are recognised to be more robust for exosome tracking experiments doi: 10.1038/ncomms8029.

Response: In our previous analysis, the dye removal step has been down in the exosome labeling step, and we are sorry for the missing description in the method section. We have supplemented the method of exosome labeling and dye removal step in the revision as follows (line 619-626):

Exosome labeling and *in vivo* live imaging

Briefly, exosomes were labeled with PKH67 (10 μ M; Sigma, catalog no. MINI67) following the manufacturer's procedures. **After staining, the exosomes were ultracentrifuged to remove the unbound dye and were washed with PBS to remove the residual dye further.** The cellular uptake and intracellular distribution of exosomes were determined with a Zeiss LSM510 confocal microscope (ZEISS). **The cell contours were imaged by staining with phalloidin (1:1000; Abcam, ab176759). The media that was not exposed to the tumor and received the same exosome isolation and fluorescent labeling procedure was used as mock control.** To trace the biodistribution of exosomes *in vivo*, mice received intratumoral injections of 200 μ g exosomes in 30 μ l PBS, and the fluorescence signal was acquired at 24 h post-injection using an In-Vivo Xtreme imaging system (Bruker).

We also thank you for your professional suggestion about the “mock EV” control

experiment. We have supplemented it as you suggested in the revision.

Phalloidin/Exosome/DAPI

Mock control

Exosome

Figure 9 | (E) Exosome uptake assay by confocal microscopy analyses of B16/F10 cells. Exosomes were fluorescently labeled using PKH67 (green) and incubated with B16/F10 cells for 24 h. The cell contours were imaged by staining with phalloidin (purple). The media that was not exposed to the tumor and received the same exosome isolation and fluorescent labeling procedure was used as mock control. The fluorescence signal was imaged by confocal microscopy. Scale bar, 20 μ m.

5. While a very convenient method Exo-Fect transfection of exosomes requires important experimental controls to ensure that the observed cargo transfer is happening by exosomes and not potential residual Exo-Fect and cargo left-over being transferred. To exclude the possibility that the DNA plasmid is being transferred outside exosomes, the authors should include a DNase I treatment step of the Exo-Fect transfected exosomes and they should also include a similar “mock EV” condition described above where the buffer without exosomes is transfected with Exo-Fect. Ideally these two control conditions can be tested by injection intratumorally in mice, or at an absolute minimum they should be tested *in vitro* on B16F10 cells followed by YTHDF1 and PYGO1 WB. These two conditions would definitively prove that the exosomes are the carriers of the DNA cargo. Furthermore, authors should include more discussion on the use of exosomes for gene editing, comparing their approach of DNA plasmid transfer via exosomes to other works transferring CRISPR Cas9 ribonucleoprotein with EVs such as <https://doi.org/10.1002/jev2.12225>. Are there any potential advantages or disadvantages in a DNA plasmid vs ribonucleotide transfer approach?

Response: We highly appreciate the professional suggestion about including DNase I treatment and mock control. As you suggested, we repeated the *in vivo* experiment of Figure 9G, 9H, 9I, and 9J by including a DNase I treatment step of the Exo-Fect

transfected exosomes and supplementing the “mock EV” control group in the revision (see the figure below).

We have revised the corresponding methods as follows (line 610-616):

Exosome loading using plasmid DNA and *in vivo* exosome treatments

The Exo-Fect Exosome Transfection Kit (System Biosciences, catalog no. EXFT20A-1) was employed to load cargo into exosomes in accordance with the recommendations of the manufacturer. Briefly, 5 μ g plasmid DNA (*Ythdf1* sgRNA-spCas9 plasmid or control sgRNA-spCas9 plasmid), 10 μ l Exo-Fect solution, 200 μ g exosomes, and sterile PBS were mixed in a 150 μ l total transfection solution and subjected to 10 minutes of incubation at 37°C in a shaker. Thirty microliters of ExoQuick-TC reagent were introduced into the transfected exosomal sample suspension with gentle shaking and incubated at 4°C for 30 minutes to terminate the reaction. The sample was further centrifuged in a microfuge for 3 minutes at 13000-14000 rpm. Next, the transfected exosomes were treated with DNase I (0.15 units/ μ L, Sigma-Aldrich) to exclude the residual plasmid DNA, and collected by ultracentrifugation at $100000 \times g$ for 70 minutes. The media without exosomes received the same plasmid DNA transfection and DNase I treatment procedure was used as mock control.

We supplemented the discussion of potential advantages or disadvantages in a DNA plasmid vs ribonucleotide transfer approach in the revision as follows (line 509-521):

There are three options for delivery CRISPR system, including plasmid DNA encoding Cas9 and sgRNA, Cas9 mRNA plus sgRNA, and Cas9 ribonucleoprotein (RNP) complexed with sgRNA. Delivery of CRISPR system by plasmid DNA presents a convenient approach and has been applied by several studies^{45, 46}.

Noteworthy, the introduction of CRISPR components as plasmids is also associated with potential off-target effects. Compared to DNA and RNA delivery, RNP delivery avoids many pitfalls allowing for fast delivery and weak off-target effects^{47, 48}. However, the therapeutic delivery of RNPs is currently hampered by the large size of RNPs and lacks resistant gene for the drug selection. In our study, we initially created engineered exosomes that transported CRISPR/Cas9 plasmid DNA for in vivo targeting of carcinogenic Ythdf1. This presents a safe approach to suppressing tumor growth by restoring tumor immune surveillance. Despite this, it should be noted that further optimization of the CRISPR system delivery is needed in our further study.

Minor points:

6. FTO is found throughout the manuscript but I think it would be useful on the first instance in the introduction to rewrite as follows: “Tumors exploit the fat mass and obesity-associated protein (FTO)-mediated regulation of glycolytic metabolism to evade immune surveillance¹³”.

Response: Thank you for your reminder. We have used “fat mass and obesity-associated protein” when first referring to FTO (line 36).

7. Figure 1 legend should be corrected as follows: (E, F) Box plots showing the differential expression of YTHDF1 in progressive disease (PD), stable disease (SD), partial response (PR), and complete response (CR) samples using RNA sequencing data from the (D) Hugo2016_PD1_Melanoma cohort (n = 18) and Riaz2016_PD1_Melanoma cohort (n = 56).

Response: Many thanks. We have revised the legend of Figure 1E in the revision (line 1104).

8. On page 4 there is a mixup about the extended figure 2. Authors refer to this being the CRISPR KO design but in fact it should be extended figure 3. Please correct.

Response: Sorry for this typing error. We have corrected it in the revision (line 138).

9. It appears various schematics were made using biorender.com, if so it should be included in the methods somewhere.

Response: Yes, schematics were generated using BioRender. We have included it in the Acknowledgment as suggested by Biorender team.

“The schematics were created with BioRender.com.”

10. Figure 5B and the manuscript text describe that there is a greater number of infiltrating CD8+ T cells. However, it is difficult to discern since the WT and KO dots are together in the same Figure 5B. Can the authors change this to instead have side by side figures with the WT and KO groups as was done for Figure 4C. Alternatively it could also work if a number was given next to the WT and KO labels.

Response: We have revised Figure 5B as follows. The cell count was also labeled in Figure 5B.

11. In lines 272-275 of the Manuscript authors state that “Tumor-infiltrating lymphocytes (TILs) from *Ythdf1*-KO tumors were highly enriched in activated effector T-cell populations, which were assigned largely to the CD8+ TEM, CD8+ TEX, and proliferating CD8+ T-cell states (Figure 5E).” However, there is no statistically significant difference in TEX and proliferating CD8+ cells. The authors should only claim something like “...*Ythdf1*-KO tumors were highly enriched in activated effector T-cell populations of the CD8+ TEM CD8+ T-cell states, while some not statistically significant increase was observed for TEX and proliferating CD8+ T-cell states (Figure 5E).”

Response: Thank you for highlighting this. As suggested by reviewer 3, we excluded Figure 5E and the corresponding description in the revision. Direct comparison of cell numbers in single cell between conditions without any normalization to the actual number of collected cells is not scientifically sound.

12. In Figure 7E, along the legend please include the number of differential TE transcripts for the highlighted m6A+*Ythdf1* target as you have done for the UP and down. Also please be consistent and use either “Up” with “Down” or “UP” with “DOWN” for the legend.

Response: We have revised the manuscript extensively according to the reviewers' comments and supplemented eCLIP-seq, proteomics, and substantial *in vitro* and *in vivo* experiments to further elucidate how tumor-intrinsic YTHDF1 remodels the tumor microenvironment. The updated Figure 7 is different from the previous one.

13. In line 408 there seems to be a typo, the decreased Tcf1 and Lef1 expression is only seen in the WB shown in Figure 8C, not in the confocal microscopy of Figure 8B and 8C. Please revise "(Figure 8B and C)." to "(Figure 8C)."

Response: Thanks for your reminder.

14. Please revise Figure 8G so that both immunodeficient and immunocompetent tumour volumes have the same y-axis range (max value and marks).

Response: Thanks for the suggestion. We have revised the manuscript extensively according to the reviewers' comments and supplemented eCLIP-seq, proteomics, and substantial *in vitro* and *in vivo* experiments to further elucidate how tumor-intrinsic YTHDF1 remodels the tumor microenvironment. The previous Figure 8 has been extensively revised from the revision.

15. Figure 9B has low resolution/pixelated axis labels and numbers. Please correct to have labels and numbers of similar resolution than the other labels throughout the manuscript.

Response: We have improved the resolution/pixelated axis labels and numbers of figure 9B

16. In multiple sections 200ug of exosomes are described to be used, however there are no details as to what is measured to be 200ug (exosome protein or lipid?) or what technique is used to quantify the exosomes (eg microBCA, Bradford, SPV, etc)

Response: Sorry for the missing information. 200ug was exosome protein quantitation determined by microBCA. We have supplemented the corresponding methods in the revision (line 614).

"For *in vivo* exosome treatments, mice received intratumoral injections of 200 µg exosome proteins in 30 µl PBS. The exosome protein quantitation was determined by bicinchoninic acid (BCA) assay."

REVIEWERS' COMMENTS

Reviewer #1 (Remarks to the Author):

The authors have done impressive amount of work during the revision. I am still a little concerned with the number of RNA transcripts bound by YTHDF1. However, i think the mechanistic figure 7, in particular the proteomic data combined with m6A and YTHDF1 targets provide a convincing pathway, which the authors further confirmed on MHC-1 analysis. I do not have additional questions as the this manuscript already contains a huge amount of data already.

Reviewer #2 (Remarks to the Author):

The authors have performed extensive work to address my (and the other reviewers') comments and concerns, which I consider substantially solved.

The new mechanism involving MHC-I is more persuasive than the previously postulated involvement of the Wnt/PYGO role, based on the data presented, and I have no comments on it.

I strongly believe that this is a relevant observation worth reporting in a broad readership journal.

Yet, I also think that it is plausible that a combination of both mechanisms (that is, the MHC-I-dependent and the Wnt/PYGO-dependent ones) might be operating downstream of YTHDF1 action.

It is striking that the authors have completely removed the mentions to the second mechanism (Wnt/PYGO), and I warmly recommend them to either mention it, together with the evidence that is currently available, or to report it somewhere else. It would be a pity if such an interesting observation goes lost or is not published.

Reviewer #3 (Remarks to the Author):

I am happy for the hard work the authors put into addressing my comments. I am satisfied with the results

Reviewer #4 (Remarks to the Author):

The authors have addressed my comments and I believe the manuscript is fit for publication

Point-by-point response

Reviewer #1 (Remarks to the Author)

The authors have done impressive amount of work during the revision. I am still a little concerned with the number of RNA transcripts bound by YTHDF1. However, i think the mechanistic figure 7, in particular the proteomic data combined with m6A and YTHDF1 targets provide a convincing pathway, which the authors further confirmed on MHC-1 analysis. I do not have additional questions as the this manuscript already contains a huge amount of data already.

Response: Thank you so much for your rigorous peer review on our study.

Reviewer #2 (Remarks to the Author)

The authors have performed extensive work to address my (and the other reviewers') comments and concerns, which I consider substantially solved.
The new mechanism involving MHC-I is more persuasive than the previously postulated involvement of the Wnt/PYGO role, based on the data presented, and I have no comments on it.
I strongly believe that this is a relevant observation worth reporting in a broad readership journal.
Yet, I also think that it is plausible that a combination of both mechanisms (that is, the MHC-I-dependent and the Wnt/PYGO-dependent ones) might be operating downstream of YTHDF1 action.
It is striking that the authors have completely removed the mentions to the second mechanism (Wnt/PYGO), and I warmly recommend them to either mention it, together with the evidence that is currently available, or to report it somewhere else. It would be a pity if such an interesting observation goes lost or is not published.

Response: We appreciate you very much for the rigorous peer review and encouragement on our manuscript. The Wnt/PYGO results will be an important piece of work of our in-progress study as our current version already contains a huge amount of data already.

Reviewer #3 (Remarks to the Author):

I am happy for the hard work the authors put into addressing my comments. I am satisfied with the results.

Response: Thank you so much for your rigorous peer review on our study.

Reviewer #4 (Remarks to the Author):

The authors have addressed my comments and I believe the manuscript is fit for publication.

Response: Thank you so much for your rigorous peer review on our study.